# Towards Graph Foundation Models:
# Learning Generalities Across Graphs via Task-Trees

**Zehong Wang** [1]  **Zheyuan Zhang** [1]  **Tianyi Ma** [1]  **Nitesh V Chawla** [1]  **Chuxu Zhang** [2]  **Yanfang Ye** [1] [*]

## Abstract

Foundation models are pretrained on large-scale corpora to learn generalizable patterns across domains and tasks—such as contours, textures, and edges in images, or tokens and sentences in text. In contrast, discovering such generalities in graph-structured data, especially across heterogeneous graph tasks, remains an open challenge. To address this, we propose a novel approach to cross-task generalization in graphs via *task-trees*, which serve as unified learning instances aligning node-, edge-, and graph-level tasks. We theoretically analyze the stability, transferability, and generalization properties of task-trees, showing that pretraining a graph neural network (GNN) on diverse task-trees with a reconstruction objective induces transferable knowledge. This enables efficient adaptation to downstream tasks with minimal fine-tuning. To validate our framework, we introduce Graph Generality Identifier on Task-Trees (GIT), a graph foundation model that demonstrates strong performance on over 30 graphs across five domains via fine-tuning, in-context learning, and zero-shot generalization. Code and data are available at `https://github.com/Zehong-Wang/GIT`.

## 1. Introduction

Foundation models have emerged as a cornerstone of general-purpose machine learning, enabling cross-task and cross-domain generalization. Representative examples include large language models (LLMs) for text (Achiam et al., 2023; Touvron et al., 2023) and large vision models (LVMs) for images (He et al., 2022; Yuan et al., 2021). Pretrained on massive datasets, these models capture trans-

ferable patterns—such as contours and textures in images, or tokens and sentences in text—that reflect modality-specific generalities. This broad knowledge base allows for efficient adaptation to downstream tasks via in-context learning (Xie et al., 2022; Chen et al., 2024c) and zero-shot generalization (Wei et al., 2021).

Despite the success of foundation models in text and vision, their extension to graph-structured data remains nascent (Liu et al., 2024a), primarily due to the high variability across graph datasets (Mao et al., 2024). Graphs from different domains often encode distinct phenomena—e.g., social networks model human relationships (Freeman, 2004), whereas molecular graphs represent chemical structures (Zeng et al., 2022)—leading to both feature (Wang et al., 2024b) and structural heterogeneity (Qiu et al., 2020; Wang et al., 2024c). Crucially, graph tasks operate on different learning units, such as nodes, edges, or entire graphs, limiting cross-task compatibility within a unified model (Wang et al., 2024b). These challenges hinder the development of graph foundation models capable of capturing transferable generalities. In this work, we specifically address the challenge of task heterogeneity.

*Is it possible to identify cross-task generalities across graphs?* Despite inherent challenges, prior work has explored this question via two main approaches. (1) A graph-theoretic perspective employs the concept of graphons (Ruiz et al., 2020) to model transferable patterns across graphs. If graphs are sampled from the same graphon, they are expected to share structural properties, enabling effective transfer (Ruiz et al., 2020; Cao et al., 2023). However, graphon-based methods rely on strong generative assumptions that rarely hold in real-world settings (Levie et al., 2021), and inferring a shared graphon from diverse graphs remains computationally intractable. (2) A substructure-based perspective seeks recurring motifs—such as triangles, stars, or $k$-cliques—across domains (Zhao et al., 2023; Mao et al., 2024). These motifs appear in various contexts (e.g., social, citation, and molecular networks), motivating methods that sample subgraphs consisting of substructures and encode them via GNNs (Sun et al., 2023; Liu et al., 2024a). However, message-passing GNNs are fundamentally limited in capturing such substructures (Garg et al., 2020; Esser

---

[1]University of Notre Dame [2]University of Connecticut. Correspondence to: Zehong Wang <zwang43@nd.edu>, Yanfang Ye <yye7@nd.edu>.

*Proceedings of the 42nd International Conference on Machine Learning*, Vancouver, Canada. PMLR 267, 2025. Copyright 2025 by the author(s).

et al., 2021; Zhang et al., 2024a), restricting their efficacy in learning transferable subgraph representations.

Given the limitations of prior approaches, we introduce a novel perspective centered on the learning dynamics of message-passing GNNs (Kipf & Welling, 2017; Hamilton et al., 2017). In such models, predictions are made based on *task-relevant nodes*: the target node in node-level tasks, edge endpoints in edge-level tasks, and all nodes in graph-level tasks (Srinivasan & Ribeiro, 2020). Regardless of task type, GNNs aggregate embeddings over these task-relevant nodes, which can be conceptualized as introducing a virtual *task node* connected to all task-relevant nodes. We define the computation tree rooted at this virtual node as a *task-tree* (Figure 1). Task-trees offer three key advantages: (1) *Learnability*: tree-structured information can be effectively captured by message-passing GNNs (Gupta et al., 2024); (2) *Uniformity*: task-trees apply seamlessly across node-, edge-, and graph-level tasks, mitigating task heterogeneity; (3) *Efficiency*: encoding task-trees operationally equals to encoding the virtual nodes appended to original graphs. Analogous to images in vision or sentences in language, task-trees serve as unified learning instances and may encode transferable patterns across graph tasks, leading to the assumption:

***Task-Tree Generality Assumption.*** *The generalities shared across graphs are (at least partially) preserved within the task-trees of the involved graphs.*

To evaluate this assumption, we conduct a theoretical analysis of task-trees with respect to stability, transferability, and generalization. Our main result shows that pretraining a GNN on diverse task-trees via a reconstruction objective yields transferable representations that adapt well to downstream tasks with moderate fine-tuning. Furthermore, the model can be specialized to specific domains via post-training (Wei et al., 2021) on domain-specific task-trees.

To empirically validate our theoretical insights, we introduce Graph Generality Identifier on Task-Trees (GIT), a graph foundation model pretrained on task-trees extracted from diverse graphs spanning multiple domains and tasks. GIT is evaluated on 32 graphs across 5 domains under three paradigms: fine-tuning, in-context learning (few-shot without fine-tuning), and zero-shot learning. Results show that pretraining on a small set of graphs significantly improves performance on a broad range of downstream tasks, supporting the hypothesis that task-trees capture transferable generalities. Additionally, we propose an instruction tuning method to adapt the general model to specific domains, yielding performance comparable to or exceeding domain-specific expert models. Our key contributions are:

- We introduce *task-trees* as unified learning instances for aligning heterogeneous graph tasks, demonstrating advantages over conventional units such as subgraphs.

- We present the first theoretical framework addressing task heterogeneity in graph learning, establishing the effectiveness of task-trees for cross-task generalization.

- We propose GIT, a graph foundation model pretrained on task-trees to acquire generalizable knowledge and support domain specialization.

- Extensive experiments across 32 graphs and five domains validate the effectiveness of GIT under fine-tuning, in-context, and zero-shot settings.

## 2. Task-Trees: Rethinking Basic Learning Instances on Graphs

### 2.1. Preliminary

We begin with a brief introduction to message-passing GNNs and some related concepts. Let $\mathcal{G} = (\mathcal{V}, \mathcal{E})$ represent a graph with node set $\mathcal{V}$ and edge set $\mathcal{E}$, where each node $v \in \mathcal{V}$ is associated with a feature vector $\mathbf{x} \in \mathbb{R}^d$. A GNN encoder $\phi$ takes the graph as input and performs message passing to learn node embeddings $\boldsymbol{Z} = \phi(\mathcal{V}, \mathcal{E})$. Specifically, a GNN encoder can be defined as:

$$\boldsymbol{z}_i^{(l)} = \sigma\Big(\boldsymbol{W}_1 \boldsymbol{z}_i^{(l-1)} + \boldsymbol{W}_2 \rho\Big(\textstyle\sum_{j \in \mathcal{N}(i)} g(\boldsymbol{z}_j^{(l-1)})\Big)\Big), \quad (1)$$

where $\mathcal{N}_i$ denotes the 1-hop neighbors of node $i$, $\boldsymbol{z}^{(l)}$ represents the node embedding at the $l$-th GNN layer with $\boldsymbol{z}^{(0)} = \boldsymbol{x}$, and $\boldsymbol{W}_1, \boldsymbol{W}_2$ are learnable matrices. The functions $\sigma$, $\rho$, and $g$ are the activation function, aggregation function and update function, respectively. To simplify the analysis, we assume $\rho$ is an averaging operation and $g$ is the identity function. Without loss of generality (WLOG), these functions can be replaced with any permutation-invariant and Lipschitz-continuous functions, respectively, without affecting the analysis in the paper.

**Definition 2.1** (Task-Relevant Nodes)**.** Graph tasks can be roughly categorized into node-level, edge-level, and graph-level tasks, where the basic learning instances are nodes, edges, and entire graphs, respectively. For node classification, the task-relevant node $v_i^t$ is the node to be classified. In edge classification, the task-relevant nodes are the start and end nodes $\{v_i^t, v_j^t\}$ of the target edge $e_{ij}$. For graph classification, the task-relevant nodes $\{v_i^t\}_{i=1}^{|\mathcal{V}|}$ include all nodes in the target graph $\mathcal{G}$.

For any graph task instance, the prediction relies solely on the embeddings of the corresponding task-relevant nodes. These node embeddings capture the surrounding subtree structures, which are also known as computation trees.

**Definition 2.2** (Computation Trees (Chuang & Jegelka, 2022))**.** Given a node $v$ in graph $\mathcal{G}$, the $L$-layer computation tree $T_v^L$ is constructed by recursively expanding the subtrees of its neighboring nodes, starting with $T_v^1 = v$.

## 2.2. Task-Tree Construction and Encoding

The learning process of message-passing GNNs can be interpreted as recursive aggregation over computation trees, where the representation of a node $v$ produced by an $L$-layer GNN corresponds to the embedding of its $L$-hop computation tree $T_v^L$. Since predictions in graph tasks rely exclusively on the embeddings of task-relevant nodes, and those embeddings are determined by their respective computation trees, we construct a unified *task-tree* for each learning instance—whether a node, edge, or entire graph—by merging the relevant computation trees, as illustrated in Figure 1.

**Definition 2.3** (Task-Trees)**.** For any graph instance—whether a node, edge, or graph—we have a set of task-relevant nodes $\{v_1^t, ..., v_n^t\}$ and their corresponding $L$-layer computation trees $\{T_1, ..., T_n\}$. These computation trees can be reformulated into a larger task-tree $T^t$ by introducing a virtual node that connects all task-relevant nodes.

To encode a task-tree, we adopt a simple yet effective aggregation strategy. Given a task-tree $T^t$ composed of a virtual node $v^t$ and task-relevant nodes $\{v_1^t, \ldots, v_n^t\}$, we compute its representation using a MEAN aggregator over the embeddings of the individual computation trees:

$$z^t = \phi(T^t) = \frac{1}{n} \sum_{i=1}^n \phi(T_i), \qquad (2)$$

where $T_i$ denotes the computation tree rooted at $v_i^t$, and $\phi$ is a shared GNN encoder. This representation serves as the input for downstream objectives such as reconstruction, classification, or alignment.

## 2.3. Comparison to Existing Works

Unlike our proposed task-trees, several existing approaches (Qiu et al., 2020; Sun et al., 2023; Huang et al., 2023; Liu et al., 2024a; He & Hooi, 2024) utilize $k$-hop subgraphs extracted from graphs as the basic learning instances. For instance, in node classification, ego-graphs are constructed around each node, where the label of the central node is assigned to the induced subgraph, effectively reformulating node classification as a subgraph classification task. A similar transformation can be applied to edge-level and graph-level tasks by converting them into subgraph-level learning problems. This method involves: (1) extracting ego-graphs centered around task-relevant nodes and (2) applying GNNs to learn graph-level embeddings for classification. However, this subgraph extraction process incurs substantial computational overhead, increasing both time and memory requirements due to the necessity of storing and processing induced subgraphs. Moreover, information within these subgraphs is not always effectively captured by message-passing GNNs, as GNNs may struggle to learn essential substructures preserved in graphs (Garg et al., 2020; Chen et al., 2020; Zhang et al., 2024a), thereby limiting the efficacy of subgraphs as learning instances.

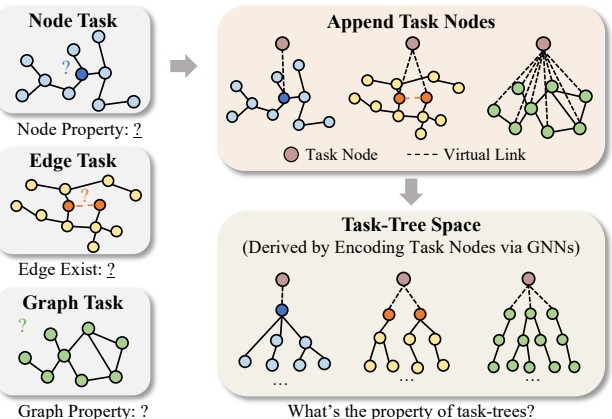

Figure 1: The formulation of task-trees.

In contrast, our proposed task-trees offer both greater efficiency and improved learnability (Table 1). Specifically, encoding task-trees for node, link, or graph-level tasks involves: (1) augmenting the original graph by adding virtual task nodes and connecting them to task-relevant nodes and (2) applying GNNs over the augmented graph to encode the embeddings of these virtual nodes for prediction. For example, in node classification, we first introduce virtual nodes connected to each node in the original graph and subsequently apply GNNs to this augmented structure, allowing virtual node embeddings to be learned for classification. Consequently, our method requires only the addition of nodes and edges to the existing graph, making it significantly more efficient than extracting and storing subgraphs. Furthermore, encoding task-trees is equivalent to directly encoding virtual nodes through message passing, ensuring that task-tree information remains fully learnable by standard GNNs. Empirically, task-trees consistently outperform subgraphs in both effectiveness and efficiency (Section 5.6 and Appendix I).

The most closely related work to our task-tree framework is GFT (Wang et al., 2024b), which introduces computation trees to align heterogeneous graph tasks. While both approaches share the core intuition of structuring task-specific trees, GFT adopts a model-centric perspective, featuring a learnable vocabulary, a multi-faceted reconstruction objective, and specialized adaptation classifiers. In contrast, our task-tree framework is theory-driven and emphasizes the design of learning instances rather than model complexity. Notably, GFT empirically demonstrates the potential of computation trees for transferability, while our work complements it with a formal theoretical foundation, offering a principled understanding of task alignment in graph learning. Together, these works represent complementary advances toward general-purpose graph foundation models. We present additional discussions on related work in Appendix A.

Table 1: Comparison of task alignment methods on graphs.

| | Without Task Alignment | Alignment via Subgraph Tasks | Alignment via Task-Tree Tasks (Ours) |
|---|---|---|---|
| | Node Task · Edge Task · Graph Task
Node Property: ? · Edge Exist: ? · Graph Property: ? | Node Task · Edge Task · Graph Task
(Sub)graph Property: ? | ● Task Node  ----- Virtual Link
Node Task · Edge Task · Graph Task
Task-Tree Property: ? |
| **Input** | Original graphs | Extracted subgraphs | Original graphs with task nodes |
| **Pre-processing** | N/A | Extract subgraphs for each learning instance (node/link/graph) | Append task nodes to the original graphs, connecting each to task-relevant nodes |
| **Encoding** | Apply GNNs to learn node embeddings | Apply GNNs on induced subgraphs to obtain subgraph embeddings | Apply GNNs directly on original graphs to learn task node embeddings |
| **Prediction** | Use node embeddings to derive node/link/graph representations | Use subgraph embeddings as representations for corresponding instances (node/link/graph) | Use task node (task-tree) embeddings as instance representations (node/link/graph) for predictions |
| **Learnability** | - | GNNs struggle to capture fundamental substructures preserved in (sub)graphs | GNNs inherently learn from tree structures, such as computation trees |
| **Efficiency** | - | Storing and encoding subgraphs introduce additional computational overhead | Encoding task-trees involves encoding augmented virtual nodes in the original graphs with minimal computational cost |
| **Theoretical Basis** | - | Lack of a well-established theoretical foundation | Supported by a rigorous theoretical framework |

## 3. Theoretical Analysis of Task-Trees

In this section, we present a theoretical analysis of task-trees, focusing on their stability, transferability, and generalization as foundational learning instances. This analysis provides formal support for the *Task-Tree Generality Assumption*, which posits that transferable patterns across graph tasks are preserved within task-tree structures.

Our goal is not to assert the universal superiority of task-trees over other learning units such as subgraphs, but rather to establish the theoretical plausibility of using task-trees to capture cross-task generalities. By grounding the construction and use of task-trees in formal guarantees, we lay the foundation for principled pretraining and transfer learning across heterogeneous graph tasks.

We begin by examining the stability of GNNs in learning task-tree representations, showing that task-trees with similar subtree structures produce analogous embeddings. To facilitate this analysis, we first define the notation for describing subtree information:

$$\boldsymbol{x}_i^{(l)} = \frac{1}{|\mathcal{N}_i|} \sum_{j \in \mathcal{N}_i} \boldsymbol{x}_j^{(l-1)}, \tag{3}$$

where $\boldsymbol{x}_i^{(0)} = \boldsymbol{x}_i$ denotes the original node feature, and $x^{(l)}$ denotes the subtree information of nodes in $l$-th layer, as illustrated in Figure 2. In this figure, for $l = 1$, only the nodes in the first layer of the tree are considered, and for $l = 2$, only the nodes in the second layer are considered.

**Theorem 3.1** (Stability on Task-Trees). *Given two $L$-layer*

*task-trees $T_t^1$ and $T_t^2$, with task-relevant nodes $\{v_1, ..., v_n\}$ and $\{v_1, ..., v_m\}$, respectively. The distance between task-trees is defined as $\Delta := \|\phi(T_1^t) - \phi(T_2^t)\|$ with*

$$\Delta = \|\phi(T_1^t) - \phi(T_2^t)\| = \|\frac{1}{n}\sum_{i=1}^{n}\phi(T_i) - \frac{1}{m}\sum_{j=1}^{m}\phi(T_j)\|$$

$$\leq \frac{1}{nm}\sum_{i=1}^{n}\sum_{j=1}^{m}\Big(\mathcal{C}_1\|\boldsymbol{x}_i^{(0)} - \boldsymbol{x}_j^{(0)}\| + ... \tag{4}$$

$$+ \mathcal{C}_1\mathcal{C}_2^{L-1}\|\boldsymbol{x}_i^{(L-1)} - \boldsymbol{x}_j^{(L-1)}\|\Big) \leq 2\mathcal{B}_{\boldsymbol{x}} \cdot \mathcal{C}_1\frac{\mathcal{C}_2^L - 1}{\mathcal{C}_2 - 1},$$

*where $\phi$ is the GNN encoder, $T_i$ is the computation tree corresponding to node $i$, and $\mathcal{C}_1, \mathcal{C}_2$ are constants related to the encoder, and $\mathcal{B}_{\boldsymbol{x}}$ represents the bounded norm of $\boldsymbol{x}$.*

Theorem 3.1 (proved in Appendix D.1) suggests that two task-trees are likely to have similar representations if their subtrees are similar. This theorem highlights the significance of similarity between pairs of subtrees, while downplaying the impact of the number of subtrees (i.e., the width of the task-trees), despite having more subtrees could potentially increase diversity and thus magnify discrepancy. The theorem also implies that increasing the number of GNN layers may lead to a loose bound, which aligns with previous analyses (Garg et al., 2020; Ju et al., 2023a).

**Illustration 3.2.** *This theorem provides theoretical support for using task-trees as basic learning instances in graph tasks. Consider two task-trees: one representing a node (with a single subtree) and the other representing a graph*

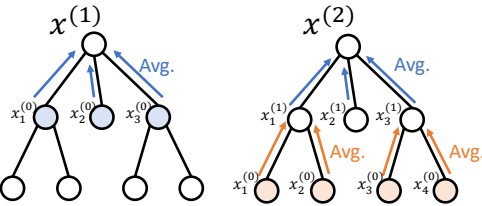

Figure 2: subtree information examples.

*(with multiple subtrees). While the widths of these task-trees differ significantly, if their subtrees share some degree of similarity, they can produce similar representations. Thus, this theorem ensures that task-trees of nodes, edges, or graphs can potentially be similar, making it possible to use a GNN encoder to capture the shared patterns among them.*

We now examine the transferability of task-trees. Specifically, assuming a model is pretrained on a task-tree reconstruction task[1], we aim to quantify how the knowledge acquired during pretraining can be transferred to downstream tasks. The pretraining objective is defined as $\mathcal{L}_{\mathcal{P}}(g \circ \phi) := \mathbb{E}_{(\hat{T},T) \sim \mathcal{P}} \|g(\phi(\hat{T})) - \phi(T)\|^2$, where $\mathcal{P}$ represents the task-tree distribution used for pretraining, $\phi \in \Phi$ and $g \in G$ are the GNN encoder and the reconstruction head, respectively. $T$ denotes the task-tree and $\hat{T}$ is the corrupted version of $T$, generated using arbitrary augmentations. Note that the reconstruction head $g$ is used only during pretraining and is discarded during fine-tuning. Then, we define the risk on downstream task as $\mathcal{R}_{\mathcal{T}}(f \circ \phi) := \mathbb{E}_{(T,y) \sim \mathcal{T}} \kappa(f(\phi(T)), y)$, where $f \in \mathcal{F}$ is a linear head for predictions, $\mathcal{T}$ represents the downstream task distribution with task-tree $T$ and label $y$, and $\kappa$ denotes the loss function.

**Theorem 3.3** (Transferability on Task-Trees). *Given two task-tree encoders $\phi, \phi' \in \Phi$, we have*

$$\min_{f \in \mathcal{F}} \mathcal{R}_{\mathcal{T}}(f \circ \phi) - \min_{f' \in \mathcal{F}} \mathcal{R}_{\mathcal{T}}(f' \circ \phi')$$
$$\leq \mathcal{C}_\delta \Big( \min_{g \in G} \mathcal{L}_{\mathcal{P}}(g \circ \phi) - \min_{g' \in G} \mathcal{L}_{\mathcal{P}}(g' \circ \phi') \Big)^\delta, \quad (5)$$

*where $\mathcal{C}_\delta \approx O(1)$ and $\delta = \frac{1}{2}$.*

The proof is provided in Appendix D.2. In summary, Theorem 3.3 demonstrates that knowledge gained through pretraining on task-tree reconstruction tasks is transferable to downstream tasks, and it quantifies the extent of this transfer. The left-hand side (LHS) of the theorem shows how different representations impact performance on downstream tasks, while the right-hand side (RHS) reflects the difference in pretraining losses between two encoders. Therefore, if

---

[1]The scope of reconstruction task is large. We consider contrastive learning is also a kind of reconstruction.

two encoders exhibit similar losses during pretraining, their transferability to a new task should be comparable.

**Illustration 3.4.** *To give a better understanding on why Theorem 3.3 imply the model pretrained on task-trees can bring transferable information to downstream tasks, we present an example. Let's consider the case where $\phi$ is the pretrained encoder and $\phi'$ is a randomly initialized encoder. The LHS term $\min_{f \in \mathcal{F}} \mathcal{R}_{\mathcal{T}}(f \circ \phi) - \min_{f' \in \mathcal{F}} \mathcal{R}_{\mathcal{T}}(f' \circ \phi')$ measures the amount of knowledge that is acquired during pretraining and is capable to be transferred to downstream tasks, and the RHS term $\min_{g \in G} \mathcal{L}_{\mathcal{P}}(g \circ \phi) - \min_{g' \in G} \mathcal{L}_{\mathcal{P}}(g' \circ \phi')$ measures the total knowledge acquired during pretraining. Thus, the constants $\mathcal{C}_\delta$ and $\delta$ quantify how much of this knowledge is transferable to downstream tasks. Since both $\mathcal{C}_\delta$ and $\delta$ are reasonably small, we conclude that pretraining on task-trees provides sufficient knowledge to benefit downstream tasks.*

To further explain why the task-tree-based pretraining and fine-tuning framework is effective for downstream tasks, we derive the following generalization bound.

**Theorem 3.5** (Generalization on Task-Trees). *Given two task-tree distributions, $\mathcal{P}$ for pretraining and $\mathcal{T}$ for fine-tuning, suppose the encoder $\phi$ is pretrained on a set of task-trees $\{T_i\}_{i=1}^m$ sampled from $\mathcal{P}$ and finetuned on task-trees $\{T_i\}_{i=1}^n$ sampled from $\mathcal{T}$, the generalization bound of the finetuned model, with probability at least $1 - v$, is*

$$\mathcal{R}_{\mathcal{T}}(f \circ \phi) \leq \min_{f' \in \mathcal{F}} \mathcal{R}_{\mathcal{T}}(f' \circ \phi^*)$$
$$+ 2\mathcal{C}_2 \Big( \sum_{x \in \mathcal{X}_\phi} \big\| \mathcal{T}_\phi(x) - \mathcal{P}_\phi(x) \big\| + 2\sqrt{\frac{\log(1/v)}{n}} \Big)$$
$$+ \mathcal{C}_\delta \Big( \mathcal{E}_{\mathcal{P}}(g, \phi) \Big)^\delta + \frac{4\mathcal{C}_1}{n} \sqrt{\sum_{i=1}^n \big\| \phi(T_i) \big\|^2}, \quad (6)$$

*where $\phi^* = \arg\min_{\phi \in \Phi} \min_{g \in G} \mathcal{L}_{\mathcal{P}}(g \circ \phi)$ is the optimal task-tree encoder obtained on $\mathcal{P}$, $\mathcal{E}_{\mathcal{P}}(g, \phi) = \mathcal{L}_{\mathcal{P}}(g \circ h) - \min_{g' \in G, \phi' \in \Phi} \mathcal{L}_{\mathcal{P}}(g' \circ \phi')$ defines the excess risk during pretraining. Constants $\mathcal{C}_1$ and $\mathcal{C}_2$ are related to downstream tasks, while $\mathcal{C}_\delta \approx O(1)$ and $\delta = \frac{1}{2}$ are the same as Theorem 3.3. $\mathcal{X}_\phi$ denotes the distribution of task-tree embeddings encoded via $\phi$, and $\|\mathcal{T}_\phi(x) - \mathcal{P}_\phi(x)\|$ measures the distance between task-tree distributions of pretraining and fine-tuning data.*

The proof can be found in Appendix D.3. This theorem outlines key factors affecting model generalization on downstream tasks, such as the transferability of task-trees ($\mathcal{C}_\delta(\mathcal{E}_{\mathcal{P}}(g, \phi))^\delta$) and the quality of the pretrained encoder ($\mathcal{E}_{\mathcal{P}}(g, \phi)$). With regard to the number of task-trees, we find that while increasing the number of fine-tuning samples contributes to more stable optimization ($\frac{4\mathcal{C}_1}{n} \sqrt{\sum_{i=1}^n \|\phi(T_i)\|^2}$), it does not significantly reduce the generalization bound

$(2\sqrt{\frac{\log(1/v)}{n}})$. This provides theoretical evidence that a reasonable number of fine-tuning samples can be sufficient for training a model with strong generalization capabilities. Moreover, the discrepancy between the pretraining and fine-tuning distributions ($\sum_{x \in \mathcal{X}_\phi} \|\mathcal{T}_\phi(x) - \mathcal{P}_\phi(x)\|$) is crucial—smaller distribution gaps lead to better generalization. This highlights the importance of increasing the diversity of pretraining data, which provides a boarder pretraining distribution $\mathcal{P}$. It also supports the potential of developing specialized models for specific domains based on a pretrained general model, discussed in Section 4.2.

# 4. Graph Generality Identifier on Task-Trees

The theoretical analysis establishes the feasibility of constructing graph foundation models based on task-trees. Building on these insights, we develop the GIT model to empirically validate the *Task-Tree Generality Assumption*.

To focus on aligning task spaces across heterogeneous graph tasks, we adopt widely used text-attributed graph benchmarks (Chen et al., 2024b; Zhang et al., 2024b; Feng et al., 2024), which simplify feature alignment across datasets. Specifically, we follow Liu et al. (2024a) and encode all node features into a shared 768-dimensional embedding space using Sentence-BERT (Reimers & Gurevych, 2019). This design allows us to isolate and examine the effect of task-tree-based pretraining while holding node features consistent across domains.

## 4.1. GIT-G: Pretraining to Acquire General Knowledge

We propose a task-tree reconstruction task as a pretext for pretraining. The key is to use two corrupted task-trees to reconstruct each other, thereby capturing corruption-invariant semantics. Given a set of task-trees $\{T_1^t, ..., T_n^t\}$ sampled from a graph database, we apply corruption techniques to generate two views of each task-tree, denoted as $\{\hat{T}_1^t, ..., \hat{T}_n^t\}$ and $\{\tilde{T}_1^t, ..., \tilde{T}_n^t\}$. For corruption, we use random edge masking and random attribute masking (Zhu et al., 2020; Wang et al., 2025c) due to its computational efficiency. We then use an encoder $\phi$ to obtain embeddings for the corrupted task-trees, resulting in $\{\hat{z}_1, ..., \hat{z}_n\}$ and $\{\tilde{z}_1, ..., \tilde{z}_n\}$. Inspired by (Thakoor et al., 2022), we perform reconstruction as

$$\mathcal{L} = \frac{1}{2n} \sum_{i=1}^{n} \Big[ \|\rho(g(\hat{z}_i)) - \text{sg}[\rho(\tilde{z}_i)]\|^2$$

$$+ \|\rho(g(\tilde{z}_i)) - \text{sg}[\rho(\hat{z}_i)]\|^2 \Big] + \sum_{i=1}^{n} D_{\text{KL}}(\boldsymbol{h} \| z_i), \quad (7)$$

where $g$ is a non-linear MLP projector, $\rho(\boldsymbol{z}) = (\boldsymbol{z}/\|\boldsymbol{z}\|)$ serves for normalization, sg is the stop-gradient operation, and $\boldsymbol{h}$ is the average of all instances $\boldsymbol{z}$. The reconstruction

loss captures the semantics of the task-trees in a predictive manner, while the KL regularizer ensures the embeddings are projected into a shared space by minimizing the KL divergence between individual instances and their center.

## 4.2. GIT-S: Specification via Instruction Tuning

Theorem 3.5 highlights the relationship between model generalization and the distribution gap between pretraining data $\mathcal{P}$ and fine-tuning data $\mathcal{T}$, showing that a smaller gap leads to better generalization. Based on this finding, it is feasible to develop a specialized model for specific domains from a pretrained general model. This is based on the mild assumption that *graphs from the same domain have similar task-tree distributions* $\{\mathcal{T}_1, .., \mathcal{T}_n\}$. If the pretrained model is post-trained on a task-tree distribution $\mathcal{P}_{post}$ sampled from $\{\mathcal{T}_1, .., \mathcal{T}_n\}$, the pretraining data distribution $\mathcal{P}$ can be adjusted towards these task-tree distributions. This reduces the discrepancy $\sum_{x \in \mathcal{X}_\phi} \|\mathcal{T}_\phi(x) - \mathcal{P}_\phi(x)\|$ in Theorem 3.5, thereby improving model generalization on the target domain. To achieve this, we propose an instruction-tuning method for post-training the pretrained model.

Instruction tuning is a supervised fine-tuning (SFT) technique designed to enhance the capabilities of a pretrained model by post-training it on a small dataset. Our goal is to fine-tune the model using instructions to specialize it for a particular domain of interest. Given a pretrained model $\phi^*$ and a set of task-trees $\{T_1, ..., T_n\}$ from the target domain, we post-train the model using the SFT loss:

$$\mathcal{L}_{SFT} = \frac{1}{n} \sum_{i=1}^{n} \kappa(\phi^*(T_i), \psi(T_i)), \quad (8)$$

where $\psi$ is the instruction generation function for each task-tree, and $\kappa$ is the corresponding loss function. In this paper, as we use text-attributed graphs in our experiments, we define instructions as the embeddings of label descriptions encoded by a LLM, which is similar to Liu et al. (2024a), and we use mean squared error as the loss function $\kappa$. Additional model analysis is provided in Appendix B.

# 5. Experiment

## 5.1. Experimental Setup

**Datasets.** We conduct experiments on over 30 text-attributed graphs spanning five domains: academic networks, e-commerce networks, knowledge graphs, molecular graphs, and temporal graphs. Pretraining is performed on a diverse subset including Arxiv (academic), Products (e-commerce), WN18RR and FB15K237 (knowledge), and Chemblpre and PCBA (molecular). Specialization is evaluated on representative datasets for each domain: Arxiv, Products, FB15K237, and PCBA. For temporal graphs, which are e-commerce temporal graphs, we also use

Table 2: We report the model performance across five graph domains: academia, e-commerce, knowledge base, molecular, and temporal graphs, with results averaged over all graphs within each domain. Note that -G and -S represent the general and specialized versions of GIT, respectively. The comprehensive results can be found in Appendix F.

| | Domain | Academic | E-commerce | KG | Molecule | Temporal | Held-out Avg. | Avg. |
|---|---|---|---|---|---|---|---|---|
| **0-shot** | Sup. GNN | - | - | - | - | - | - | - |
| | GraphMAE | 15.42 | 8.19 | - | 47.19 | - | 26.67 | 25.11 |
| | OFA | 13.98 | 8.73 | - | 50.49 | - | 27.20 | 26.14 |
| | GIT - G | 14.88 | 8.79 | - | 53.34 | - | 28.56 | 27.50 |
| | GIT - S | **23.45** | **17.06** | - | **62.83** | - | **35.19** | **36.32** |
| **3-shot** | Sup. GNN | - | - | - | - | - | - | - |
| | GraphMAE | 49.25 | 48.20 | 56.56 | 56.01 | 40.31 | 50.15 | 52.07 |
| | OFA | 45.93 | 57.06 | 56.97 | 57.03 | 38.92 | 51.84 | 53.70 |
| | GIT - G | 54.00 | 57.22 | 67.55 | 55.96 | 39.95 | 56.09 | 57.82 |
| | GIT - S | **55.18** | **58.01** | **67.80** | **62.82** | **41.38** | **58.69** | **60.15** |
| **Finetune** | Sup. GNN | 73.57 | 78.21 | 66.86 | 73.65 | 62.61 | 71.14 | 72.25 |
| | GraphMAE | 73.81 | 76.57 | 72.61 | 71.41 | 62.75 | 71.37 | 72.79 |
| | OFA | 72.18 | 76.64 | 72.38 | 74.03 | 62.31 | 71.48 | 73.08 |
| | GIT - G | 75.82 | 78.55 | 75.73 | 74.57 | 64.59 | 73.84 | 75.37 |
| | GIT - S | **75.88** | **78.83** | **76.15** | **75.20** | **64.68** | **74.19** | **75.72** |

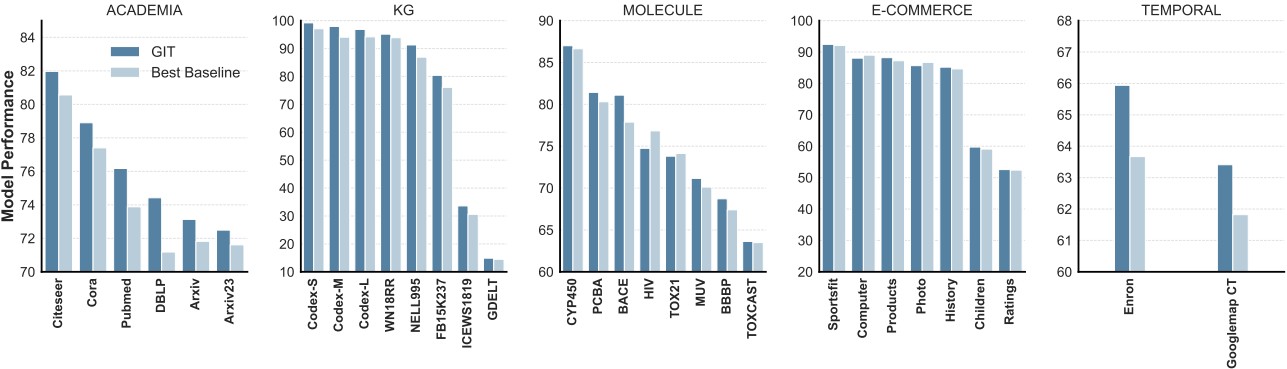

Figure 3: The model performance on all datasets in the fine-tuning setting. For GIT, the best result is selected between GIT-G and GIT-S, while the best baseline performance is chosen among GCN/GAT/GIN, BGRL, GraphMAE, and OFA.

`Products` for SFT to assess robustness under temporal distribution shifts. We provide the full dataset details in Appendix E.1.

**Baselines.** We compare against a broad spectrum of baselines, including supervised GNNs (GCN (Kipf & Welling, 2017), GAT (Veličković et al., 2018), GIN (Xu et al., 2019)), self-supervised models (BGRL (Thakoor et al., 2022), GraphMAE (Hou et al., 2022)), and graph foundation models (OFA (Liu et al., 2024a), GraphPrompt+ (Liu et al., 2023), All in One (Sun et al., 2023), OpenGraph (Xia et al., 2024), AnyGraph (Xia & Huang, 2024)). We also include domain-specific expert models for comparison. See Appendix E.2 for full descriptions.

**Experimental Protocols.** We use GraphSAGE (Hamilton et al., 2017) as the encoder and repeat each experiment five times with different random seeds. We evaluate three paradigms: fine-tuning, in-context learning, and zero-shot learning. Fine-tuning updates all model parameters on downstream data. In in-context learning (few-shot without fine-tuning), we randomly sample $k$ instances per class, compute prototype embeddings via averaging, and classify using nearest-prototype inference. Following Liu et al. (2024a); He & Hooi (2024), we sample 500 5-way 3-shot tasks; if the number of classes is fewer than 5, we use the actual number of classes as the number of ways. Zero-shot learning replaces prototypes with class description embeddings generated by an LLM. For evaluation, we report accuracy for node classification and edge classification, and AUC for graph classification and link prediction. Additional details are provided in Appendix E.

Table 3: Performance comparison to SOTA methods.

|  | Academic | KG | Molecule |
|---|---|---|---|
| GraphPrompt+ | 74.80 | 74.78 | 72.99 |
| All in one | 75.25 | 74.92 | 71.87 |
| OpenGraph | 74.64 | 71.38 | 72.84 |
| AnyGraph | 75.01 | 74.30 | 72.49 |
| GIT - G | **75.82** | **75.73** | **74.57** |

Table 4: Ablation study on training strategies, where the performance is averaged across all academic networks.

|  | Base Model | | | Expert Model | | |
|---|---|---|---|---|---|---|
|  | 0-shot | 3-shot | Finetune | 0-shot | 3-shot | Finetune |
| GraphMAE | 15.30 | 51.51 | **75.57** | 17.89 | **55.88** | 75.26 |
| OFA | 14.19 | 50.15 | 75.12 | 17.26 | 54.88 | **75.97** |
| GIT | **15.36** | **53.31** | 75.53 | **18.38** | 55.10 | 75.47 |

|  | General Model | | | Specialized Model | | |
|---|---|---|---|---|---|---|
|  | 0-shot | 3-shot | Finetune | 0-shot | 3-shot | Finetune |
| GraphMAE | **15.42** | 49.25 | 73.81 | 20.31 | 51.21 | 74.05 |
| OFA | 13.98 | 45.93 | 72.18 | 20.05 | 46.87 | 73.04 |
| GIT | 14.88 | **54.00** | **75.82** | **23.45** | **55.18** | **75.88** |

## 5.2. Main Results

**Domain-Wise Performance.** Table 2 reports average performance across five domains, with detailed per-dataset results provided in Appendix F. The "held-out avg." denotes the average score on all graphs excluded from pretraining and specialization, offering an unbiased evaluation of generalization. Notably, the general-purpose model GIT-G already outperforms strong baselines in several domains. With domain-specific specialization, GIT-S achieves further gains—especially in zero-shot and in-context settings—highlighting the benefits of post-training in adapting pretrained models to specific domains. These results align with our theoretical findings, demonstrating enhanced adaptability through specialization.

**Dataset-Wise Performance.** Figure 3 shows fine-tuning performance across all datasets. GIT consistently outperforms the strongest baseline in the majority of cases, validating the effectiveness of task-trees as generalizable learning units across heterogeneous graph tasks.

**Effect of Specialization.** We summarize three key observations: (1) Specialization (GIT-S) enhances the performance of GIT-G across most settings (Table 2). However, the impact of specialization varies depending on the dataset used (Appendix Figure 9 and Table 25). (2) Specialization does not significantly degrade model performance on other domains (Table 27). (3) Applying the proposed specialization approach to other models also yields performance improvements in specific domains (Table 4), showing the generalization of post-training in enhancing model capacity.

## 5.3. Comparison to State-of-The-Art Methods

To evaluate the effectiveness of GIT, we compare it against recent graph foundation models, including GraphPrompt+ (Liu et al., 2023), All in One (Sun et al., 2023), OpenGraph (Xia et al., 2024), and AnyGraph (Xia & Huang, 2024), under the pretrain-then-finetune paradigm. As shown in Table 3, we benchmark GIT-G and all baselines across three representative domains: academic networks (node classification), knowledge graphs (edge classification), and molecular graphs (graph classification). GIT-G consistently outperforms all competing methods across tasks and domains. While many existing models pursue broad generalization by

addressing multiple aspects—e.g., domain shifts, modality fusion, and prompt engineering—GIT adopts a targeted approach centered on task alignment. Its core innovation lies in the use of *task-trees*, which provide a unified abstraction across graph tasks.

## 5.4. Ablation on Training Strategies

We conduct an ablation study on academic networks to evaluate the impact of different training strategies. Specifically, we examine four approaches: (1) *Base Model*: Pretraining on the target graph. (2) *Expert Model*: Pretraining on all academic networks. (3) *General Model*: Pretraining on the default pretraining datasets. (4) *Specialized Model*: Pretraining on the default datasets followed by specialization on `Arxiv`. The results, averaged across all academic graphs, are reported in Table 4. Notably, the general model of GIT maintains stable performance relative to both the base and expert models. In contrast, GraphMAE and OFA exhibit performance degradation when transitioning from the base and expert models to the general model. This finding underscores the potential of task-trees in mitigating such negative transfer. Furthermore, specialization in GIT allows its performance to closely approximate that of expert models trained exclusively on academic graphs.

## 5.5. Comparison to Domain Experts

We compare GIT to domain experts in molecular and knowledge graphs, as these domains require domain-specific knowledge. Our findings indicate that the specialized GIT (GIT-S) closely approximates the performance of these experts. In molecular graphs, GIT-S achieves an average performance of 62.83, ranking second to the state-of-the-art GIMLET (64.15) (Zhao et al., 2023) while outperforming other domain experts (Zeng et al., 2022; Su et al., 2022; Taylor et al., 2022), whose best performance is 55.82 (Table 24). Similarly, in knowledge graphs, GIT-S attains an average score of 67.80, approaching the KG expert ULTRA (68.53) (Galkin et al., 2024), as shown in Table 23 in Appendix.

Table 5: Performance comparison between methods with different basic learning instances.

| Instances | Subgraph | | Tree |
|---|---|---|---|
| Domains | OFA | GIT - SubG | GIT - Tree |
| Academia | 72.18 | 73.48 | **75.82** |
| KG | 72.38 | 73.59 | **75.73** |
| Molecule | 74.03 | 72.67 | **75.73** |
| Held-out Avg. | 71.31 | 70.88 | **73.81** |
| Avg. | 72.93 | 73.01 | **75.33** |

Table 6: Results on non-text-attributed graphs.

| Setting | GraphMAE | GIT-G |
|---|---|---|
| w/o SVD + w/o pretrain | **95.02** | 94.27 |
| w. SVD + w/o pretrain | 94.89 | **95.50** |
| **w. SVD + w. pretrain** | 95.10 | **95.70** |

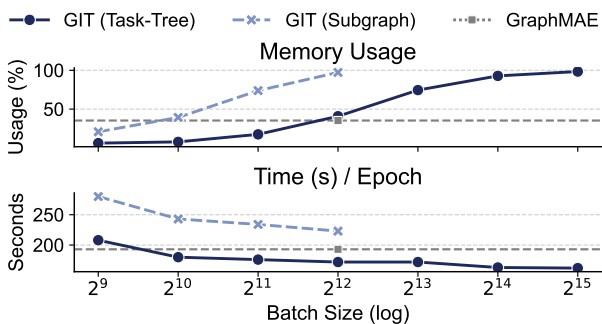

Figure 4: Training efficiency between task-tree and subgraph versions of GIT.

### 5.6. Task-Trees vs. Subgraphs: Efficiency and Performance

We compare the efficiency and effectiveness of task-trees against subgraph-based learning units in the fine-tuning setting, as shown in Figure 4 and Table 5. For a fair comparison, we implement a subgraph-based variant of our model, denoted **GIT-SubG**, by substituting task-trees with subgraphs while keeping all other components unchanged.

Experimental results across three domains—academic (node classification), knowledge graphs (edge classification), and molecular graphs (graph classification)—demonstrate that task-trees consistently outperform subgraphs in both computational efficiency (Figure 4) and predictive performance (Table 5). This highlights the advantage of task-trees in serving as compact, expressive, and structurally aligned learning instances, especially in cross-task and cross-domain generalization scenarios.

### 5.7. Results on Non-Text-Attributed Graphs

While prior experiments use text-attributed graphs, GIT does not inherently rely on textual information. We adopt text-attributed datasets to isolate the impact of task heterogeneity, while sidestepping the confounding effects of feature heterogeneity. Textual attributes enable feature alignment across graphs via a shared encoder, allowing us to more clearly assess the benefits of task-tree generalization. Importantly, GIT is compatible with non-textual graphs. To demonstrate this, we introduce a lightweight module that addresses feature heterogeneity by applying SVD to project features into a shared space. As shown in Table 6, we pretrain GIT-G on non-text-attributed datasets—PubMed (node classification), Citeseer (link prediction), and IMDB-

B (graph classification)—and fine-tune on Cora (link prediction). Results, evaluated via AUC, confirm that GIT-G remains effective without relying on textual information, validating its broader applicability.

## 6. Conclusion and Limitations

We introduce *task-trees* as unified learning instances for aligning heterogeneous graph-based tasks, supported by both theoretical and empirical analyses. Based on this abstraction, we develop GIT, a pretrained graph foundation model that leverages a small set of source graphs to generalize across over 30 datasets spanning five diverse domains. Our results demonstrate that task-trees preserve transferable patterns across tasks, offering a scalable and principled pathway for cross-domain generalization in graph learning.

**Limitations and Future Work.** While task-trees serve as a promising abstraction—analogous to images in vision and sentences in language—several challenges remain. First, although our results suggest that task-trees capture generalities across tasks, pinpointing the precise transferable patterns remains an open question. Future work should explore this from both theoretical and empirical perspectives. Second, transforming graph neighborhoods into tree structures may lead to information loss due to the limitations of message-passing GNNs. This can affect model expressiveness, especially for graphs with complex higher-order structures. Incorporating more expressive architectures, such as higher-order GNNs (Morris et al., 2019; 2023) or advanced graph modeling (Wang et al., 2025b), may help mitigate this issue. Finally, while GIT demonstrates strong performance with a lightweight design, exploring more sophisticated variants—e.g., incorporating attention mechanisms, graph prompting, or task-conditioned decoders—offers exciting directions for extending the generality and adaptability of graph foundation models. We outline several of these opportunities in Appendix C.

## Acknowledgments

This work was partially supported by the NSF under grants IIS-2321504, IIS-2334193, IIS-2340346, IIS-2217239, CNS-2426514, and CMMI-2146076, and Notre Dame Strategic Framework Research Grant (2025). Any opinions, findings, and conclusions or recommendations expressed in this material are those of the authors and do not necessarily reflect the views of the sponsors.

## Impact Statement

From an industry perspective, we offer GIT as a foundational tool for graph-structured data. Additionally, since GIT can be quickly adapted to specific domains, we hope it will support applications where label acquisition is difficult and model training is time-consuming. There is no clear ethical considerations on the model architecture itself.

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

# A. Related Work

**Graph Neural Networks.** GNNs are a class of learning models specifically designed to operate on graph-structured data and have demonstrated substantial success across a variety of domains. Their strength lies in their ability to perform relational learning, where information from neighboring nodes is aggregated and used to enhance node representations. For instance, GCN (Kipf & Welling, 2017) utilizes message-passing to aggregate information from neighboring nodes to central nodes. Building on this, models such as GraphSAGE (Hamilton et al., 2017) and GAT (Veličković et al., 2018) introduce innovative techniques like neighborhood sampling and attention mechanisms, respectively, further advancing performance on graph learning tasks. However, these methods are limited to solving a single task by training from the scratch (Zhang et al., 2019; Fan et al., 2022; Qian et al., 2025; Ma et al., 2023; Qian et al., 2024).

**Transferability of GNNs.** Existing works that analyze the shared concepts (generalities) across different graphs primarily follow two approaches. The first is graphon theory, which provides bounds on the distance between graphs generated from the same graphon. This method has been used to study transferability in pretraining and fine-tuning settings (Cao et al., 2023), to develop more expressive fine-tuning techniques (Sun et al., 2024), and to design new model architectures (Ruiz et al., 2020). However, despite its theoretical advantages, graphon-based approaches face practical challenges, particularly the strong assumptions required and the difficulty of identifying graphons in large-scale graphs, which limits their applicability in building graph foundation models. The second approach involves leveraging substructures within graphs to identify transferable patterns (Mao et al., 2024). This method focuses on extracting subgraphs composed of meaningful substructures for prediction tasks. While this approach offers theoretical insights into stability (Levie et al., 2019; Zhu et al., 2021), it struggles to fully capture substructures that are beneficial for downstream tasks (Zhang et al., 2024a).

**Graph Foundation Models.** Foundation models are designed as general-purpose solvers capable of handling various tasks across different domains. For instance, LLMs, the foundation models in natural language processing, are capable of performing tasks such as summarization, translation, and entity recognition, as well as question-answering. However, building such versatile foundation models for graphs presents unique challenges due to the inherent feature, structural, and task heterogeneity across different graph domains and tasks. To address these challenges, Qiu et al. (2020) pretrained GNNs using subgraphs as basic units, mitigating structural heterogeneity. Building on this, Sun et al. (2023) reformulated node-, edge-, and graph-level tasks into subgraph-level tasks, tackling task heterogeneity. Additionally, Huang et al. (2023) and Liu et al. (2024a) applied LLMs to unify the feature spaces of cross-domain graphs, addressing feature heterogeneity. These approaches enable models to operate on cross-domain and cross-task graphs. Further advancements, such as He & Hooi (2024) and Li et al. (2024), improve node embeddings by jointly optimizing GNN and LLM encoders, facilitating various downstream tasks like few-shot learning and zero-shot learning. However, most of these approaches rely on subgraphs as the primary learning instances, which can result in inefficient training and reduced expressiveness, as discussed in the main paper. Other efforts to resolve feature heterogeneity include methods like singular vector decomposition (SVD) (Zhao et al., 2024a; Yu et al., 2024), non-parametric encoders (Zhao et al., 2024b), or synthesizing text-attributed graphs (Wang et al., 2025a). Notably, a recent study by Wang et al. (2024b) explores computation trees as transferable patterns in graphs. However, our work differs in three key aspects: (1) While Wang et al. (2024b) focus on identifying transferable patterns across graphs, our approach is centered on designing fundamental learning instances. (2) Our theoretical framework also serves as a foundational basis for Wang et al. (2024b). (3) Our proposed GIT significantly simplifies the model proposed by Wang et al. (2024b), demonstrating a minimally applicable design that retains effectiveness.

Another line of research focuses on designing GFMs for single tasks or domains, thereby avoiding the complexities of feature, structural, or task heterogeneity. For example, Galkin et al. (2024) propose a foundation model for reasoning tasks on knowledge graphs, using triplets as basic transferable patterns. Zhao et al. (2023) introduce a foundation model for molecular graphs, employing LLMs to align semantics between datasets and encode key motifs. In node classification, Li et al. (2024) propose a zero-shot learning foundation model, while Zhao et al. (2024a) present a feature alignment method based on SVD for node-level graph foundation models. Recently, Zhao et al. (2024b) designed a foundation model for node classification using a non-parametric classifier. Meanwhile, Chen et al. (2024a), Tang et al. (2024), Guo et al. (2023), Liu et al. (2024b), and Wang et al. (2024a) have explored using LLMs as graph reasoners to solve graph tasks, similar to their role in vision language models. While these methods excel at specific tasks or domains, they are not suitable as general graph solvers across diverse tasks. In contrast to these approaches, our proposed GIT model is pretrained on diverse task trees to acquire general reasoning capabilities, allowing it to quickly specialize in specific domains through instruction tuning.

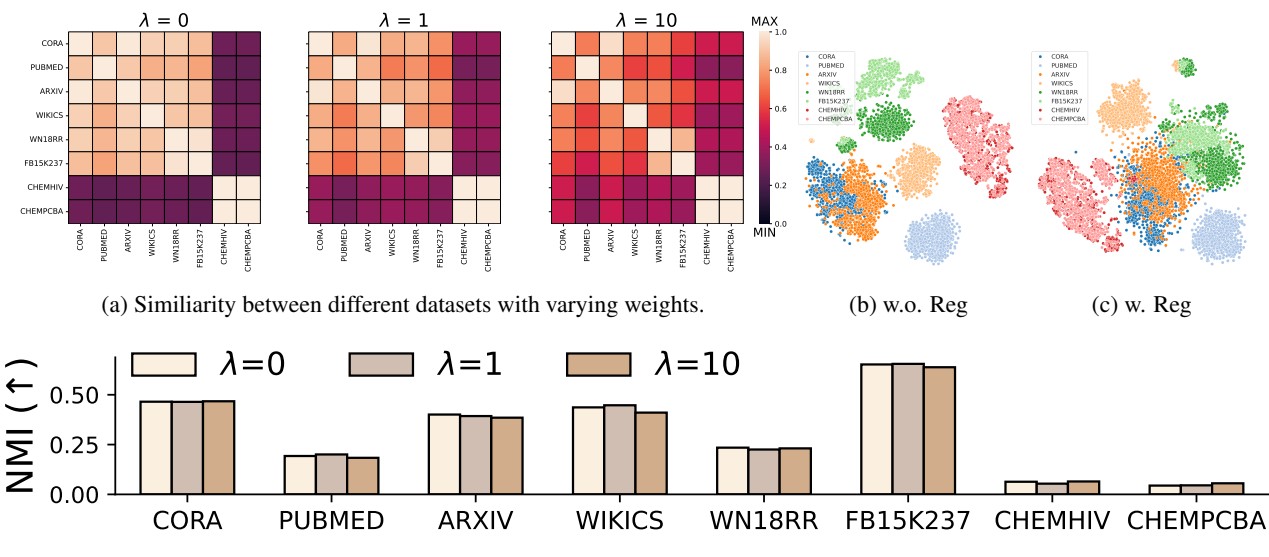

(a) Similiarity between different datasets with varying weights.  (b) w.o. Reg  (c) w. Reg

(d) The regularizer marginally affects the task structures for each dataset.

Figure 5: The domain regularizer controls the distance between datasets while preserving the structure within each of them.

Table 7: Model performance across settings with different scaling weights of domain regularizer.

|          | $\lambda = 0$ | $\lambda = 1$ | $\lambda = 10$ |
| -------- | ------------- | ------------- | -------------- |
| 0-shot   | 20.39         | 27.55         | **29.60**      |
| 3-shot   | 53.10         | 57.53         | **60.21**      |
| Finetune | 74.78         | **75.41**     | 75.37          |

## B. Additional Model Discussion

### B.1. Why Does the General Model Need Specialization?

It is challenging for a single graph model to handle tasks across various domains due to pattern conflicts, where the same structural pattern can have different meanings in different domains. To illustrate this issue, we provide an intuitive example[2]. Consider a pretraining process involving datasets from multiple domains, such as social networks, molecular networks, academic networks, and knowledge graphs. Suppose the model learns triangle structures during pretraining. In social networks, the semantic meaning of these triangles is *stable*, following the principle of "the friend of my friend is my friend". However, in molecular graphs, the meaning of triangle patterns may be *unstable* due to chemical properties. This pattern conflict can significantly degrade the performance of graph models (Cao et al., 2023; Mao et al., 2024). Specialization helps resolve this issue by aligning the meanings of certain structural patterns with the semantics specific to the target domain.

### B.2. More Analysis on Domain Regularizer

**The Necessity of Domain Alignment.** Datasets from multiple domains are often projected into different subspaces, potentially due to misalignment of node attributes (Chen et al., 2024b) and the frequent patterns across domains. As a result, the model may "memorize" information specific to each domain rather than learning transferable patterns. This can lead to misunderstandings when the same pattern appeared across different graphs is projected into different subspaces. Consequently, the model struggles to acquire transferable knowledge that would benefit unseen tasks and specialized domains. Properly aligning the embedding spaces of different domains is crucial for obtaining transferable knowledge and improving performance on unseen graphs and specialized domains.

---

[2]This example was first illustrated in Cao et al. (2023)

**How to Regulate Domain Distances?** We propose a domain regularizer to control domain distances by projecting cross-domain graphs with different characteristics into a shared embedding space. Specifically, we define a shared subspace across domains and pull the subspaces of other domains into alignment with this defined space. The shared subspace should be positioned at the center of the cross-domain datasets to minimize the effort required to adjust the subspaces of all domains. In particular, the basis vector of the shared subspace is defined as the average of all instances:

$$\boldsymbol{h}_{Basis} = \mathbb{E}_{D\sim P(\mathcal{D})}\mathbb{E}_{\mathcal{T}_i\sim P(\mathcal{T}_D)}\phi(T_i), \tag{9}$$

where $P(\mathcal{D})$ represents the domain distribution, $\mathcal{T}_D$ is a distribution of task-trees within domain $D$, and $\phi(T_i)$ is the embedding of the task-tree $T_i$. Given the shared subspace basis, we optimize the KL divergence between each instance and the basis. However, obtaining the global basis vector $\boldsymbol{h}_{Basis}$ directly is impractical due to dataset size, so we approximate it by averaging the embeddings of all instances within a batch to compute the local basis $\hat{\boldsymbol{h}}_{Basis}$. We then optimize the KL divergence for all instances in the batch. To mitigate randomness, we empirically use a relatively large batch size (4,096). Formally, the domain regularizer is defined as

$$\mathcal{L}_{align} = \lambda \cdot \frac{1}{|\mathcal{B}|}\sum_{i\in\mathcal{B}}\mathrm{KL}(H\|Z_i) = -\lambda \cdot \frac{1}{|\mathcal{B}|}\sum_{i\in\mathcal{B}}\sum_{j}H(j)\log\Big(\frac{H(j)}{Z_i(j)}\Big), \tag{10}$$

where $\mathcal{B}$ denotes the batch, and $H$ and $Z_i$ represent the distributions of the local basis vector $\hat{\boldsymbol{h}}_{Basis}$ and instance embedding $\boldsymbol{z}_i$, respectively.

**How the Domain Regularizer Works?** To better understand how the domain regularizer functions, we conduct an analysis to demonstrate its benefits in regulating domain distances while preserving task structures for each dataset. We use eight datasets provided by Liu et al. (2024a) for pretraining: `Cora`, `Pubmed`, `Arxiv`, `WikiCS`, `WN18RR`, `FB15K237`, `CHEMHIV`, and `CHEMPCBA`. The analysis results are presented in Figure 5.

In the figure, we display (a) a heatmap of similarity between different datasets with varying weights, and visualizations of the embedding space before (b) and after (c) applying the domain regularizer. The results show that the domain regularizer effectively adjusts the distances between datasets by pushing apart overly similar graphs and bringing closer those that are too distinct. Furthermore, we show that the regularizer does not significantly alter the task structures of each dataset, as illustrated in subfigure (d). In this subfigure, we apply $k$-means algorithm on each dataset, setting $k$ to the number of classes, and compare to the ground-truth by using NMI as the metric. The assumption is that if two sets of vectors yield similar clustering results, the classification outcomes of the same classifier will be similar, indicating that the task structure across the two sets is consistent. The results demonstrate that changing the regularizer weight does not significantly affect task structures. This may be because the regularizer acts by translating vectors toward a central point without altering the relationships between individual pairs of vectors. To further evaluate the impact of domain regularizer for the downstream tasks, we present the model performance average over the used eight datasets across all settings in Table 7. We observe the use of domain regularizer can boost the model performance, especially in in-context and zero-shot settings. In addition, we empirically find that $\lambda = 10$ can lead to better performance. Thus, we set $\lambda = 10$ as the default weight in this paper.

### B.3. Discussion on Homophily and Heterophily

In node-level tasks, it is important to consider both graph homophily and heterophily (Ma et al., 2022). Homophily describes the close relationships between connected entities, while heterophily refers to distant relationships between connected entities. Empirically, basic message-passing GNNs tend to perform well on homophily graphs but struggle with heterophily graphs (Luan et al., 2022). Despite using GraphSAGE as the backbone in our GIT, it still performs well on heterophily graphs, such as `Children` and `Ratings`[3]. The experimental results for node classification and link prediction on these two graphs are presented in Table 17 and Table 18, where GIT generally achieves the best performance. We hypothesize that the proposed task-tree structure captures not only homophily relationships but also heterophily relationships. A potential question is whether our message-passing GNN can effectively capture these heterophily relationships, despite Ma et al. (2022) suggesting that basic GNNs may handle heterophily graphs by memorizing the patterns between the target node and its neighbors. We plan to use more advanced GNNs capable of encoding heterophily structures to further validate our hypothesis.

---

[3]Our collected graphs include two heterophily graphs, `Children` and `Ratings`, with homophily ratios of 0.42 and 0.38, respectively, whereas other graphs generally have a homophily ratio greater than 0.60 (Table 12, (Chen et al., 2024b)).

### B.4. Discussion on Model Expressiveness

**Node-level Task.** The task-tree structure is an approximation of the original graph structure, but converting graphs into tree structures inevitably results in some loss of information. To better preserve the structural details of the original graph, one could use more expressive or advanced GNNs, thereby expanding the potential tree vocabulary (Mao et al., 2024).

**Edge-level Task.** Existing message-passing GNNs struggle with the edge isomorphism problem (Srinivasan & Ribeiro, 2020). For instance, in Figure 6, the links $(v_1, v_2)$ and $(v_3, v_4)$ are isomorphic, while $(v_1, v_2)$ and $(v_1, v_3)$ are not. However, when using a mean aggregator to learn edge embeddings, the embeddings of $(v_1, v_2)$ and $(v_1, v_3)$ become indistinguishable. We consider that GIT may still encounter this issue, as the task-tree encoding currently averages the embeddings of task-relevant nodes. Addressing this challenge could involve techniques like Zhang et al. (2021), which ensure that isomorphic edges have distinct embeddings without impairing the model's basic inductive learning capabilities.

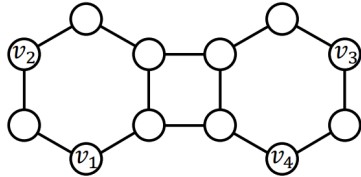

Figure 6: Edge Isomorphic (Figure 1, (Zhang et al., 2021)).

**Graph-level Task.** Message-passing GNNs are limited by the 1-WL test (Xu et al., 2019), which can restrict their performance on graph-level tasks. As we apply GraphSAGE as the backbone, our GIT also encounters this limitation. Zhang et al. (2024a) analyze the ability of different GNNs to detect graph substructures and conclude that more expressive GNNs, beyond the 1-WL test, can learn graph embeddings with richer information. Therefore, to improve model expressiveness, one can employ more expressive GNNs in our GIT. Additionally, techniques like Zhang et al. (2021) can be used to further enhance the model's discriminative capabilities.

### B.5. Scaling Law

**Model Size.** We evaluated the model's performance with different hidden dimensions, with results by domain presented in Figure 7. The results cover both basic fine-tuning and in-context learning, and comprehensive details are provided in Appendix K. We observe that increasing the number of hidden dimensions from 128 to 2,048 significantly improves model performance across all domains. We hypothesize that this improvement is due to the additional parameters, which enhance the model's ability to memorize shared patterns across graphs. The observation indicates the potential existence of scaling laws when using task-trees as the basic learning instances.

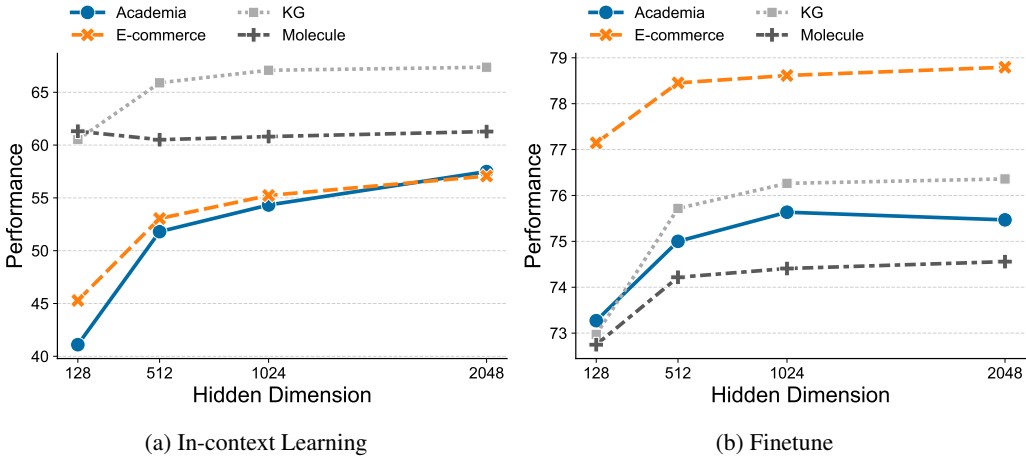

(a) In-context Learning       (b) Finetune

Figure 7: The impact of model sizes on performance.

**Data Size.** We attempted to evaluate the scaling law by increasing the pretraining data, but unfortunately, we did not

observe a clear trend where more data led to better performance. We consider there are three potential reasons. (1) From a model perspective, we use a GraphSAGE encoder with limited layers and parameters, which may not fully capture the knowledge contained in the pretraining data. Additionally, we apply basic mean pooling to derive task-tree embeddings from task-relevant node embeddings, which may prevent the model from identifying the relative importance of task-relevant nodes, thereby limiting its expressiveness. (2) From a training paradigm perspective, we employ a negative-free contrastive learning framework similar to Thakoor et al. (2022), but this basic approach may not be expressive enough to extract meaningful knowledge from the pretraining graphs. (3) From a data perspective, despite using over 30 graphs in this study, the number of instances is still significantly lower than that of textual or visual instances extracted from the Internet. Furthermore, the pretraining datasets may not be well-aligned. Although we used a textual encoder to align node features, we cannot guarantee that the encoded node features are in the same embedding space (Chen et al., 2024b).

## C. Potential Model Extensions

### C.1. Pretraining

**How to Design Reconstruction Tasks?** Theorem 3.5 suggests that a well-designed encoder, capable of effectively handling reconstruction tasks during pretraining, can improve the model's generalization ability. One approach is to use more powerful encoders to enhance reconstruction performance. Another approach is to introduce additional reconstruction losses to further refine the encoder. For example, methods such as those proposed by Qiu et al. (2020), and Hou et al. (2022), Ju et al. (2023b), and Wen et al. (2024), or designing more comprehensive reconstruction objectives could be explored.

**How to Improve Transferability?** The pretraining task, i.e., task-tree reconstruction, differs from the downstream task of task-tree classification, as the task heterogeneity may hinder model transferability (Hu et al., 2020). To mitigate this, one could develop more effective adaptation methods, such as graph prompt learning (Sun et al., 2022), to reduce task heterogeneity.

### C.2. Specialization via Instruction Tuning

**How to Define Instructions?** In this paper, as we focus on experiments with text-attributed graphs, we define instructions as label descriptions encoded by LLMs. However, this approach is not applicable to non-textual graphs. Other methods could be explored to define instructions, such as using proxy models (Hu et al., 2019) or graph heuristics (Jin et al., 2020) to generate instructions.

**How to Choose SFT Data?** We manually select graphs as supervised fine-tuning datasets for each domain, though the selected graphs may not be fully representative. Unlike textual data, evaluating the quality of graph datasets poses a challenge. Improved dataset selection methods could enhance the SFT process by identifying more representative or diverse data from graph databases. Additionally, while we perform instruction tuning over entire graphs, it is possible that only specific subgraphs are beneficial (Hashemi et al., 2024). Developing data selection methods that focus on high-quality subgraphs within a single SFT dataset could improve task-tree selection. Another worthy research line is to select SFT data that aligns with user preferences (Song et al., 2024).

**How to Leverage SFT Data?** In scenarios with limited instructions, standard supervised fine-tuning may struggle to capture sufficient knowledge of the target domain. To address this, methods could be proposed to better utilize the unlabeled instances in the SFT dataset, thus enhancing model adaptation (Sohn et al., 2020; Ni et al., 2025).

**How to Maintain General Inference Capability?** While instruction tuning specializes the model for a specific domain, it may compromise the model's general inference capabilities across other domains. This could hinder the model's performance when it needs to function both as a domain expert and a general reasoner. To mitigate this, regularization techniques could be designed to preserve the general knowledge encoded in the model during the instruction tuning process.

**Why SFT Works on Graphs?** Instruction tuning is a common post-training process in modern large language models (e.g., LLAMA, GPT) that significantly improves instruction-following capabilities. The success of this method in LLMs may stem from the fact that natural language serves as an interface between humans and models (Wei et al., 2021). However, the reason instruction tuning works for graphs remains an open question and presents a potential direction for future research.

## C.3. More Scenarios.

The paper leverages text-attributed graphs to align node features. However, the pre-processing of TAGs can be time-consuming, raising the challenge of how to effectively apply the model to graphs without aligned node features. Furthermore, while we primarily focus on homogeneous graphs in this work, most real-world applications involve heterogeneous graphs. Addressing the question of how to design a single model capable of handling various types of graphs remains an open challenge. Finally, applying the model to specific applications Zhang et al. (2024c), which may exhibit unique characteristics, is another important consideration for future research.

# D. Proof

## D.1. Proof of Theorem 3.1

*Proof.* We begin by introducing the basic GNN architecture used in the proof. Given a GNN encoder $\phi(\cdot)$ with parameters $\boldsymbol{W} = (\boldsymbol{W}_1, \boldsymbol{W}_2)$, we use a GraphSAGE-like architecture, defined as follows (with some notation abuse):

$$\boldsymbol{z}_i = \phi(T_i^L) = \sigma\Big(\boldsymbol{W}_1 \boldsymbol{x}_i + \boldsymbol{W}_2 \frac{1}{|\mathcal{N}_i|} \sum_{k \in \mathcal{N}_i} \phi(T_k^{L-1})\Big),$$

where $\sigma$ is the non-linear activation function, $\boldsymbol{x}_i$ is the node feature of node $i$, and $\mathcal{N}_i$ represents the neighbors of node $i$, corresponding to its children in the computation tree. $T_i^L$ denotes the computation tree of node $i$ with $L$ layers. Neighborhood information is incorporated by averaging the embeddings of neighboring nodes. WLOG, the averaging operation can be replaced with any permutation-invariant set operation without affecting the analysis in this paper. For simplicity, we assume all GNN layers share the same parameters; this assumption does not affect the validity of our proofs. Since these functions and neural networks exhibit Lipschitz continuity, we denote the Lipschitz constant of $\sigma(\cdot)$ as $\mathcal{C}_\sigma$. Additionally, we assume the norm of node features is bounded by $\|\boldsymbol{x}\| \leq \mathcal{B}_{\boldsymbol{x}}$, and the model weights by $\|\boldsymbol{W}_1\| \leq \mathcal{B}_{\boldsymbol{W}_1}$ and $\|\boldsymbol{W}_2\| \leq \mathcal{B}_{\boldsymbol{W}_2}$. While real-world graphs typically exhibit varied node features, standard techniques (as employed in this paper) like normalization can ensure that $\mathcal{B}_{\boldsymbol{x}}$ remains a small value. We define the distance between task-trees $T_1^t$ with $n$ task-relevant nodes $\{v_1, ..., v_n\}$ and $T_2^t$ with $m$ task-relevant nodes $\{v_1, ..., v_m\}$ as:

$$\Delta := \Big\|\phi(T_1^t) - \phi(T_2^t)\Big\| = \Big\|\frac{1}{n}\sum_{i=1}^{n}\phi(T_i) - \frac{1}{m}\sum_{j=1}^{m}\phi(T_j)\Big\|,$$

where $\|\cdot\|$ is the L2 distance. Following, we expand the stability term $\Delta$ as:

$$
\begin{aligned}
\Delta &= \Big\|\frac{1}{n}\sum_{i=1}^{n}\phi(T_i) - \frac{1}{m}\sum_{j=1}^{m}\phi(T_j)\Big\| \\
&= \Big\|\frac{1}{n}\sum_{i=1}^{n}\sigma\Big(\boldsymbol{W}_1\boldsymbol{x}_i + \boldsymbol{W}_2\frac{1}{|\mathcal{N}_i|}\sum_{k \in \mathcal{N}_i}\phi(T_k^{L-1})\Big) - \frac{1}{m}\sum_{j=1}^{m}\sigma\Big(\boldsymbol{W}_1\boldsymbol{x}_j + \boldsymbol{W}_2\frac{1}{|\mathcal{N}_j|}\sum_{k \in \mathcal{N}_j}\phi(T_k^{L-1})\Big)\Big\| \\
&\leq \mathcal{C}_\sigma\Big\|\frac{1}{n}\sum_{i=1}^{n}\Big(\boldsymbol{W}_1\boldsymbol{x}_i + \boldsymbol{W}_2\frac{1}{|\mathcal{N}_i|}\sum_{k \in \mathcal{N}_i}\phi(T_k^{L-1})\Big) - \frac{1}{m}\sum_{j=1}^{m}\Big(\boldsymbol{W}_1\boldsymbol{x}_j + \boldsymbol{W}_2\frac{1}{|\mathcal{N}_j|}\sum_{k \in \mathcal{N}_j}\phi(T_k^{L-1})\Big)\Big\| \\
&\leq \mathcal{C}_\sigma\underbrace{\Big\|\frac{1}{n}\sum_{i=1}^{n}\boldsymbol{W}_1\boldsymbol{x}_i - \frac{1}{m}\sum_{j=1}^{m}\boldsymbol{W}_1\boldsymbol{x}_j\Big\|}_{(a)} \\
&\quad + \mathcal{C}_\sigma\underbrace{\Big\|\frac{1}{n}\sum_{i=1}^{n}\boldsymbol{W}_2\frac{1}{|\mathcal{N}_i|}\sum_{k \in \mathcal{N}_i}\phi(T_k^{L-1}) - \frac{1}{m}\sum_{j=1}^{m}\boldsymbol{W}_2\frac{1}{|\mathcal{N}_j|}\sum_{k \in \mathcal{N}_j}\phi(T_k^{L-1})\Big\|}_{(b)}.
\end{aligned}
$$

Then, we separately analyze the term (a) and term (b). The term (a) can be bounded as follows:

$$\text{Term (a)} = \mathcal{C}_\sigma\Big\|\frac{1}{n}\sum_{i=1}^{n}\boldsymbol{W}_1\boldsymbol{x}_i - \frac{1}{m}\sum_{j=1}^{m}\boldsymbol{W}_1\boldsymbol{x}_j\Big\| \leq \mathcal{C}_\sigma\mathcal{B}_{\boldsymbol{W}_1}\Big\|\frac{1}{n}\sum_{i=1}^{n}\boldsymbol{x}_i - \frac{1}{m}\sum_{j=1}^{m}\boldsymbol{x}_j\Big\|.$$

That is, term (a) is bounded by the distance between the average features of nodes in the first layer of the task-trees (i.e., the nodes directly connected to the root). Next, we bound term (b):

$$
\begin{aligned}
\text{Term (b)} = \mathcal{C}_\sigma &\left\| \frac{1}{n} \sum_{i=1}^{n} \boldsymbol{W}_2 \frac{1}{|\mathcal{N}_i|} \sum_{k1 \in \mathcal{N}_i} \phi(T_{k1}^{L-1}) - \frac{1}{m} \sum_{j=1}^{m} \boldsymbol{W}_2 \frac{1}{|\mathcal{N}_j|} \sum_{k2 \in \mathcal{N}_j} \phi(T_{k2}^{L-1}) \right\| \\
\leq \mathcal{C}_\sigma \mathcal{B}_{\boldsymbol{W}_2} &\left\| \frac{1}{n} \sum_{i=1}^{n} \frac{1}{|\mathcal{N}_i|} \sum_{k1 \in \mathcal{N}_i} \phi(T_{k1}^{L-1}) - \frac{1}{m} \sum_{j=1}^{m} \frac{1}{|\mathcal{N}_j|} \sum_{k2 \in \mathcal{N}_j} \phi(T_{k2}^{L-1}) \right\| \\
= \mathcal{C}_\sigma \mathcal{B}_{\boldsymbol{W}_2} &\left\| \frac{1}{n} \sum_{i=1}^{n} \frac{1}{|\mathcal{N}_i|} \sum_{k1 \in \mathcal{N}_i} \sigma\left( \boldsymbol{W}_1 \boldsymbol{x}_{k1} + \boldsymbol{W}_2 \frac{1}{|\mathcal{N}_{k1}|} \sum_{s1 \in \mathcal{N}_{k1}} \phi(T_{s1}^{L-2}) \right) \right. \\
&\left. - \frac{1}{m} \sum_{j=1}^{m} \frac{1}{|\mathcal{N}_j|} \sum_{k2 \in \mathcal{N}_j} \sigma\left( \boldsymbol{W}_1 \boldsymbol{x}_{k2} + \boldsymbol{W}_2 \frac{1}{|\mathcal{N}_{k2}|} \sum_{s2 \in \mathcal{N}_{k2}} \phi(T_{s2}^{L-2}) \right) \right\| \\
\leq \mathcal{C}_\sigma \mathcal{B}_{\boldsymbol{W}_2} \mathcal{C}_\sigma \mathcal{B}_{\boldsymbol{W}_1} &\underbrace{\left\| \frac{1}{n} \sum_{i=1}^{n} \frac{1}{|\mathcal{N}_i|} \sum_{k1 \in \mathcal{N}_i} \boldsymbol{x}_{k1} - \frac{1}{m} \sum_{j=1}^{m} \frac{1}{|\mathcal{N}_j|} \sum_{k2 \in \mathcal{N}_j} \boldsymbol{x}_{k2} \right\|}_{(c)} \\
+ \mathcal{C}_\sigma \mathcal{B}_{\boldsymbol{W}_2} \mathcal{C}_\sigma \mathcal{B}_{\boldsymbol{W}_2} &\underbrace{\left\| \frac{1}{n} \sum_{i=1}^{n} \frac{1}{|\mathcal{N}_i|} \sum_{k1 \in \mathcal{N}_i} \frac{1}{|\mathcal{N}_{k1}|} \sum_{s1 \in \mathcal{N}_{k1}} \phi(T_{s1}^{L-2}) - \frac{1}{m} \sum_{j=1}^{m} \frac{1}{|\mathcal{N}_j|} \sum_{k2 \in \mathcal{N}_j} \frac{1}{|\mathcal{N}_{k2}|} \sum_{s2 \in \mathcal{N}_{k2}} \phi(T_{s2}^{L-2}) \right\|}_{(d)}.
\end{aligned}
$$

Term (c) describes the distance between the average features of nodes in the second layer of the task-trees, while term (d) follows a recursive formula, similar to term (b). By combining terms (a) and (b), we have:

$$
\begin{aligned}
\Delta \leq \mathcal{C}_\sigma \mathcal{B}_{\boldsymbol{W}_1} &\left\| \frac{1}{n} \sum_{i=1}^{n} \boldsymbol{x}_i - \frac{1}{m} \sum_{j=1}^{m} \boldsymbol{x}_j \right\| \\
+ \mathcal{C}_\sigma \mathcal{B}_{\boldsymbol{W}_2} \mathcal{C}_\sigma \mathcal{B}_{\boldsymbol{W}_1} &\left\| \frac{1}{n} \sum_{i=1}^{n} \frac{1}{|\mathcal{N}_i|} \sum_{k1 \in \mathcal{N}_i} \boldsymbol{x}_{k1} - \frac{1}{m} \sum_{j=1}^{m} \frac{1}{|\mathcal{N}_j|} \sum_{k2 \in \mathcal{N}_j} \boldsymbol{x}_{k2} \right\| \\
+ \mathcal{C}_\sigma \mathcal{B}_{\boldsymbol{W}_2} \mathcal{C}_\sigma \mathcal{B}_{\boldsymbol{W}_2} &\left\| \frac{1}{n} \sum_{i=1}^{n} \frac{1}{|\mathcal{N}_i|} \sum_{k1 \in \mathcal{N}_i} \frac{1}{|\mathcal{N}_{k1}|} \sum_{s1 \in \mathcal{N}_{k1}} \phi(T_{s1}^{L-2}) - \frac{1}{m} \sum_{j=1}^{m} \frac{1}{|\mathcal{N}_j|} \sum_{k2 \in \mathcal{N}_j} \frac{1}{|\mathcal{N}_{k2}|} \sum_{s2 \in \mathcal{N}_{k2}} \phi(T_{s2}^{L-2}) \right\|.
\end{aligned}
$$

We can extend the formula recursively through all layers until the final layer. The recursive nature of the formula allows us to more easily reformulate the bound:

$$
\Delta \leq \mathcal{C}_1 \Delta_1 + \mathcal{C}_2 \Delta_2 + ... + \mathcal{C}_{L-1} \Delta_{L-1}, \tag{11}
$$

where $\Delta_l$ denotes the distance between task-trees at the $l$-th layer. For clarity, we can interpret $\mathcal{C}_1 \Delta_1$ as corresponding to Term (a) and $\mathcal{C}_2 \Delta_2$ as corresponding to Term (c). Next, we explain how to determine $\mathcal{C}_l$ and $\Delta_l$ for each layer.

By analyzing the recursive formula, we determine $\mathcal{C}_l$ as follows:

$$
\begin{cases}
\mathcal{C}_1 &= \mathcal{C}_\sigma \mathcal{B}_{\boldsymbol{W}_1}, \\
\mathcal{C}_2 &= \mathcal{C}_\sigma \mathcal{B}_{\boldsymbol{W}_2} \times \mathcal{C}_\sigma \mathcal{B}_{\boldsymbol{W}_1}, \\
... \\
\mathcal{C}_l &= (\mathcal{C}_\sigma \mathcal{B}_{\boldsymbol{W}_2})^l \times \mathcal{C}_\sigma \mathcal{B}_{\boldsymbol{W}_1}.
\end{cases}
$$

We then define the $\Delta_l$. For a concise definition, we introduce an additional notation for describing the subtree information:

$$\begin{cases} \boldsymbol{x}_i^{(0)} &= \boldsymbol{x}_i, \\ \boldsymbol{x}_i^{(1)} &= \frac{1}{|\mathcal{N}_i|} \sum_{j \in \mathcal{N}_i} \boldsymbol{x}_j^{(0)}, \\ \boldsymbol{x}_i^{(2)} &= \frac{1}{|\mathcal{N}_i|} \sum_{j \in \mathcal{N}_i} \boldsymbol{x}_j^{(1)}, \\ \dots \\ \boldsymbol{x}_i^{(l)} &= \frac{1}{|\mathcal{N}_i|} \sum_{j \in \mathcal{N}_i} \boldsymbol{x}_j^{(l-1)}. \end{cases}$$

By using the term, we can define the $\Delta_l$ as:

$$\begin{cases} \Delta_1 &= \left\| \frac{1}{n} \sum_{i=1}^n \boldsymbol{x}_i^{(0)} - \frac{1}{m} \sum_{j=1}^m \boldsymbol{x}_j^{(0)} \right\|, \\ \Delta_2 &= \left\| \frac{1}{n} \sum_{i=1}^n \boldsymbol{x}_i^{(1)} - \frac{1}{m} \sum_{j=1}^m \boldsymbol{x}_j^{(1)} \right\|, \\ \dots \\ \Delta_l &= \left\| \frac{1}{n} \sum_{i=1}^n \boldsymbol{x}_i^{(l-1)} - \frac{1}{m} \sum_{j=1}^m \boldsymbol{x}_j^{(l-1)} \right\|. \end{cases}$$

By using a formulation like expression 11, we can decompose the impact of different layers, facilitating further analysis of the upper bound on the distance between two task-trees. Next, we will analyze the upper bound of each term. To begin, we first introduce a lemma.

**Lemma D.1.** *Given two sets of random vectors $\mathcal{S}_1 = \{\boldsymbol{v}_1, ..., \boldsymbol{v}_n\}$ and $\mathcal{S}_2 = \{\boldsymbol{v}_1, ..., \boldsymbol{v}_m\}$, the following holds:*

$$\left\| \frac{1}{n} \sum_{i=1}^n \boldsymbol{v}_i - \frac{1}{m} \sum_{j=1}^m \boldsymbol{v}_j \right\| \le \frac{1}{nm} \sum_{i=1}^n \sum_{j=1}^m \left\| \boldsymbol{v}_i - \boldsymbol{v}_j \right\|.$$

*Proof.* Let's consider two sets $A = \{\boldsymbol{a}_1, \boldsymbol{a}_2\}$ and $B = \{\boldsymbol{b}_1, \boldsymbol{b}_2\}$, and $\overline{\boldsymbol{a}} = (\boldsymbol{a}_1 + \boldsymbol{a}_2)/2$, $\overline{\boldsymbol{b}} = (\boldsymbol{b}_1 + \boldsymbol{b}_2)/2$. We have:

$$\begin{aligned} \|\overline{\boldsymbol{a}} - \overline{\boldsymbol{b}}\| &= \|(\boldsymbol{a}_1 + \boldsymbol{a}_2)/2 - (\boldsymbol{b}_1 + \boldsymbol{b}_2)/2\| \\ &= \|2\boldsymbol{a}_1 + 2\boldsymbol{a}_2 - 2\boldsymbol{b}_1 + 2\boldsymbol{b}_2\|/4 \\ &= \|(\boldsymbol{a}_1 - \boldsymbol{b}_1) + (\boldsymbol{a}_1 - \boldsymbol{b}_2) + (\boldsymbol{a}_2 - \boldsymbol{b}_1) + (\boldsymbol{a}_2 - \boldsymbol{b}_2)\|/4 \\ &\le (\|\boldsymbol{a}_1 - \boldsymbol{b}_1\| + \|\boldsymbol{a}_1 - \boldsymbol{b}_2\| + \|\boldsymbol{a}_2 - \boldsymbol{b}_1\| + \|\boldsymbol{a}_2 - \boldsymbol{b}_2\|)/4 \end{aligned}$$

WLOG, this analysis can be extended to cases where the size of $A$ is $n$ and the size of $B$ is $m$. $\square$

Based on the Lemma, we have

$$\Delta \le \mathcal{C}_1 \left\| \frac{1}{n} \sum_{i=1}^n \boldsymbol{x}_i^{(0)} - \frac{1}{m} \sum_{j=1}^m \boldsymbol{x}_j^{(0)} \right\| + ... + \mathcal{C}_1 \mathcal{C}_2^{L-1} \left\| \frac{1}{n} \sum_{i=1}^n \boldsymbol{x}_i^{(L-1)} - \frac{1}{m} \sum_{j=1}^m \boldsymbol{x}_j^{(L-1)} \right\|$$

$$\le \frac{1}{nm} \sum_{i=1}^n \sum_{j=1}^m \left( \mathcal{C}_1 \|\boldsymbol{x}_i^{(0)} - \boldsymbol{x}_j^{(0)}\| + ... + \mathcal{C}_1 \mathcal{C}_2^{L-1} \|\boldsymbol{x}_i^{(L-1)} - \boldsymbol{x}_j^{(L-1)}\| \right),$$

which is displayed in Theorem 3.1.

We then use the Lemma to bound the $\Delta_l$. For example, the upper bound of $\Delta_1$ is:

$$\begin{aligned} \Delta_1 &= \left\| \frac{1}{n} \sum_{i=1}^n \boldsymbol{x}_i^{(0)} - \frac{1}{m} \sum_{j=1}^m \boldsymbol{x}_j^{(0)} \right\| \\ &\le \frac{1}{nm} \sum_{i=1}^n \sum_{j=1}^m \|\boldsymbol{x}_i^{(0)} - \boldsymbol{x}_j^{(0)}\| \\ &\le \frac{1}{nm} \sum_{i=1}^n \sum_{j=1}^m (\|\boldsymbol{x}_i^{(0)}\| + \|\boldsymbol{x}_j^{(0)}\|) \le 2\mathcal{B}_{\boldsymbol{x}}. \end{aligned}$$

Similarly, the upper bound of $\Delta_2$ is:

$$\Delta_2 = \left\| \frac{1}{n}\sum_{i=1}^{n} \boldsymbol{x}_i^{(1)} - \frac{1}{m}\sum_{j=1}^{m} \boldsymbol{x}_j^{(1)} \right\|$$

$$\leq \frac{1}{nm}\sum_{i=1}^{n}\sum_{j=1}^{m} \left\| \boldsymbol{x}_i^{(1)} - \boldsymbol{x}_j^{(1)} \right\|$$

$$= \frac{1}{nm}\sum_{i=1}^{n}\sum_{j=1}^{m} \left\| \frac{1}{d_i}\sum_{k_i^{(1)}\in\mathcal{N}_i} \boldsymbol{x}_{k_i^{(1)}}^{(0)} - \frac{1}{d_j}\sum_{k_j^{(1)}\in\mathcal{N}_j} \boldsymbol{x}_{k_j^{(1)}}^{(0)} \right\|$$

$$\leq \frac{1}{nm}\sum_{i=1}^{n}\sum_{j=1}^{m} \frac{1}{d_i d_j} \sum_{k_i^{(1)}\in\mathcal{N}_i}\sum_{k_j^{(1)}\in\mathcal{N}_j} \| \boldsymbol{x}_{k_i^{(1)}}^{(0)} - \boldsymbol{x}_{k_j^{(1)}}^{(0)} \| \leq 2\mathcal{B}_{\boldsymbol{x}},$$

where $d_i = |\mathcal{N}_i|$ and $d_j = |\mathcal{N}_j|$ represent the number of children (i.e., the degree) of nodes $i$ and $j$, respectively. Thus, the upper bound of $\Delta_l$ is:

$$\Delta_l \leq \frac{1}{nm}\sum_{i=1}^{n}\sum_{j=1}^{m} \frac{1}{d_i d_j} \sum_{k_i^{(1)}\in\mathcal{N}_i}\sum_{k_j^{(1)}\in\mathcal{N}_j} \underbrace{\frac{1}{d_{k_i^{(1)}} d_{k_j^{(1)}}} \sum_{k_i^{(2)}\in\mathcal{N}_{k_i^{(1)}}}\sum_{k_j^{(2)}\in\mathcal{N}_{k_j^{(1)}}} ... \| \boldsymbol{x}_{k_i^{(l-1)}}^{(0)} - \boldsymbol{x}_{k_j^{(l-1)}}^{(0)} \|}_{(l-1)\times} \leq 2\mathcal{B}_{\boldsymbol{x}}.$$

Now we can have the highest upper bound of $\Delta$ as:

$$\Delta \leq (\mathcal{C}_1 + \mathcal{C}_2 + ... + \mathcal{C}_L) 2\mathcal{B}_{\boldsymbol{x}}$$

$$= (\mathcal{C}_\sigma \mathcal{B}_{\boldsymbol{W}_1} + ... + (\mathcal{C}_\sigma \mathcal{B}_{\boldsymbol{W}_2})^l \times \mathcal{C}_\sigma \mathcal{B}_{\boldsymbol{W}_1}) 2\mathcal{B}_{\boldsymbol{x}}$$

$$= 2\mathcal{B}_{\boldsymbol{x}} \cdot \mathcal{C}_\sigma \mathcal{B}_{\boldsymbol{W}_1} \frac{(\mathcal{C}_\sigma \mathcal{B}_{\boldsymbol{W}_2})^L - 1}{\mathcal{C}_\sigma \mathcal{B}_{\boldsymbol{W}_2} - 1}.$$

Note that this upper bound is an extreme case; the bound can be tightened by adding supplementary information or making additional assumptions. Additionally, the distance between task-trees can be reduced by applying techniques like normalization, which lowers the values of the constants.

$$\square$$

### D.2. Proof of Theorem 3.3

*Proof.* We begin by restating the notations used in the theorem. Let $\mathcal{P}$ represent the task-tree distribution, $\mathcal{T}$ the downstream task distribution, $\phi \in \Phi$ the GNN encoder, $g \in G$ the predictor head used during pretraining, and $f \in \mathcal{F}$ the predictor head for the downstream task. The pretraining objective is defined as $\mathcal{L}_\mathcal{P}(g \circ \phi) := \mathbb{E}_{(\hat{T},T)\sim\mathcal{P}}\|g(\phi(\hat{T})) - \phi(T)\|^2$, where $T$ is the task-tree and $\hat{T}$ is the corresponding corrupted task-tree obtained via arbitrary corruption functions. The risk on the downstream task is defined as $\mathcal{R}_\mathcal{T}(f \circ \phi) := \mathbb{E}_{(T,y)\sim\mathcal{T}}\kappa(f(\phi(T)), y)$, where $T$ is the task-tree, $y$ is the associated label, and $\kappa$ denotes the loss function. Before we begin the proof, we present an additional helper proposition.

**Proposition D.2** (Ruhe's Trace Inequality (Ruhe, 1970))**.** *If $\boldsymbol{X}$ and $\boldsymbol{Y}$ are positive semi-definite Hermitian matrices with eigenvalues, $x_1 \geq x_2 \geq ... \geq x_n \geq 0$ and $y_1 \geq y_2 \geq ... \geq y_n \geq 0$, then*

$$\sum_{i=1}^{n} x_i y_{n-i+1} \leq \text{tr}(\boldsymbol{X}\boldsymbol{Y}) \leq \sum_{i=1}^{n} x_i y_i.$$

Note that we are going to prove

$$\min_{f\in\mathcal{F}} \mathcal{R}_\mathcal{T}(f \circ \phi) - \min_{f'\in\mathcal{F}} \mathcal{R}_\mathcal{T}(f' \circ \phi') \leq \mathcal{C}_\delta \left( \min_{g\in G} \mathcal{L}_\mathcal{P}(g \circ \phi) - \min_{g'\in G} \mathcal{L}_\mathcal{P}(g' \circ \phi') \right)^\delta,$$

where $\mathcal{C}_\delta \approx O(1)$ and $\delta = \frac{1}{2}$. The proof involves deriving the upper bound for the term $\min_{f \in \mathcal{F}} \mathcal{R}_\mathcal{T}(f \circ \phi) - \min_{f' \in \mathcal{F}} \mathcal{R}_\mathcal{T}(f' \circ \phi')$, followed by the lower bound for $\min_{g \in G} \mathcal{L}_\mathcal{P}(g \circ \phi) - \min_{g' \in G} \mathcal{L}_\mathcal{P}(g' \circ \phi')$. To simplify the proof, we assume the downstream task is a binary classification task, though this approach can be extended to multi-classification scenarios.

We analyze the upper bound of $\min_{f \in \mathcal{F}} \mathcal{R}_\mathcal{T}(f \circ \phi) - \min_{f' \in \mathcal{F}} \mathcal{R}_\mathcal{T}(f' \circ \phi')$ as follows:

$$\min_{f \in \mathcal{F}} \mathcal{R}_\mathcal{T}(f \circ \phi) - \min_{f' \in \mathcal{F}} \mathcal{R}_\mathcal{T}(f' \circ \phi') = \min_{f \in \mathcal{F}} \mathbb{E}_\mathcal{T} \kappa(f(\phi(T))) - \min_{f' \in \mathcal{F}} \mathbb{E}_\mathcal{T} \kappa(f'(\phi'(T))),$$

where $(T, y) \sim \mathcal{T}$ and $\kappa(f(\phi(T)))$ is shorthand for $\kappa(f(\phi(T)), y)$ for notational convenience. Since we define $f \in \mathcal{F}$ as a linear predictor for binary classification, we can rewrite this equation in the following form:

$$\min_{\|\boldsymbol{\theta}\| \leq \mathcal{B}_{\boldsymbol{\theta}}} \mathbb{E}_\mathcal{T} \kappa(\boldsymbol{\theta}^\top \phi(T)) - \min_{\|\boldsymbol{\theta}\| \leq \mathcal{B}_{\boldsymbol{\theta}}} \mathbb{E}_\mathcal{T} \kappa(\boldsymbol{\theta}'^\top \phi'(T))$$

$$\leq \min_{\|\boldsymbol{\theta}\| \leq \mathcal{B}_{\boldsymbol{\theta}}} \mathbb{E}_\mathcal{T} \left| \boldsymbol{\theta}^\top \phi(T) - \boldsymbol{\theta}'^\top \phi'(T) \right|$$

$$\leq \min_{\|\boldsymbol{\theta}\| \leq \mathcal{B}_{\boldsymbol{\theta}}} \sqrt{\mathbb{E}_\mathcal{T} \left( \boldsymbol{\theta}^\top \phi(T) - \boldsymbol{\theta}'^\top \phi'(T) \right)^2}$$

$$= \min_{\|\boldsymbol{\theta}\| \leq \mathcal{B}_{\boldsymbol{\theta}}} \sqrt{\mathbb{E}_\mathcal{T} \left( \boldsymbol{\theta}^\top \phi(T) \phi(T)^\top \boldsymbol{\theta} + \boldsymbol{\theta}'^\top \phi'(T) \phi'(T)^\top \boldsymbol{\theta}' - 2\boldsymbol{\theta}^\top \phi(T) \phi'(T)^\top \boldsymbol{\theta}' \right)}$$

$$= \min_{\|\boldsymbol{\theta}\| \leq \mathcal{B}_{\boldsymbol{\theta}}} \sqrt{\boldsymbol{\theta}^\top \mathbb{E}[\phi(T) \phi(T)^\top] \boldsymbol{\theta} + \boldsymbol{\theta}'^\top \mathbb{E}[\phi'(T) \phi'(T)^\top] \boldsymbol{\theta}' - 2\boldsymbol{\theta}^\top \mathbb{E}[\phi(T) \phi'(T)^\top] \boldsymbol{\theta}'}$$

$$\leq \sqrt{\boldsymbol{\theta}'^\top \Lambda \boldsymbol{\theta}'}, \Lambda = \mathbb{E}[\phi(T) \phi(T)^\top] - \mathbb{E}[\phi(T) \phi'(T)^\top] \left( \mathbb{E}[\phi'(T) \phi'(T)^\top] \right)^\dagger \mathbb{E}[\phi(T) \phi'(T)^\top]$$

Note that in the previous formula, we define $\boldsymbol{\theta} = (\mathbb{E}[\phi(T) \phi(T)^\top])^\dagger (\mathbb{E}[\phi(T) \phi'(T)^\top]) \boldsymbol{\theta}'$. Under the unconstrained setting, the minimum of $\boldsymbol{\theta}^\top \mathbb{E}[\phi(T) \phi(T)^\top] \boldsymbol{\theta} + \boldsymbol{\theta}'^\top \mathbb{E}[\phi'(T) \phi'(T)^\top] \boldsymbol{\theta}' - 2\boldsymbol{\theta}^\top \mathbb{E}[\phi(T) \phi'(T)^\top] \boldsymbol{\theta}'$ reduces to $\boldsymbol{\theta}'^\top \Lambda \boldsymbol{\theta}'$ (Deng et al., 2024). We can select a sufficiently large $\mathcal{B}_{\boldsymbol{\theta}}$ to ensure an adequately large function space. Additionally, we define $\boldsymbol{\theta}'$ as the optimal head for the encoder $\phi'$. The expression $\sqrt{\boldsymbol{\theta}'^\top \Lambda \boldsymbol{\theta}'}$ is equivalent to $\sqrt{\mathrm{tr}(\Lambda \boldsymbol{\theta}'^\top \boldsymbol{\theta}')}$, which can be further simplified using Proposition D.2.

$$\sqrt{\mathrm{tr}(\Lambda \boldsymbol{\theta}'^\top \boldsymbol{\theta}')} \leq \sqrt{\sum_{i=1}^d \sigma_i(\Lambda) \sigma_i(\boldsymbol{\theta}' \boldsymbol{\theta}'^\top)} \leq \sqrt{d \sigma_{\max}(\Lambda) \sigma_{\max}(\boldsymbol{\theta}' \boldsymbol{\theta}'^\top)},$$

where $\sigma_i$ is the $i$-th eigenvalue for the matrix and $\sigma_{\max}$ denotes the maximum eigenvalue.

Then, we are going to demonstrate the lower bound of $\min_{g \in G} \mathcal{L}_\mathcal{P}(g \circ \phi) - \min_{g' \in G} \mathcal{L}_\mathcal{P}(g' \circ \phi')$.

$$\min_{g \in G} \mathcal{L}_\mathcal{P}(g \circ \phi) - \min_{g' \in G} \mathcal{L}_\mathcal{P}(g' \circ \phi') = \min_{g \in G} \mathbb{E}_\mathcal{P} \|g(\phi(\hat{T})) - \phi(T)\|^2 - \min_{g' \in G} \mathbb{E}_\mathcal{P} \|g'(\phi'(\hat{T})) - \phi'(T)\|^2,$$

where $(\hat{T}, T) \sim \mathcal{P}$. Here we also consider the predictor $g$ as a linear function, so that we have the following form:

$$\min_{\|\boldsymbol{W}\| \in \mathcal{B}_{\boldsymbol{W}}} \mathbb{E}_\mathcal{P} \|\boldsymbol{W} \phi(\hat{T}) - \phi(T)\|^2 - \min_{\|\boldsymbol{W}'\| \in \mathcal{B}_{\boldsymbol{W}}} \mathbb{E}_\mathcal{P} \|\boldsymbol{W}' \phi'(\hat{T}) - \phi'(T)\|^2$$

$$= \min_{\|\boldsymbol{W}\| \in \mathcal{B}_{\boldsymbol{W}}} \mathbb{E}_\mathcal{P} \|\boldsymbol{W} \phi(\hat{T}) - \boldsymbol{W}' \phi'(\hat{T})\|^2 + \mathcal{C}_\mathcal{P}$$

$$\geq \sum_{r=1}^d \min_{\boldsymbol{w}_r} \mathbb{E}_\mathcal{P} \|\boldsymbol{w}_r^\top \phi(\hat{T}) - \boldsymbol{w}_r'^\top \phi'(\hat{T})\|^2$$

$$\geq \sum_{r=1}^d \min_{\boldsymbol{w}_r} \mathbb{E}_\mathcal{P} (\boldsymbol{w}_r^\top \phi(\hat{T}) \phi(\hat{T})^\top \boldsymbol{w}_r + \boldsymbol{w}_r'^\top \phi'(\hat{T}) \phi'(\hat{T})^\top \boldsymbol{w}_r' - 2\boldsymbol{w}_r^\top \phi(\hat{T}) \phi'(\hat{T})^\top \boldsymbol{w}_r')$$

$$\geq \sum_{r=1}^d \boldsymbol{w}_r'^\top \Lambda \boldsymbol{w}_r', \Lambda = \mathbb{E}[\phi(\hat{T}) \phi(\hat{T})^\top] - \mathbb{E}[\phi(\hat{T}) \phi'(\hat{T})^\top] \left( \mathbb{E}[\phi'(\hat{T}) \phi'(\hat{T})^\top] \right)^\dagger \mathbb{E}[\phi(\hat{T}) \phi'(\hat{T})^\top].$$

where $\mathcal{C}_{\mathcal{P}} = \mathbb{E}_{\mathcal{P}} \|\phi'(T) - \phi(T)\|^2$ is a constant, and $\boldsymbol{W}'$ is defined as the optimal transformation matrix for the encoder $\phi'$. Based on this formula, we can further simplify the bound on

$$\sum_{r=1}^{d} \boldsymbol{w}_r'^{\top} \Lambda \boldsymbol{w}_r' = \text{tr}(\Lambda \sum_{r=1}^{d} \boldsymbol{w}_r' \boldsymbol{w}_r'^{\top}) \geq \sigma_{\max}(\Lambda) \sigma_{\min}(\sum_{r=1}^{d} \boldsymbol{w}_r' \boldsymbol{w}_r'^{\top}).$$

Now that we have the upper bound for $\min_{f \in \mathcal{F}} \mathcal{R}_{\mathcal{T}}(f \circ \phi) - \min_{f' \in \mathcal{F}} \mathcal{R}_{\mathcal{T}}(f' \circ \phi')$ and the lower bound for $\min_{g \in G} \mathcal{L}_{\mathcal{P}}(g \circ \phi) - \min_{g' \in G} \mathcal{L}_{\mathcal{P}}(g' \circ \phi')$, we can establish the relationship between them as follows:

$$\min_{f \in \mathcal{F}} \mathcal{R}_{\mathcal{T}}(f \circ \phi) - \min_{f' \in \mathcal{F}} \mathcal{R}_{\mathcal{T}}(f' \circ \phi')$$
$$\leq O\left( \frac{\sqrt{d\sigma_{\max}(\boldsymbol{\theta}' \boldsymbol{\theta}'^{\top})}}{\sqrt{\sigma_{\min}(\sum_{r=1}^{d} \boldsymbol{w}_r' \boldsymbol{w}_r'^{\top})}} \right) (\min_{g \in G} \mathcal{L}_{\mathcal{P}}(g \circ \phi) - \min_{g' \in G} \mathcal{L}_{\mathcal{P}}(g' \circ \phi')).$$

Based on our definition, $\boldsymbol{\theta}'$ and $\boldsymbol{W}'$ are optimal heads for the encoder $\phi'$, the complexity term $O\left( \frac{\sqrt{d\sigma_{\max}(\boldsymbol{\theta}' \boldsymbol{\theta}'^{\top})}}{\sqrt{\sigma_{\min}(\sum_{r=1}^{d} \boldsymbol{w}_r' \boldsymbol{w}_r'^{\top})}} \right)$ would be a constant which in the order of $O(1)$.

$\square$

### D.3. Proof of Theorem 3.5

*Proof.* We begin by introducing some essential notations. Let $\mathcal{P}$ represent the pretraining task-tree distribution and $\mathcal{T}$ the downstream task-tree distribution. The pair $(T, y) \sim \mathcal{T}$ denotes a task-tree and its corresponding label, where we define the labeling function as $\psi$, meaning $y = \psi(T)$. The GNN encoder is $\phi \in \Phi$, the pretraining predictor is $g \in G$, and the downstream predictor head is $f \in \mathcal{F}$. As in the previous proof, we consider a binary classification task for simplicity, though this can be extended to multi-class settings. The downstream risk is given by $\mathcal{R}_{\mathcal{T}}(f \circ \phi) := \mathbb{E}_{(T, \psi(T)) \sim \mathcal{T}} \kappa(f(\phi(T)), \psi(T))$, where $\kappa$ is a loss function.

Then, we define the excess risk on the downstream distribution $\mathcal{T}$ as

$$\begin{aligned}
\mathcal{E}(f, \phi) = & \mathcal{R}_{\mathcal{T}}(f \circ \phi) - \min_{f' \in \mathcal{F}, \phi' \in \Phi} \mathcal{R}_{\mathcal{T}}(f' \circ \phi') \\
= & \left( \mathcal{R}_{\mathcal{T}}(f \circ \phi) - \min_{f' \in \mathcal{F}, \phi' \in \Phi} \mathcal{R}_{\mathcal{T}}(f' \circ \phi') \right) + \left( \min_{f' \in \mathcal{F}} \mathcal{R}_{\mathcal{T}}(f' \circ \phi) - \min_{f' \in \mathcal{F}} \mathcal{R}_{\mathcal{T}}(f' \circ \phi) \right) \\
& + \left( \min_{f' \in \mathcal{F}} \mathcal{R}_{\mathcal{P}}(f' \circ \phi) - \min_{f' \in \mathcal{F}} \mathcal{R}_{\mathcal{P}}(f' \circ \phi) \right) + \left( \min_{f' \in \mathcal{F}} \mathcal{R}_{\mathcal{P}}(f' \circ \phi^*) - \min_{f' \in \mathcal{F}} \mathcal{R}_{\mathcal{P}}(f' \circ \phi^*) \right) \\
= & \underbrace{\mathcal{R}_{\mathcal{T}}(f \circ \phi) - \min_{f' \in \mathcal{F}} \mathcal{R}_{\mathcal{T}}(f' \circ \phi)}_{(a)} + \underbrace{\min_{f' \in \mathcal{F}} \mathcal{R}_{\mathcal{T}}(f' \circ \phi) - \min_{f' \in \mathcal{F}} \mathcal{R}_{\mathcal{P}}(f' \circ \phi)}_{(b)} \\
& + \underbrace{\min_{f' \in \mathcal{F}} \mathcal{R}_{\mathcal{P}}(f' \circ \phi) - \min_{f' \in \mathcal{F}} \mathcal{R}_{\mathcal{P}}(f' \circ \phi^*)}_{(c)} + \underbrace{\min_{f' \in \mathcal{F}} \mathcal{R}_{\mathcal{P}}(f' \circ \phi^*) - \min_{f' \in \mathcal{F}, \phi' \in \Phi} \mathcal{R}_{\mathcal{T}}(f' \circ \phi')}_{(d)},
\end{aligned}$$

where $\phi$ and $f$ represent encoder obtained during pretraining and the prediction head learned in downstream task, respectively, while $\phi'$ and $f'$ are the optimal encoder and predictor head on the downstream distribution. $\phi^*$ is the optimal encoder obtained during pretraining, defined as $\phi^* = \arg\min_{\phi \in \Phi} \min_{g \in G} \mathcal{L}_{\mathcal{P}}(g \circ \phi)$. We will analyze these four terms separately.

To bound the term (a), we need to introduce the empirical Rademacher complexity (Definition 1, (Deng et al., 2024)) as

$$\hat{\mathfrak{R}}_{\mathcal{T}} := \mathbb{E}_{\varepsilon \in \{\pm 1\}^n} \left[ \sup_{f \in \mathcal{F}} \frac{1}{n} \sum_{i=1}^{n} \varepsilon_i \kappa(f \circ \phi(T), \psi(T)) \right],$$

where $\varepsilon_i$ is i.i.d., and $\mathbb{P}(\varepsilon = 1) = \mathbb{P}(\varepsilon = -1) = \frac{1}{2}$.

Using this definition, we can bound the term (a):

$$\text{Term (a)} = \mathcal{R}_{\mathcal{T}}(f \circ \phi) - \min_{f' \in \mathcal{F}} \mathcal{R}_{\mathcal{T}}(f' \circ \phi)$$

$$= \underbrace{\mathcal{R}_{\mathcal{T}}(f \circ \phi) - \hat{\mathcal{R}}_{\mathcal{T}}(f \circ \phi)}_{(a.1)} + \underbrace{\hat{\mathcal{R}}_{\mathcal{T}}(f^* \circ \phi) - \min_{f' \in \mathcal{F}} \mathcal{R}_{\mathcal{T}}(f' \circ \phi)}_{(a.2)} + \underbrace{\hat{\mathcal{R}}_{\mathcal{T}}(f \circ \phi) - \hat{\mathcal{R}}_{\mathcal{T}}(f^* \circ \phi)}_{(a.3)},$$

where $f^*$ is the optimal predictor head over the distribution $\mathcal{T}$, defined as $f^* = \arg\min_{f \in \mathcal{F}} \mathcal{R}_{\mathcal{T}}(f \circ \phi)$. The term (a.3) represents the empirical risk gap between the learned head $f$ and the best head $f^*$, which implies that the term is a constant greater than or equal to 0. Term (a.1) and (a.2) describe the gap between the risk and the empirical risk. According to uniform convergence, these two terms can be expressed in terms of empirical Rademacher complexity. Thus, term (a) can be bounded as:

$$\text{Term (a)} \leq 4\hat{\mathfrak{R}}_{\mathcal{T}} + 4\mathcal{B}_\kappa \sqrt{\frac{\log(1/v)}{n}},$$

where $\mathcal{B}_\kappa$ is the bound of the Lipschitz of loss function $\kappa$. We then further simplify the empirical Rademacher complexity for a more reasonable expression.

$$\text{Term (a)} \leq 4\mathbb{E}_{\varepsilon \in \{\pm 1\}^n} \left[ \sup_{f \in \mathcal{F}} \frac{1}{n} \sum_{i=1}^n \varepsilon_i \kappa(f \circ \phi(T_i), \psi(T_i)) \right] + 4\mathcal{B}_\kappa \sqrt{\frac{\log(1/v)}{n}}$$

$$\leq 4\mathcal{C}_\kappa \mathbb{E}_{\varepsilon \in \{\pm 1\}^n} \left[ \sup_{f \in \mathcal{F}} \frac{1}{n} \sum_{i=1}^n \varepsilon_i f \circ \phi(T_i) \right] + 4\mathcal{B}_\kappa \sqrt{\frac{\log(1/v)}{n}}$$

$$\leq 4\mathcal{C}_\kappa \mathcal{C}_f \mathbb{E}_{\varepsilon \in \{\pm 1\}^n} \left\| \frac{1}{n} \sum_{i=1}^n \varepsilon_i \phi(T_i) \right\| + 4\mathcal{B}_\kappa \sqrt{\frac{\log(1/v)}{n}}$$

$$\leq \frac{4}{n} \mathcal{C}_\kappa \mathcal{C}_f \mathbb{E}_{\varepsilon \in \{\pm 1\}^n} \sqrt{\left\| \sum_{i=1}^n \varepsilon_i \phi(T_i) \right\|^2} + 4\mathcal{B}_\kappa \sqrt{\frac{\log(1/v)}{n}}.$$

As the $\varepsilon_i$ are i.i.d. with zero mean as our definition, we cancel the term, thus

$$\text{Term (a)} \leq \frac{4}{n} \mathcal{C}_\kappa \mathcal{C}_f \sqrt{\sum_{i=1}^n \left\| \phi(T_i) \right\|^2} + 4\mathcal{B}_\kappa \sqrt{\frac{\log(1/v)}{n}}.$$

Then, we bound the term (b). To do this, we introduce a notation $f_{\mathcal{P}}^* = \arg\min_{f' \in \mathcal{F}} \mathcal{R}_{\mathcal{P}}(f' \circ h)$.

$$\text{Term (b)} = \min_{f' \in \mathcal{F}} \mathcal{R}_{\mathcal{T}}(f' \circ \phi) - \min_{f' \in \mathcal{F}} \mathcal{R}_{\mathcal{P}}(f' \circ \phi)$$

$$\leq \mathcal{R}_{\mathcal{T}}(f_{\mathcal{P}}^* \circ \phi) - \mathcal{R}_{\mathcal{P}}(f_{\mathcal{P}}^* \circ \phi)$$

$$= \mathbb{E}_{T \sim \mathcal{T}} \left[ \kappa(f_{\mathcal{P}}^* \circ \phi(T), \psi(T)) \right] - \mathbb{E}_{T \sim \mathcal{P}} \left[ \kappa(f_{\mathcal{P}}^* \circ \phi(T), \psi(T)) \right]$$

$$= \mathbb{E}_{x \sim \mathcal{T}_\phi} \left[ \kappa(f_{\mathcal{P}}^*(x), \psi(T)) \right] - \mathbb{E}_{x \sim \mathcal{P}_\phi} \left[ \kappa(f_{\mathcal{P}}^*(x), \psi(T)) \right]$$

$$\leq \mathcal{B}_\kappa \sum_{x \in \mathcal{X}_\phi} \left\| \mathcal{T}_\phi(x) - \mathcal{P}_\phi(x) \right\|,$$

where $\mathcal{B}_\kappa$ represents the upper bound of the Lipschitz constant of $\kappa$, and $\mathcal{X}_\phi$ denotes the distribution of task-tree embeddings produced by the encoder $\phi$. This term measures the distributional distance of task-trees between the pretraining and downstream distributions.

Following, we bound the term (c), as

$$\text{Term (c)} = \min_{f' \in \mathcal{F}} \mathcal{R}_{\mathcal{P}}(f' \circ \phi) - \min_{f' \in \mathcal{F}} \mathcal{R}_{\mathcal{P}}(f' \circ \phi^*)$$

$$\leq \mathcal{C}_{\delta} \Big( \min_{g' \in G} \mathcal{L}_{\mathcal{P}}(g' \circ h) - \min_{g' \in G} \mathcal{L}_{\mathcal{P}}(g' \circ \phi^*) \Big)^{\delta}$$

$$\leq \mathcal{C}_{\delta} \Big( \mathcal{L}_{\mathcal{P}}(g \circ h) - \min_{g' \in G, \phi' \in \Phi} \mathcal{L}_{\mathcal{P}}(g' \circ \phi') \Big)^{\delta}.$$

The term $\mathcal{L}_{\mathcal{P}}(g \circ h) - \min_{g' \in G, \phi' \in \Phi} \mathcal{L}_{\mathcal{P}}(g' \circ \phi')$ describes the excess risk on pretraining task, which can be replaced by a notation $\mathcal{E}_{\mathcal{P}}(g, \phi)$.

Lastly, we bound the term (d),

$$\text{Term (d)} = \min_{f' \in \mathcal{F}} \mathcal{R}_{\mathcal{P}}(f' \circ \phi^*) - \min_{f' \in \mathcal{F}, \phi' \in \Phi} \mathcal{R}_{\mathcal{T}}(f' \circ \phi')$$

$$= \min_{f' \in \mathcal{F}} \mathcal{R}_{\mathcal{P}}(f' \circ \phi^*) - \min_{f' \in \mathcal{F}} \mathcal{R}_{\mathcal{T}}(f \circ \phi^*) + \min_{f' \in \mathcal{F}} \mathcal{R}_{\mathcal{T}}(f \circ \phi^*) - \min_{f' \in \mathcal{F}, \phi' \in \Phi} \mathcal{R}_{\mathcal{T}}(f' \circ \phi')$$

$$\leq \mathcal{B}_{\kappa} \sum_{x \in \mathcal{X}_{\phi}} \Big\| \mathcal{T}_{\phi}(x) - \mathcal{P}_{\phi}(x) \Big\| + \min_{f' \in \mathcal{F}} \mathcal{R}_{\mathcal{T}}(f' \circ \phi^*) - \min_{f' \in \mathcal{F}, \phi' \in \Phi} \mathcal{R}_{\mathcal{T}}(f' \circ \phi').$$

By combining the four terms, we obtain the generalization bound for a model pretrained on task-tree distribution $\mathcal{P}$ and fine-tuned on task-tree distribution $\mathcal{T}$:

$$\mathcal{R}_{\mathcal{T}}(f \circ \phi) \leq \mathcal{C}_{\delta} \Big( \mathcal{E}_{\mathcal{P}}(g, \phi) \Big)^{\delta} + \frac{4\mathcal{C}_{\kappa}\mathcal{C}_f}{n} \sqrt{\sum_{i=1}^{n} \Big\| \phi(T_i) \Big\|^2} + \min_{f' \in \mathcal{F}} \mathcal{R}_{\mathcal{T}}(f' \circ \phi^*)$$

$$+ 2\mathcal{B}_{\kappa} \Big( \sum_{x \in \mathcal{X}_{\phi}} \Big\| \mathcal{T}_{\phi}(x) - \mathcal{P}_{\phi}(x) \Big\| + 2\sqrt{\frac{\log(1/v)}{n}} \Big).$$

We can set $\mathcal{C}_1 = \mathcal{C}_{\kappa}\mathcal{C}_f$ and $\mathcal{C}_2 = \mathcal{B}_{\kappa}$ as two downstream task-related constants. $\square$

## E. Experimental Settings

### E.1. Datasets

**Dataset Statistics.** We utilize 32 datasets spanning five domains in this paper. Since these datasets are text-attributed graphs, we use Sentence-BERT (Reimers & Gurevych, 2019) to align the node textual features into 768-dimensional vectors. The dataset statistics are presented in Table 8. For the temporal graphs, we split each graph into 10 snapshots, with the statistics shown in Figure 8. We classify `Children` and `Ratings` as heterophily graphs due to their relatively low homophily ratios (Chen et al., 2024b).

**Splitter.** For each dataset, we use the same splitting strategy as provided in the original paper (Chen et al., 2024b; Galkin et al., 2024; Feng et al., 2024; Zhang et al., 2024b). If multiple splits are provided, we evaluate model performance on each split using different random seeds. For datasets with a single split, we repeat the experiments five times with different random seeds. For `GDELT` and `ICEWS1819`, which are originally temporal knowledge graphs, we apply an 80%/10%/10% split based on timestamps for train/validation/test settings. For the temporal graphs `Enron` and `Googlemap CT` used for edge classification, we split each snapshot by timestamps, using the first 70% for training, the next 15% for validation, and the remaining 15% for testing.

### E.2. Baselines

BASELINES APPLICABLE FOR ALL GRAPHS

**GCN (Kipf & Welling, 2017).** A supervised message-passing GNN trained from scratch for each task. As a result, it cannot be applied to in-context learning or zero-shot learning.

Table 8: Statistics of 32 graphs used in the paper.

| Dataset | Domain | Task | # Nodes | # Edges | # Classes | # Task-Trees | Source |
|---|---|---|---|---|---|---|---|
| Products | E-commerce | Node, Link | 316,513 | 19,337,745 | 39 | 316,513 | (Chen et al., 2024b) |
| History | E-commerce | Node, Link | 41,551 | 503,180 | 12 | 41,551 | (Chen et al., 2024b) |
| Children | E-commerce | Node, Link | 76,875 | 2,325,044 | 24 | 76,875 | (Chen et al., 2024b) |
| Computer | E-commerce | Node, Link | 87,229 | 1,256,548 | 10 | 87,229 | (Chen et al., 2024b) |
| Photo | E-commerce | Node, Link | 48,362 | 873,793 | 12 | 48,362 | (Chen et al., 2024b) |
| Sportsfit | E-commerce | Node, Link | 173,055 | 3,020,134 | 13 | 173,055 | (Chen et al., 2024b) |
| Ratings | E-commerce | Node, Link | 24,492 | 186,100 | 5 | 24,492 | (Chen et al., 2024b) |
| Arxiv | Academia | Node, Link | 169,343 | 2,315,598 | 40 | 169,343 | (Chen et al., 2024b) |
| Cora | Academia | Node, Link | 2,708 | 10,556 | 7 | 2,708 | (Chen et al., 2024b) |
| Citeseer | Academia | Node, Link | 3,186 | 8,450 | 6 | 3,186 | (Chen et al., 2024b) |
| Pubmed | Academia | Node, Link | 19,717 | 88,648 | 3 | 19,717 | (Chen et al., 2024b) |
| Arxiv 23 | Academia | Node, Link | 46,198 | 77,726 | 40 | 46,198 | (Chen et al., 2024b) |
| DBLP | Academia | Node, Link | 14,376 | 431,326 | 4 | 14,376 | (Chen et al., 2024b) |
| WN18RR | knowledge Base | Link | 40,943 | 93,003 | 11 | 93,003 | (Galkin et al., 2024) |
| FB15K237 | knowledge Base | Link | 14,541 | 310,116 | 237 | 310,116 | (Galkin et al., 2024) |
| Codex Small | knowledge Base | Link | 2,034 | 36,543 | 42 | 36,543 | (Galkin et al., 2024) |
| Codex Median | knowledge Base | Link | 17,050 | 206,205 | 51 | 206,205 | (Galkin et al., 2024) |
| Codex Large | knowledge Base | Link | 77,951 | 612,437 | 69 | 612,437 | (Galkin et al., 2024) |
| NELL995 | knowledge Base | Link | 74,536 | 153,039 | 200 | 153,039 | (Galkin et al., 2024) |
| GDELT | knowledge Base | Link | 5,849 | 943,956 | 237 | 943,956 | (Zhang et al., 2024b) |
| ICEWS1819 | knowledge Base | Link | 31,796 | 1,100,071 | 266 | 1,100,071 | (Zhang et al., 2024b) |
| Chemblpre | Molecule | Graph | 8,845,648 | 19,123,034 | 1,295 | 341,952 | (Feng et al., 2024) |
| PCBA | Molecule | Graph | 11,349,235 | 24,566,048 | 128 | 437,092 | (Feng et al., 2024) |
| HIV | Molecule | Graph | 1,049,163 | 2,259,376 | 1 | 41,127 | (Feng et al., 2024) |
| BBBP | Molecule | Graph | 49,068 | 105,842 | 1 | 2,039 | (Feng et al., 2024) |
| BACE | Molecule | Graph | 51,577 | 111,536 | 1 | 1,513 | (Feng et al., 2024) |
| TOXCAST | Molecule | Graph | 161,002 | 330,180 | 588 | 8,575 | (Feng et al., 2024) |
| CYP450 | Molecule | Graph | 414,367 | 895,886 | 5 | 16,896 | (Feng et al., 2024) |
| TOX21 | Molecule | Graph | 145,459 | 302,190 | 12 | 7,831 | (Feng et al., 2024) |
| MUV | Molecule | Graph | 2,255,846 | 4,892,252 | 17 | 93,087 | (Feng et al., 2024) |
| Enron | Temporal | Link | 42,712 | 797,907 | 10 | 797,907 | (Zhang et al., 2024b) |
| Googlemap CT | Temporal | Link | 111,169 | 1,380,623 | 5 | 1,380,623 | (Zhang et al., 2024b) |

**GAT (Veličković et al., 2018).** A supervised GNN that uses an attention mechanism to learn the importance of received messages.

**GIN (Xu et al., 2019).** A supervised GNN specifically designed for graph-level tasks.

**BGRL (Thakoor et al., 2022).** A popular self-supervised learning framework for graphs that employs a contrastive learning loss without negative samples.

**GraphMAE (Hou et al., 2022).** A graph learning framework pretrained in a masked auto-encoder fashion.

**OFA (Liu et al., 2024a).** A cross-task and cross-domain graph foundation model that treats subgraphs as the basic learning instances. It introduces a graph prompt learning framework to enable in-context and zero-shot learning.

EXPERT MODELS DESIGNED FOR SPECIFIC DOMAINS

**ULTRA (Galkin et al., 2024).** A foundation model designed specifically for knowledge graphs, which we treat as the domain expert for KGs.

**KVPLM (Zeng et al., 2022).** A language model based on SMILES representations of molecules, serving as an expert model for molecular graphs.

**MoMu (Su et al., 2022).** Another expert model for molecules that leverages GNNs to improve molecular representations.

**Galactica (Taylor et al., 2022).** A foundation model for molecular graphs that utilizes multi-task learning with instructions.

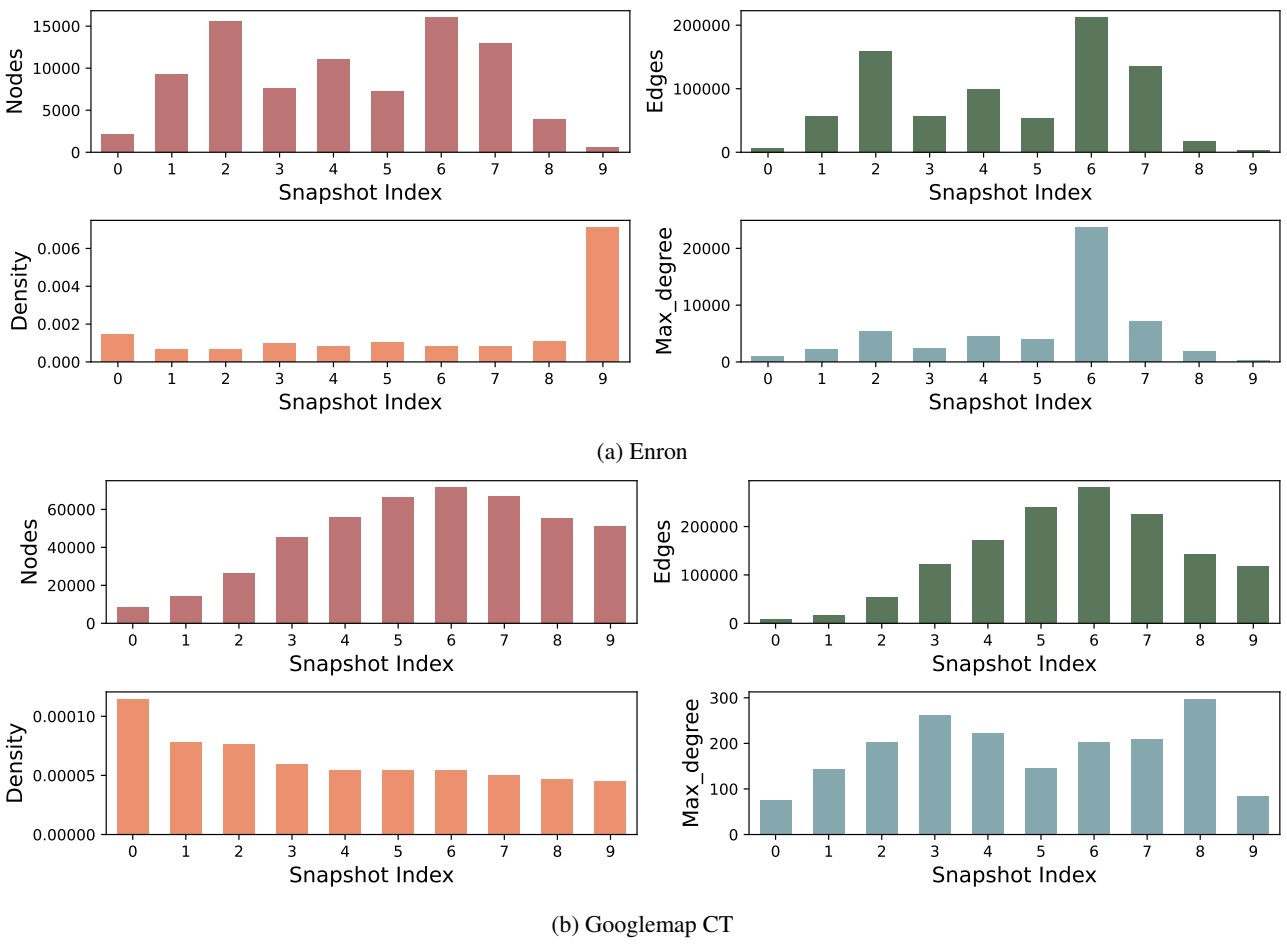

(a) Enron

(b) Googlemap CT

Figure 8: The statistics of temporal graphs.

**GIMLET (Zhao et al., 2023).** A foundation model for molecules that incorporates advanced models and instruction-based learning.

### E.3. Evaluation Protocol

**Pretraining Datasets.** We select six datasets for pretraining, including `Arxiv`, `Products`, `WN18RR`, `FB15K237`, `Chemblpre`, and `PCBA`, due to their diversity in domains and tasks. For self-supervised learning methods, these six datasets are used for pretraining unless otherwise specified.

**SFT Datasets.** For specialization via SFT in each domain, we use `Arxiv` for academic networks, `Products` for e-commerce networks, `FB15K237` for knowledge graphs, and `PCBA` for molecular networks. For temporal graphs, which are e-commerce-based, we also use `Products` for SFT to evaluate robustness under temporal distribution shifts.

**Backbone.** We use a GraphSAGE-like encoder (Hamilton et al., 2017). Following the encoding of task-trees, we add an additional linear transformation as the projector. Note that we does not leverage edge features to make the task harder except for `Enron` and `Googlemap CT` where node features are IDs and edge contains messages. As the edge information may significantly benefit some tasks like knowledge graph completion and molecule property prediction.

### E.4. Evaluation Settings

**Finetune.** This is the basic setting that directly finetune the full parameters of the pretrained model by appending a linear classifier on the top of the model encoder.

Table 9: The hyper-parameters used in the pretraining.

| Hidden Dim | Layers | Dropout | Activation | Epochs | LR |
|:---:|:---:|:---:|:---:|:---:|:---:|
| 768 | 2 | 0.15 | ReLU | 10 | 1e-7 |

| Feature Drop | Edge Drop | $\lambda$ | Decay | BS | Fanout |
|:---:|:---:|:---:|:---:|:---:|:---:|
| 0.2 | 0.2 | 10 | 1e-8 | 4,096 | 10 |

Table 10: The hyper-parameters used in fine-tuning on academic networks.

| **Academia** | Cora | Citeseer | Pubmed | Arxiv23 | DBLP | Arxiv |
|:---|:---:|:---:|:---:|:---:|:---:|:---:|
| Normalize | None | BN | None | None | None | BN |
| Learning Rate | 1e-4 | 1e-4 | 1e-5 | 1e-4 | 1e-4 | 1e-3 |
| Weight Decay | 0 | 0 | 1e-6 | 0 | 1e-6 | 1e-6 |
| Epochs | 1000 | 1000 | 1000 | 1000 | 1000 | 1000 |
| Early Stop | 200 | 200 | 200 | 200 | 200 | 200 |
| SFT Learning Rate | 1e-7 | 1e-4 | 1e-6 | 1e-5 | 1e-7 | 1e-6 |
| SFT Epochs | 100 | 100 | 100 | 100 | 100 | 100 |

Table 11: The hyper-parameters used in fine-tuning on e-commerce networks.

| **E-commerce** | History | Children | Computer | Photo | Sportsfit | Ratings | Products |
|:---|:---:|:---:|:---:|:---:|:---:|:---:|:---:|
| Normalize | None | None | None | None | BN | BN | BN |
| Learning Rate | 1e-3 | 1e-4 | 1e-3 | 1e-4 | 1e-4 | 1e-4 | 1e-4 |
| Weight Decay | 0 | 1e-6 | 0 | 1e-6 | 1e-6 | 1e-6 | 1e-6 |
| Epochs | 1000 | 1000 | 1000 | 1000 | 1000 | 1000 | 1000 |
| Early Stop | 200 | 200 | 200 | 200 | 200 | 200 | 200 |
| SFT Learning Rate | 1e-5 | 1e-7 | 1e-7 | 1e-7 | 1e-8 | 1e-7 | 1e-7 |
| SFT Epochs | 100 | 100 | 100 | 100 | 100 | 100 | 100 |

**In-context Learning.** This is a kind of few-shot learning without fine-tuning the model parameters. We randomly select $k$ samples from a certain class, and average the selected samples to form prototypes, which is used for classification. We follow existing GFM works (Liu et al., 2024a; He & Hooi, 2024) to conduct 500 randomly sampled 5-way 3-shot learning tasks. If the number of classes is less than 5, the number of ways is set to the number of classes.

**Zero-shot Learning.** The zero-shot learning is similar to in-context learning, yet we use the LLM-encoded class description embeddings as the prototypes for prediction. Similar to in-context learning, we also randomly sample 500 tasks for evaluation. Another zero-shot setting involves using an additional LLM for zero-shot inference (Chen et al., 2024a). We leave this in our future work.

### E.5. Hyper-Parameters

**Baselines.** For the baseline methods, we follow the hyperparameters reported in (Liu et al., 2024a; Chen et al., 2024b). If the hyperparameters are not provided, we set the number of epochs to 1,000, the batch size to 4,096, early stopping at 200, and the hidden dimension to 768, using a 2-layer GraphSAGE as the backbone with batch normalization and ReLU activation. For optimization, we use AdamW with a weight decay of 1e-6 and tune the learning rate from 1e-3, 1e-4, 1e-5, reporting the best performance. For methods with attention mechanisms, we set 4 attention heads.

**GIT.** The model architecture and pretraining parameters of our GIT are presented in Table 9. The specific fine-tuning

Table 12: The hyper-parameters used in fine-tuning on knowledge graphs.

| KG | WN18RR | Codex-S | Codex-M | Codex-L | NELL995 | GDELT | ICEWS1819 | FB15K237 |
|---|---|---|---|---|---|---|---|---|
| Normalize | BN | BN | BN | BN | BN | BN | BN | BN |
| Learning Rate | 1e-4 | 1e-4 | 1e-3 | 1e-3 | 1e-3 | 1e-3 | 1e-3 | 1e-3 |
| Weight Decay | 1e-6 | 1e-6 | 1e-6 | 1e-6 | 0 | 0 | 1e-6 | 1e-6 |
| Epochs | 1000 | 1000 | 1000 | 1000 | 1000 | 1000 | 1000 | 1000 |
| Early Stop | 200 | 200 | 200 | 200 | 200 | 200 | 200 | 200 |
| SFT Learning Rate | 1e-8 | 1e-7 | 1e-5 | 1e-7 | 1e-8 | 1e-4 | 1e-8 | 1e-7 |
| SFT Epochs | 100 | 100 | 100 | 100 | 100 | 100 | 100 | 100 |

Table 13: The hyper-parameters used in fine-tuning on molecule graphs.

| Molecule | BBBP | BACE | TOXCAST | TOX21 | CYP450 | HIV | MIV | PCBA |
|---|---|---|---|---|---|---|---|---|
| Normalize | BN | BN | BN | BN | BN | BN | None | BN |
| Learning Rate | 1e-3 | 1e-4 | 1e-4 | 1e-4 | 1e-4 | 1e-5 | 1e-4 | 1e-5 |
| Weight Decay | 1e-6 | 0 | 1e-6 | 1e-6 | 1e-6 | 0 | 0 | 0 |
| Epochs | 300 | 300 | 300 | 300 | 300 | 300 | 300 | 300 |
| Early Stop | 30 | 30 | 30 | 30 | 30 | 30 | 30 | 30 |
| SFT Learning Rate | 1e-7 | 1e-6 | 1e-8 | 1e-7 | 1e-7 | 1e-7 | 1e-6 | 1e-7 |
| SFT Epochs | 10 | 10 | 10 | 10 | 10 | 10 | 10 | 10 |

Table 14: The hyper-parameters used in fine-tuning on temporal graphs.

| Temporal | Enron | Googlemap CT |
|---|---|---|
| Normalize | None | None |
| Learning Rate | 1e-3 | 1e-3 |
| Weight Decay | 1e-6 | 1e-6 |
| Epochs | 1000 | 1000 |
| Early Stop | 200 | 200 |
| SFT Learning Rate | 1e-6 | 1e-6 |
| SFT Epochs | 100 | 100 |

hyperparameters, categorized by domain, are shown in Tables 10, 11, 12, 13, and 14. For in-context learning and zero-shot learning results without fine-tuning, the general model does not involve any hyperparameters. For the specialized model, we tune the hyperparameters of SFT epochs from 10 to 500, in steps of 10, the SFT learning rate from 1e-4, 1e-5, 1e-6, 1e-7, 1e-8, and the normalization method from None, BN.

## F. Comprehensive Model Results

### F.1. Domain: Academia

**Node Classification.** We perform node classification on academic networks across three settings: basic fine-tuning, 3-shot in-context learning, and zero-shot learning. The comprehensive node classification results on academic networks, measured in terms of accuracy, are presented in Table 15. Notably, the specialized model (GIT-S) does not always outperform the general model (GIT-G). This may be because the manually selected SFT data does not adequately capture the underlying distribution of the domain. It would be valuable to explore dataset selection or instance selection methods to better optimize the choice of SFT data.

Table 15: Node classification results on academic networks in terms of accuracy.

| | | Cora | Citeseer | Pubmed | Arxiv23 | DBLP | Arxiv | **Avg.** |
|---|---|---|---|---|---|---|---|---|
| **0-shot** | GCN | - | - | - | - | - | - | - |
| | BGRL | 14.37 ± 0.38 | 15.09 ± 0.40 | 33.94 ± 0.46 | 2.44 ± 0.23 | 25.53 ± 0.27 | 2.10 ± 0.14 | 15.58 |
| | GraphMAE | 13.88 ± 0.41 | 13.48 ± 0.83 | 32.62 ± 0.67 | 2.51 ± 0.37 | 27.83 ± 0.40 | 2.17 ± 0.26 | 15.42 |
| | OFA | 14.58 ± 0.43 | 13.28 ± 0.12 | 30.89 ± 0.10 | 2.08 ± 0.03 | 21.00 ± 0.27 | 2.05 ± 0.18 | 13.98 |
| | GIT - G | 15.31 ± 0.27 | 16.04 ± 0.31 | 29.66 ± 0.60 | 2.89 ± 0.25 | 21.80 ± 0.35 | 3.57 ± 0.18 | 14.88 |
| | GIT - S | **18.26 ± 0.29** | **20.35 ± 0.29** | **39.12 ± 0.55** | **9.08 ± 0.32** | **36.40 ± 0.58** | **17.50 ± 0.66** | **23.45** |
| **3-shot** | GCN | - | - | - | - | - | - | - |
| | BGRL | 61.24 ± 0.50 | 44.97 ± 0.43 | 54.55 ± 0.81 | 43.17 ± 0.93 | 42.89 ± 0.61 | 59.09 ± 0.24 | 50.99 |
| | GraphMAE | 62.02 ± 0.58 | 44.08 ± 0.59 | 55.98 ± 0.68 | 31.64 ± 0.28 | 38.16 ± 0.54 | 63.62 ± 0.79 | 49.25 |
| | OFA | 55.92 ± 0.40 | 41.57 ± 0.32 | 40.89 ± 0.79 | 37.01 ± 0.41 | 43.08 ± 0.51 | 57.08 ± 0.48 | 45.93 |
| | GIT - G | 60.93 ± 0.47 | 48.32 ± 0.53 | **60.30 ± 0.76** | 45.62 ± 0.35 | 44.76 ± 0.54 | **64.07 ± 0.50** | 54.00 |
| | GIT - S | **63.23 ± 0.29** | **49.55 ± 0.33** | 59.62 ± 0.54 | **47.21 ± 0.31** | **47.40 ± 0.43** | 64.06 ± 0.58 | **55.18** |
| **Finetune** | GCN | 77.40 ± 1.36 | 80.19 ± 1.30 | 72.44 ± 2.08 | 71.61 ± 0.02 | 68.15 ± 0.14 | 71.65 ± 0.02 | 73.57 |
| | BGRL | 71.06 ± 2.84 | 80.56 ± 1.59 | 68.75 ± 3.69 | 69.23 ± 0.19 | 55.66 ± 2.00 | 67.62 ± 0.19 | 68.81 |
| | GraphMAE | 76.34 ± 1.49 | 79.19 ± 1.32 | 73.88 ± 1.16 | 70.46 ± 0.04 | 71.18 ± 0.13 | 71.82 ± 0.05 | 73.81 |
| | OFA | 70.63 ± 1.03 | 79.13 ± 2.53 | 70.95 ± 1.02 | 70.43 ± 0.12 | 70.67 ± 0.21 | 71.28 ± 0.24 | 72.18 |
| | GIT - G | 78.74 ± 1.12 | 81.03 ± 0.78 | 75.26 ± 2.81 | **72.49 ± 0.07** | **74.42 ± 0.15** | 72.99 ± 0.10 | 75.82 |
| | GIT - S | **78.90 ± 1.44** | **81.97 ± 0.80** | **76.17 ± 1.70** | 71.50 ± 0.08 | 73.59 ± 0.08 | **73.13 ± 0.11** | **75.88** |

**Link Prediction.** We present the link prediction results, measured by AUC, on academic networks in Table 16. The train/val/test sets are randomly split in a 70%/15%/15% ratio. GIT outperforms all baselines across all settings. Additionally, the specialized GIT surpasses the general GIT, highlighting the potential of specialization to enhance performance on other tasks within the same domain. This finding underscores the cross-task transferability of the proposed specialization process.

Table 16: Link prediction results on academic networks in terms of AUC.

| | Cora | Citeseer | Pubmed | Arxiv23 | DBLP | Arxiv | **Avg.** |
|---|---|---|---|---|---|---|---|
| GCN | 87.34 ± 0.88 | 87.52 ± 0.98 | 84.41 ± 0.17 | 89.67 ± 0.24 | **98.29 ± 0.07** | **97.50 ± 0.08** | 90.79 |
| BGRL | 83.96 ± 0.36 | 81.51 ± 0.85 | 84.01 ± 0.60 | 86.42 ± 0.08 | 97.24 ± 0.06 | 96.80 ± 0.04 | 88.32 |
| GraphMAE | 85.57 ± 0.27 | 84.55 ± 0.69 | **89.83 ± 0.35** | 91.45 ± 0.44 | 98.05 ± 0.06 | 96.31 ± 0.02 | 90.96 |
| OFA | 82.82 ± 0.72 | 81.52 ± 1.16 | 84.78 ± 1.08 | 85.40 ± 0.62 | 97.23 ± 0.14 | 96.46 ± 0.05 | 88.04 |
| GIT - G | 87.79 ± 2.07 | 87.59 ± 0.96 | 84.35 ± 0.26 | 91.47 ± 0.46 | 98.25 ± 0.09 | 97.14 ± 0.06 | 91.10 |
| GIT - S | **88.58 ± 1.88** | **88.50 ± 1.15** | 87.78 ± 0.13 | **91.86 ± 0.38** | 98.27 ± 0.05 | 97.30 ± 0.05 | **92.05** |

### F.2. Domain: E-Commerce

**Node Classification.** The comprehensive node classification results on e-commerce datasets are presented in Table 17. Our proposed GIT model outperforms the baselines in most settings, particularly for the specialized version. Specialization significantly improves performance in zero-shot and in-context learning, highlighting the advantages of using task-trees as the basic learning instances. In the basic fine-tuning setting, we also observe that supervised methods (GCN and GAT) generally outperform self-supervised methods, such as GraphMAE (Hou et al., 2022) and OFA (Liu et al., 2024a), indicating the occurrence of negative transfer. However, GIT surpasses these supervised methods on 5 out of 7 datasets, further demonstrating the benefits of task-trees as basic learning instances. It it important to note that we consider `Children` and `Ratings` as heterophily graphs (Chen et al., 2024b) due to their low homophily ratio.

**Link Prediction.** The link prediction results on e-commerce networks (`History`, `Photo`, `Ratings`) are presented in Table 18. We randomly select 70% of the edges for training, 15% for validation, and the remaining 15% for testing. Our GIT model achieves the best average performance across these three e-commerce graphs. However, other baselines like BGRL,

Table 17: Node classification results on e-commerce networks in terms of accuracy.

| | | History | Children | Computer | Photo | Sportsfit | Ratings | Products | **Avg.** |
|---|---|---|---|---|---|---|---|---|---|
| **0-shot** | GCN | - | - | - | - | - | - | - | - |
| | GAT | - | - | - | - | - | - | - | - |
| | BGRL | 6.76 ± 0.18 | 4.26 ± 0.14 | 9.70 ± 0.39 | 6.32 ± 0.20 | 7.91 ± 0.31 | 17.50 ± 0.65 | 0.58 ± 0.19 | 7.58 |
| | GraphMAE | 9.20 ± 0.23 | 4.25 ± 0.13 | 7.86 ± 0.23 | 8.02 ± 0.40 | 7.70 ± 0.34 | 20.26 ± 0.16 | 0.07 ± 0.03 | 8.19 |
| | OFA | 8.84 ± 0.52 | 4.22 ± 0.19 | 10.83 ± 0.32 | 8.46 ± 0.34 | 7.28 ± 0.52 | 18.43 ± 0.50 | 3.02 ± 0.37 | 8.73 |
| | GIT - G | 4.72 ± 0.31 | 4.34 ± 0.19 | 8.85 ± 0.30 | **11.78 ± 0.26** | 7.20 ± 0.18 | 21.00 ± 0.06 | 3.64 ± 0.25 | 8.79 |
| | GIT - S | **9.94 ± 0.54** | **7.49 ± 0.12** | **14.87 ± 0.40** | 9.69 ± 0.22 | **9.23 ± 0.65** | **21.55 ± 0.31** | **46.62 ± 1.06** | **17.06** |
| **3-shot** | GCN | - | - | - | - | - | - | - | - |
| | GAT | - | - | - | - | - | - | - | - |
| | BGRL | 38.35 ± 0.51 | 32.93 ± 0.75 | 50.90 ± 0.82 | 61.64 ± 0.51 | 42.99 ± 0.48 | **21.67 ± 0.21** | 71.71 ± 0.23 | 45.74 |
| | GraphMAE | 42.28 ± 0.38 | 38.71 ± 0.49 | 58.24 ± 0.79 | 59.47 ± 0.25 | 46.57 ± 0.46 | 21.11 ± 0.56 | 71.01 ± 0.67 | 48.20 |
| | OFA | 48.87 ± 0.26 | 47.13 ± 0.32 | 68.14 ± 0.49 | 75.73 ± 0.24 | 63.56 ± 0.57 | 21.38 ± 0.16 | 74.58 ± 0.33 | 57.06 |
| | GIT - G | 50.78 ± 0.41 | 47.55 ± 0.26 | 66.64 ± 0.50 | 75.43 ± 0.26 | 64.56 ± 0.43 | 21.21 ± 0.37 | 74.35 ± 0.48 | 57.22 |
| | GIT - S | **50.99 ± 0.64** | **47.65 ± 0.36** | **69.29 ± 0.48** | **76.32 ± 0.55** | **65.84 ± 0.53** | 21.17 ± 0.40 | **74.80 ± 0.54** | **58.01** |
| **Finetune** | GCN | 84.62 ± 0.06 | 58.08 ± 0.08 | 88.41 ± 0.06 | 86.39 ± 0.11 | 92.07 ± 0.02 | 50.99 ± 0.23 | 86.91 ± 0.05 | 78.21 |
| | GAT | 84.54 ± 0.07 | 59.09 ± 0.05 | **89.00 ± 0.04** | **86.70 ± 0.07** | 91.12 ± 0.05 | 51.19 ± 0.15 | 87.22 ± 0.05 | 78.41 |
| | GraphMAE | 82.51 ± 0.05 | 56.76 ± 0.09 | 84.31 ± 0.06 | 83.26 ± 0.06 | 90.47 ± 0.03 | 52.39 ± 0.29 | 86.30 ± 0.07 | 76.57 |
| | OFA | 82.81 ± 0.11 | 55.43 ± 0.08 | 85.78 ± 0.13 | 83.21 ± 0.25 | 91.23 ± 0.07 | 51.79 ± 0.18 | 86.23 ± 0.07 | 76.64 |
| | GIT - G | 84.94 ± 0.10 | 59.09 ± 0.15 | 87.81 ± 0.10 | 85.66 ± 0.06 | 92.17 ± 0.06 | 52.45 ± 0.26 | 87.75 ± 0.04 | 78.61 |
| | GIT - S | **85.18 ± 0.11** | **59.73 ± 0.12** | 88.05 ± 0.18 | 85.66 ± 0.05 | **92.44 ± 0.02** | **52.56 ± 0.29** | **88.20 ± 0.05** | **78.83** |

GraphMAE, and OFA fail to outperform the basic GCN. This may be because they struggle to acquire useful knowledge during pretraining for tasks that require structural insight, such as link prediction. These results underscore the advantages of using task-trees as the basic learning instances.

Table 18: Link prediction results on e-commerce networks in terms of AUC.

| | History | Photo | Ratings | **Avg.** |
|---|---|---|---|---|
| GCN | **97.87 ± 0.06** | 97.37 ± 0.03 | 97.77 ± 0.07 | 97.67 |
| BGRL | 96.40 ± 0.08 | 97.58 ± 0.04 | 98.05 ± 0.04 | 97.34 |
| GraphMAE | 97.59 ± 0.06 | 98.09 ± 0.05 | 95.35 ± 0.15 | 97.01 |
| OFA | 95.86 ± 0.09 | 97.05 ± 0.06 | 97.79 ± 0.12 | 96.90 |
| GIT - G | 96.55 ± 0.07 | 96.24 ± 0.05 | 98.45 ± 0.07 | 97.08 |
| GIT - S | 97.08 ± 0.05 | **97.80 ± 0.06** | **98.49 ± 0.10** | **97.79** |

## F.3. Domain: Knowledge Base

**Edge Classification.** The edge classification results on knowledge graphs are presented in Table 19. In this domain, our GIT model significantly outperforms the existing baselines, demonstrating the advantages of using task-trees as the basic learning instances for knowledge bases, even though these KGs represent different scenarios. We hypothesize that this improvement stems from the nature of relation triplets in KGs, where each relation inherently describes the aggregation of the head and tail nodes, aligning with the concept of task-trees.

## F.4. Domain: Molecule

**Graph Classification.** We evaluate fine-tuning, in-context learning, and zero-shot learning in this domain. The fine-tuning and in-context learning settings are consistent with those used in previous domains. For zero-shot learning, however, we follow the approach of Zhao et al. (2023) by assessing zero-shot performance on the original test set. The graph classification results are presented in Table 20. Our GIT model achieves the best average performance across the three evaluated settings. We also observe that specialization consistently improves performance across different graphs, aligning with our theoretical

Table 19: Edge classification results on knowledge graphs in terms of accuracy.

| | | WN18RR | Codex-S | Codex-M | Codex-L | NELL995 | GDELT | ICEWS1819 | FB15K237 | **Avg.** |
|---|---|---|---|---|---|---|---|---|---|---|
| **3-shot** | GCN | - | - | - | - | - | - | - | - | - |
| | GraphMAE | 55.20 ± 0.52 | 61.41 ± 0.86 | 54.30 ± 0.42 | 61.01 ± 0.55 | 86.42 ± 0.53 | 32.43 ± 0.48 | 31.58 ± 0.39 | 70.15 ± 0.75 | 56.56 |
| | OFA | 55.27 ± 0.64 | 55.14 ± 0.34 | 50.20 ± 0.68 | 62.40 ± 0.46 | 88.41 ± 0.38 | 30.23 ± 0.50 | 34.94 ± 0.32 | 79.15 ± 0.45 | 56.97 |
| | GIT - G | 55.80 ± 0.32 | 76.96 ± 0.43 | **73.79 ± 0.43** | **78.54 ± 0.51** | 89.13 ± 0.48 | 34.30 ± 0.68 | **42.07 ± 0.75** | 89.78 ± 0.46 | 67.55 |
| | GIT - S | **57.90 ± 0.97** | **77.19 ± 0.32** | 72.14 ± 0.84 | 76.99 ± 0.72 | **90.80 ± 0.51** | **34.85 ± 0.69** | 42.02 ± 0.65 | **90.49 ± 0.32** | **67.80** |
| **Finetune** | GCN | 86.77 ± 0.30 | 93.56 ± 2.11 | 85.73 ± 1.84 | 84.45 ± 0.18 | 79.06 ± 0.32 | 11.72 ± 0.05 | 27.53 ± 0.06 | 66.07 ± 0.26 | 66.86 |
| | GraphMAE | 93.87 ± 0.35 | 97.09 ± 0.72 | 94.07 ± 0.60 | 94.18 ± 0.19 | 86.10 ± 0.42 | 13.12 ± 0.04 | 28.91 ± 0.06 | 73.52 ± 0.12 | 72.61 |
| | OFA | 93.10 ± 0.31 | 90.78 ± 5.46 | 93.83 ± 3.28 | 93.26 ± 0.59 | 86.91 ± 1.50 | 14.48 ± 0.03 | 30.60 ± 0.63 | 76.08 ± 1.95 | 72.38 |
| | GIT - G | 94.16 ± 0.11 | 98.08 ± 0.08 | 97.89 ± 0.04 | **96.85 ± 0.03** | 90.10 ± 0.23 | 14.86 ± 0.12 | **33.49 ± 0.06** | **80.39 ± 0.13** | 75.73 |
| | GIT - S | **95.15 ± 0.07** | **99.19 ± 0.04** | **97.92 ± 0.04** | 96.83 ± 0.04 | **91.28 ± 0.41** | **14.89 ± 0.05** | 33.61 ± 0.10 | 80.32 ± 0.07 | **76.15** |

analysis.

Table 20: Graph classification results on molecule graphs in terms of AUC.

| | | HIV | BBBP | BACE | TOXCAST | CYP450 | TOX21 | MUV | PCBA | **Avg.** |
|---|---|---|---|---|---|---|---|---|---|---|
| **0-shot** | GIN | - | - | - | - | - | - | - | - | - |
| | BGRL | 55.27 | 53.72 | 33.74 | 49.00 | 60.99 | 46.40 | 39.90 | 42.39 | 47.68 |
| | GraphMAE | 46.48 | 49.08 | 30.76 | 48.22 | 60.55 | 49.17 | 48.17 | 45.10 | 47.19 |
| | OFA | 47.96 | 50.61 | 34.35 | 49.70 | 61.96 | 52.73 | 52.48 | 54.14 | 50.49 |
| | GIT - G | 56.76 | 54.76 | 33.66 | 51.55 | 63.21 | 56.83 | 53.71 | 56.25 | 53.34 |
| | GIT - S | **66.14** | **62.16** | **52.27** | **58.30** | **69.75** | **63.45** | **65.32** | **65.26** | **62.83** |
| **3-shot** | GIN | - | - | - | - | - | - | - | - | - |
| | BGRL | 52.72 ± 1.84 | 49.12 ± 0.78 | 59.58 ± 0.89 | **57.27 ± 0.05** | 67.49 ± 0.56 | 59.26 ± 0.19 | 52.61 ± 0.23 | 51.48 ± 0.22 | 56.19 |
| | GraphMAE | 54.40 ± 1.04 | 48.41 ± 1.34 | 60.78 ± 1.01 | 56.99 ± 0.06 | 66.93 ± 0.91 | 58.40 ± 0.22 | 51.95 ± 0.18 | 50.24 ± 0.23 | 56.01 |
| | OFA | **56.04 ± 1.49** | 50.69 ± 1.36 | 60.21 ± 0.64 | 56.40 ± 0.05 | 68.76 ± 0.16 | 57.18 ± 0.29 | 56.17 ± 0.23 | 50.77 ± 0.30 | 57.03 |
| | GIT - G | 52.42 ± 1.74 | 48.22 ± 1.14 | 59.32 ± 0.91 | 56.32 ± 0.04 | 66.77 ± 0.45 | 58.53 ± 0.36 | 55.98 ± 0.19 | 50.09 ± 0.30 | 55.96 |
| | GIT - S | 54.12 ± 1.66 | **66.74 ± 1.34** | **61.76 ± 0.92** | 55.53 ± 0.03 | **81.50 ± 0.23** | **65.16 ± 0.27** | **66.14 ± 0.30** | **51.58 ± 0.30** | 62.82 |
| **Finetune** | GIN | **76.83 ± 1.32** | 67.36 ± 1.39 | 75.55 ± 2.91 | 62.92 ± 0.42 | 85.82 ± 0.77 | 72.26 ± 0.24 | 70.12 ± 0.39 | 78.34 ± 0.51 | 73.65 |
| | BGRL | 72.18 ± 1.24 | 67.40 ± 1.45 | 73.75 ± 3.69 | 62.52 ± 0.10 | 83.10 ± 0.26 | 72.97 ± 0.54 | 68.46 ± 0.63 | 76.69 ± 1.40 | 72.13 |
| | GraphMAE | 69.54 ± 2.59 | 66.43 ± 2.48 | 66.56 ± 4.73 | 62.52 ± 0.14 | 86.64 ± 0.27 | **74.13 ± 0.41** | 70.12 ± 0.40 | 75.34 ± 1.33 | 71.41 |
| | OFA | 76.48 ± 2.11 | 65.79 ± 0.96 | 77.88 ± 1.08 | 63.49 ± 0.61 | 85.77 ± 0.32 | 73.00 ± 0.67 | 69.53 ± 0.56 | 80.31 ± 1.20 | 74.03 |
| | GIT - G | 73.63 ± 0.77 | 68.33 ± 1.06 | 79.28 ± 2.71 | 63.00 ± 0.43 | 86.86 ± 0.22 | 73.81 ± 0.33 | 70.49 ± 0.51 | 81.13 ± 0.53 | 74.57 |
| | GIT - S | 74.75 ± 0.42 | **68.72 ± 1.13** | **81.10 ± 0.61** | **63.63 ± 0.61** | **87.00 ± 0.37** | 73.78 ± 0.77 | **71.16 ± 0.51** | **81.43 ± 0.34** | **75.20** |

## F.5. Domain: Temporal E-Commerce

**Edge Classification.** We report the experimental results on two temporal graphs, `Enron` and `Googlemap CT`, in Table 21 and Table 22, respectively. The original graph is split into ten snapshots based on timestamps, and the model performance is evaluated separately on each snapshot. Since we fine-tuned the pretrained model on `Products`, these experiments assess the model's robustness to temporal distribution shifts. The results demonstrate GIT's capability to effectively handle temporal information in graphs.

Table 21: Edge classification results on temporal graph `Enron`.

| | | Enron 1 | Enron 2 | Enron 3 | Enron 4 | Enron 5 | Enron 6 | Enron 7 | Enron 8 | Enron 9 | Enron 10 | **Avg.** |
|---|---|---|---|---|---|---|---|---|---|---|---|---|
| **Finetune** | GAT | 81.36 ± 0.08 | 60.60 ± 0.88 | 62.40 ± 1.83 | 83.49 ± 0.25 | 45.88 ± 0.34 | 65.97 ± 1.07 | 48.14 ± 0.23 | 59.15 ± 0.65 | **82.39 ± 1.98** | 45.35 ± 0.43 | 63.47 |
| | GraphMAE | 81.29 ± 0.01 | 59.52 ± 0.10 | 66.13 ± 1.42 | 82.84 ± 0.67 | 50.01 ± 0.34 | 64.46 ± 0.75 | 45.16 ± 0.15 | 67.25 ± 0.21 | 72.05 ± 3.27 | 48.00 ± 0.01 | 63.67 |
| | GIT - G | **81.48 ± 0.28** | 61.25 ± 0.25 | 67.56 ± 2.16 | 84.50 ± 0.21 | **52.52 ± 0.80** | 67.69 ± 0.54 | **50.32 ± 0.17** | 68.35 ± 0.51 | 76.92 ± 1.16 | 48.28 ± 0.15 | 65.89 |
| | GIT - S | 81.27 ± 0.12 | **61.42 ± 0.12** | **69.15 ± 0.43** | **84.51 ± 0.17** | 51.93 ± 0.48 | 66.74 ± 1.24 | 50.12 ± 0.32 | **68.89 ± 0.64** | 77.03 ± 2.05 | **48.35 ± 0.02** | **65.94** |
| **3-shot** | GAT | - | - | - | - | - | - | - | - | - | - | - |
| | OFA | 68.91 ± 0.31 | 58.27 ± 0.41 | **62.43 ± 0.60** | 55.48 ± 0.59 | 61.46 ± 0.22 | 50.35 ± 0.75 | 53.44 ± 0.37 | 49.01 ± 0.55 | 56.43 ± 0.70 | 59.01 ± 0.19 | 57.48 |
| | GraphMAE | 73.23 ± 0.76 | 58.53 ± 0.66 | 61.66 ± 0.61 | 58.15 ± 0.52 | 59.81 ± 0.50 | **50.59 ± 0.60** | **56.89 ± 0.74** | **56.08 ± 0.45** | 59.69 ± 0.44 | 63.63 ± 0.62 | 59.83 |
| | GIT - G | 71.67 ± 0.43 | **60.31 ± 0.49** | 61.46 ± 0.59 | 57.62 ± 0.56 | 59.60 ± 0.93 | 50.82 ± 0.40 | 54.02 ± 0.58 | 52.22 ± 0.32 | 60.61 ± 0.29 | 62.17 ± 0.34 | 59.05 |
| | GIT - S | **73.73 ± 0.50** | 58.96 ± 0.43 | 60.08 ± 0.45 | **59.38 ± 0.56** | **61.84 ± 0.78** | 50.43 ± 0.72 | 54.92 ± 0.22 | 56.03 ± 0.43 | **61.99 ± 0.80** | **64.85 ± 0.50** | **60.22** |

Table 22: Edge classification results on temporal graph `Googlemap CT`.

| | | GCT 1 | GCT 2 | GCT 3 | GCT 4 | GCT 5 | GCT 6 | GCT 7 | GCT 8 | GCT 9 | GCT 10 | **Avg.** |
|---|---|---|---|---|---|---|---|---|---|---|---|---|
| **Finetune** | GAT | 61.29 ± 0.04 | 56.29 ± 0.03 | 56.13 ± 0.08 | 57.32 ± 0.06 | 60.12 ± 0.08 | 61.65 ± 0.13 | 63.37 ± 0.06 | 64.71 ± 0.06 | 67.08 ± 0.06 | 69.46 ± 0.04 | 61.74 |
| | GraphMAE | 64.60 ± 0.42 | 57.61 ± 0.32 | 55.63 ± 0.24 | 57.08 ± 0.25 | 60.36 ± 0.19 | 60.99 ± 0.08 | 62.90 ± 0.06 | 63.83 ± 0.09 | 66.89 ± 0.12 | 68.35 ± 0.06 | 61.82 |
| | GIT - G | 64.21 ± 1.10 | 59.06 ± 0.20 | **57.12 ± 0.23** | **59.85 ± 0.20** | 61.92 ± 0.11 | **62.91 ± 0.10** | 64.02 ± 0.04 | **65.62 ± 0.14** | **67.66 ± 0.11** | 70.51 ± 0.10 | 63.29 |
| | GIT - S | **66.52 ± 0.29** | **58.63 ± 0.69** | 56.82 ± 0.23 | 59.77 ± 0.46 | **61.93 ± 0.22** | 62.72 ± 0.18 | **64.08 ± 0.07** | 65.56 ± 0.11 | 67.49 ± 0.18 | **70.62 ± 0.11** | **63.41** |
| **3-shot** | GAT | - | - | - | - | - | - | - | - | - | - | - |
| | OFA | 20.62 ± 0.34 | 21.22 ± 0.56 | 20.10 ± 0.32 | 20.16 ± 0.21 | 20.25 ± 0.36 | 20.39 ± 0.50 | 20.13 ± 0.14 | 20.21 ± 0.25 | 19.90 ± 0.30 | 20.59 ± 0.22 | 20.36 |
| | GraphMAE | 21.15 ± 0.44 | 21.03 ± 0.39 | 21.73 ± 0.23 | 21.60 ± 0.53 | 19.73 ± 0.41 | 20.38 ± 0.28 | 20.62 ± 0.22 | 20.51 ± 0.27 | 19.63 ± 0.43 | 21.38 ± 0.35 | 20.78 |
| | GIT - G | 21.81 ± 0.29 | 21.94 ± 0.23 | 20.78 ± 0.31 | 20.61 ± 0.40 | 20.73 ± 0.37 | 20.33 ± 0.56 | 20.90 ± 0.32 | 20.57 ± 0.46 | 20.58 ± 0.33 | 20.28 ± 0.31 | 20.85 |
| | GIT - S | **25.21 ± 0.53** | **24.21 ± 0.43** | **23.43 ± 0.44** | **22.41 ± 0.14** | **21.83 ± 0.59** | **21.65 ± 0.33** | **21.41 ± 0.50** | **21.76 ± 0.57** | **21.72 ± 0.22** | **21.72 ± 0.50** | **22.54** |

# G. Comparison to Domain Experts

### G.1. Domain: Knowledge Base

In addition to comparing GIT to standard baselines applicable for all graphs, we also evaluate it against ULTRA, a foundation model specifically designed for knowledge graphs. As a domain expert, ULTRA is compared to Expert GIT (pretrained on all KGs) and Specialized GIT (pretrained on default datasets and fine-tuned on `FB15K237`), with the results presented in Table 23. We find that the two domain experts, ULTRA and Expert GIT, achieve comparable performance, though ULTRA significantly outperforms Expert GIT in certain settings. This may be due to ULTRA learning more fine-grained relational information within KGs. Notably, Specialized GIT also performs comparably to both domain experts, highlighting the potential of specialization. We believe this is because the distributions of KGs are more similar to each other compared to graphs from other domains.

Table 23: Comparison between GIT and ULTRA, a foundation model designed for knowledge graphs. The Expert GIT is pretrained on all KGs used in the paper.

| | **3-shot** | | | **Finetune** | | |
|---|---|---|---|---|---|---|
| | ULTRA | Expert GIT | Specialized GIT | ULTRA | Expert GIT | Specialized GIT |
| `WN18RR` | **60.69 ± 0.82** | 55.83 ± 0.44 | 57.90 ± 0.97 | **96.35 ± 0.22** | 95.12 ± 0.05 | 95.15 ± 0.07 |
| `Codex-S` | **82.45 ± 0.53** | 76.07 ± 0.41 | 77.19 ± 0.32 | 98.27 ± 0.36 | 99.14 ± 0.07 | **99.19 ± 0.04** |
| `Codex-M` | **74.35 ± 0.23** | 73.54 ± 0.46 | 72.14 ± 0.84 | 96.90 ± 0.11 | 97.90 ± 0.06 | **97.92 ± 0.04** |
| `Codex-L` | 75.98 ± 0.48 | **78.13 ± 0.36** | 76.99 ± 0.72 | 96.22 ± 0.04 | **96.84 ± 0.04** | 96.83 ± 0.04 |
| `NELL995` | 90.22 ± 0.46 | 89.99 ± 0.24 | **90.80 ± 0.51** | 89.46 ± 0.28 | 90.55 ± 0.59 | **91.28 ± 0.41** |
| `GDELT` | 33.89 ± 0.33 | **34.92 ± 0.55** | 34.85 ± 0.69 | 14.63 ± 0.02 | **14.91 ± 0.10** | 14.89 ± 0.05 |
| `ICEWS1819` | 41.37 ± 0.53 | **42.42 ± 0.64** | 42.02 ± 0.65 | **35.95 ± 0.03** | 33.62 ± 0.13 | 33.61 ± 0.10 |
| `FB15K237` | 89.29 ± 0.40 | **90.83 ± 0.30** | 90.49 ± 0.32 | **82.28 ± 0.08** | 80.18 ± 0.29 | 80.32 ± 0.07 |
| **Average** | **68.53** | 67.72 | 67.80 | **76.26** | 76.03 | 76.15 |

### G.2. Domain: Molecule

In addition to general GNN baselines applicable across various graphs, we compare our GIT model to domain experts specifically designed for molecules, including KVPLM (Zeng et al., 2022), MoMu (Su et al., 2022), Galactica (Taylor et al., 2022), and the recent SOTA model, GIMLET (Zhao et al., 2023). The results are presented in Table 24. We find that the general model pretrained on large-scale graphs generally underperforms compared to these domain experts. However, after specialization, the specialized model surpasses 3 out of 4 domain experts on average and outperforms the best expert model, GIMLET, on 4 out of 8 datasets. This observation demonstrates that post-training the general model with a reasonable number of domain-specific instances can enable it to match or even surpass expert models designed for that domain. These results strongly support the effectiveness of task-trees in designing graph foundation models.

Table 24: Comparison between our GIT and domain experts of molecule graphs in zero-shot setting.

| | HIV | BBBP | BACE | TOXCAST | CYP450 | TOX21 | MUV | PCBA | **Avg.** |
|---|---|---|---|---|---|---|---|---|---|
| KVPLM* | 61.20 | 60.20 | 51.26 | 50.96 | 59.22 | 49.17 | 61.72 | 48.11 | 55.23 |
| MoMu* | 50.26 | 49.81 | 66.56 | 52.38 | 57.98 | 57.57 | 60.51 | 51.50 | 55.82 |
| Galactica-1.3B* | 33.85 | 53.94 | 56.48 | 51.23 | 46.86 | 49.46 | 57.15 | 52.02 | 50.12 |
| GIMLET* | **66.24** | **59.39** | **69.57** | **59.04** | **71.25** | **61.19** | **64.39** | **62.11** | **64.15** |
| GIT - G | 56.76 | 54.76 | 33.66 | 51.55 | 63.21 | 56.83 | 53.71 | 56.25 | 53.34 |
| GIT - S | **66.14** | **62.16** | **52.27** | **58.30** | **69.75** | **63.45** | **65.32** | **65.26** | **62.83** |

\* indicates the results from paper (Zhao et al., 2023).

## H. Ablation on Specialization (SFT)

**Ablation Study on SFT Data used for Specialization.** We also evaluate the impact of the SFT dataset used for specialization. The model's zero-shot performance is reported in Table 25, comparing the default SFT dataset PCBA with another SFT dataset, HIV. We find that the model performance with HIV as the SFT data is lower than with PCBA. We hypothesize that this is due to HIV having fewer graphs and tasks, which may provide less information for reducing the distribution discrepancy. Nevertheless, HIV still improves the model's performance over the general model on 5 out of 8 datasets.

Table 25: The impact of SFT datasets in zero-shot setting.

| **SFT Data** | HIV | BBBP | BACE | TOXCAST | CYP450 | TOX21 | MUV | PCBA | **Avg.** |
|---|---|---|---|---|---|---|---|---|---|
| PCBA | 66.14 | 62.16 | 52.27 | 58.30 | 69.75 | 63.45 | 65.32 | 65.26 | 62.83 |
| HIV | 66.28 | 45.97 | 43.35 | 52.78 | 64.50 | 57.86 | 53.46 | 46.57 | 53.85 |

**Ablation Study on SFT Data used for Specialization.** We analyze the impact of SFT data in the experiments, as shown in Figure 9. The results show that changes in SFT data do not significantly affect model performance, particularly in fine-tuning and in-context learning settings. Even when the SFT and downstream data are the same, the model does not necessarily outperform models fine-tuned on other SFT datasets. This observation supports the motivation behind our proposed specialization method, which aims to shift the pretraining distribution $\mathcal{P}$ toward the distribution of target domains. It also highlights the importance of designing an instance selection method to identify the most effective SFT data.

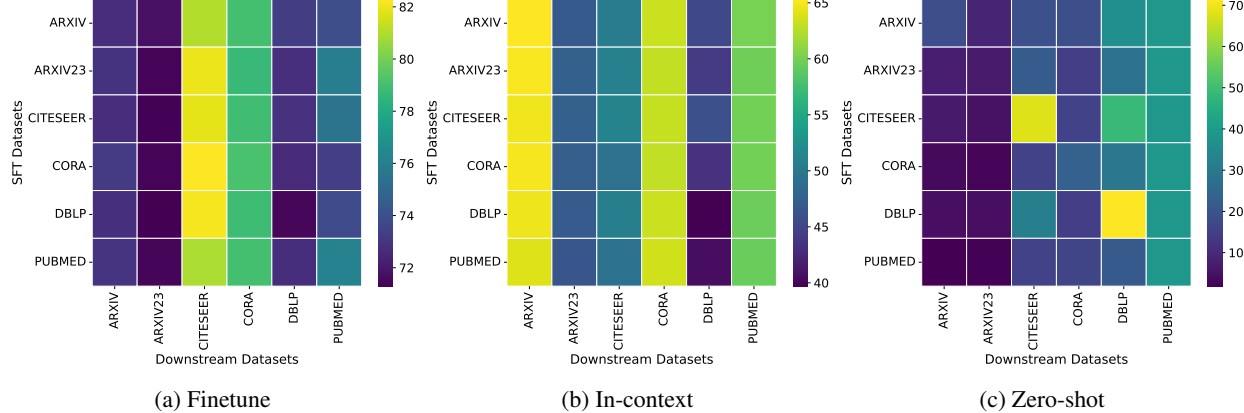

| (a) Finetune | (b) In-context | (c) Zero-shot |
|---|---|---|

Figure 9: The impact of different SFT datasets used for specialization in academic networks.

## I. Efficiency Analysis on Industry-scale Data

In order to further evaluate the efficiency of task-trees over subgraphs, we conduct experiments on a synthetic graph, termed as **ogbn-products x50**. This dataset is derived from the existing ogbn-products graph, which contains 316,513 nodes and 19,337,745 edges (we use a subset of original ogbn-products in our experiments). Specifically, we replicate the graph 50 times, reindex the nodes in each replicated graph, and combine them into a single dataset. This synthetic dataset contains 15,825,650 nodes (there are 111,059,956 nodes in ogbn-papers100M) and 966,887,250 edges (there are 1,615,685,872 edges in ogbn-papers100M). To ensure connectivity among these replicated graphs, we randomly flip 5% of the edges to connect the individual components, creating a more realistic large-scale graph structure.

We evaluated the time consumption per epoch on the ogbn-products x50 dataset and compared the efficiency of our proposed method, GIT-task-tree, against GIT-SubG (a variation that replaces task-trees with subgraphs). With a batch size of 512—selected to avoid out-of-memory issues in subgraph-based methods—the results are as follows:

Table 26: Efficiency comparison between subgraphs and task-trees in ogbn-products x50 (15,825,650 nodes and 966,887,250 edges).

|                          | GIT-SubG | GIT-Task-Tree |
| ------------------------ | -------- | ------------- |
| Sampling                 | 546s     | 546s          |
| Subgraph/Tree Extraction | 948s     | 0s            |
| Encoding                 | 5,717s   | 2,429s        |
| All                      | 7,211s   | 2,975s        |

These results underscore the significant efficiency advantages of our method. Notably, the efficiency gap between GIT-task-tree and GIT-SubG on this dataset is larger than what we observed during pretraining (i.e., 208 seconds per epoch for GIT-task-tree versus 280 seconds per epoch for GIT-SubG). We attribute this to the substantial overlap between subgraphs in this dataset, which increases the space and time required for processing.

## J. General Reasoning Capability of Specialized Models

We further analyze the performance of specialized models on general reasoning tasks beyond their specific domains. We assess the model's performance on other domains as a measure of its general reasoning ability. For example, if a model is specialized for academic networks, its general reasoning capability refers to its performance on graphs from other domains, such as e-commerce networks, knowledge graphs, and molecular graphs. The results are presented in Table 27. We report in-context learning performance rather than basic fine-tuning due to computational efficiency. Additionally, we include the performance of the pretrained general model without specialization as a baseline. If the specialized model performs worse than the general model, it suggests that specialization may diminish GIT's general reasoning capability. From the table, it is clear that while specialized models excel in their specific domains, they struggle in other domains. This degradation of general inference capability, often referred to as the *specialization tax*, is a common challenge in building specialized large language models. The specialization tax can limit the model's practicality in scenarios requiring both domain-specific knowledge and the ability to handle general tasks. Thus, balancing domain-specific performance with maintaining general reasoning capability is an important research direction.

Table 27: The in-context learning performance of GIT with different specialization datasets (SFT Data) on four domains. The results of each domain is the average of all datasets within the domain.

|                       | SFT Data          | Academia | E-commerce | KG        | Molecule  |
| --------------------- | ----------------- | -------- | ---------- | --------- | --------- |
| **General Model**     | -                 | 54.00 | 57.22 | 67.55 | 55.96 |
|                       | Arxiv (academia)  | **55.18** | 57.63     | 66.80     | 54.42     |
| **Specialized Model** | Products (E-com)  | 50.09    | **58.01**  | 65.06     | 55.75     |
|                       | FB15K237 (KG)     | 54.70    | 56.57      | **67.80** | 55.37     |
|                       | PCBA (Mol)        | 50.49    | 56.87      | 61.49     | **62.82** |

## K. More Parameters Enhance Model Performance

We present comprehensive results of general GIT with different hidden dimensions in Figure 10. For computational efficiency, we does not report results on datasets needing intensive computing resources. We observe that increasing the number of model parameters consistently improves performance across both basic fine-tuning and in-context learning settings. Notably, the performance improvement is more pronounced in in-context learning as the model size increases. This may be because, in in-context learning, the model is not fine-tuned on downstream tasks, making the knowledge retained in the original model more crucial. Larger hidden dimensions allow the model to preserve more knowledge. However, when the model is fine-tuned on downstream tasks, the pretraining knowledge is adapted to the specific task, reducing the reliance on the original model's knowledge and leading to a relatively smaller performance gap.

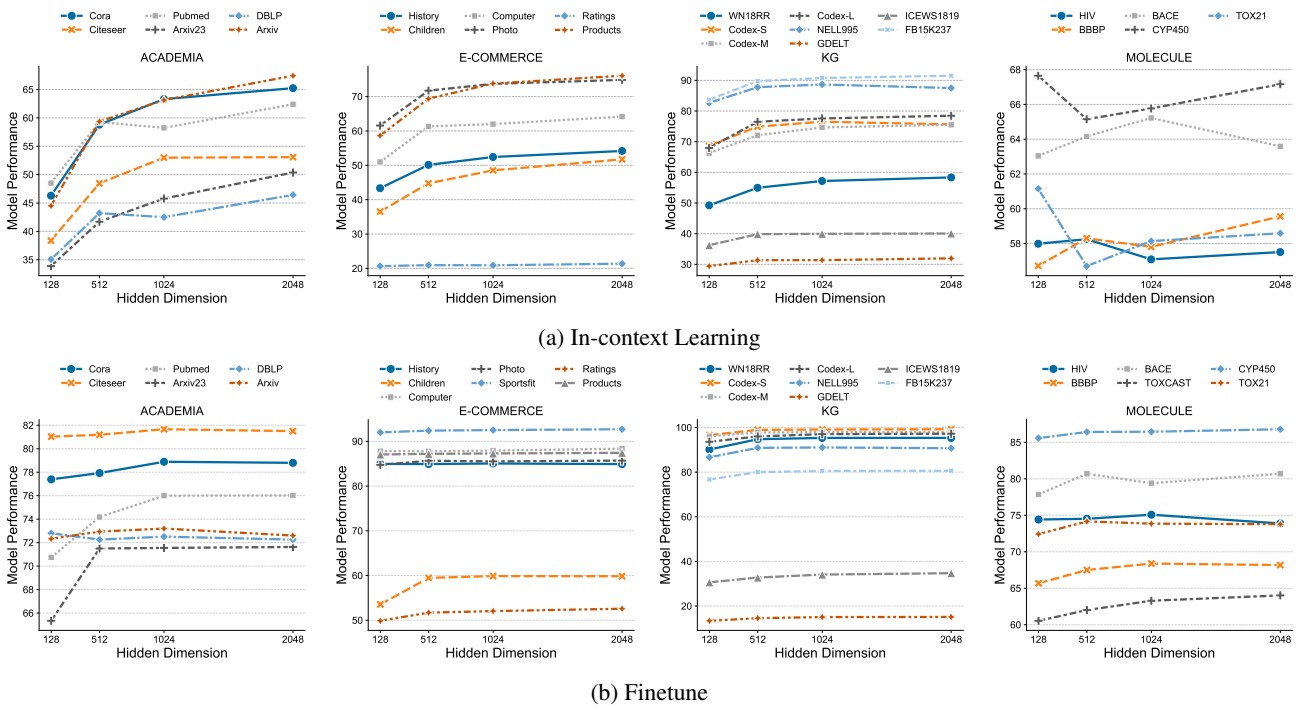

Figure 10: The comprehensive results of the impact of hidden dimensions on model performance.

## L. Comparison to Few-Shot Learning Methods

We compare the performance of GIT with methods designed for few-shot learning on Arxiv in Table 28. We include a wide range of baselines: GPN (Ding et al., 2020) is a framework that leverages GNN and meta-learning to address few-shot node classification by learning a transferable metric space. TENT (Wang et al., 2022b) introduces three levels of adaptation—node-level, class-level, and task-level—to mitigate task variance and improve the model's generalization performance across different meta-tasks. GLITTER (Wang et al., 2022a) enhances few-shot node classification by learning task-specific graph structures for each meta-task using node influence and mutual information. TLP (Tan et al., 2022) transfers pretrained node embeddings fine-tunes a simple linear classifier on novel classes. Prodigy (Huang et al., 2023) enables in-context learning over graphs by designing graph prompt learning template. It is important to note that we implement an in-context version of few-shot learning without fine-tuning GIT on the few-shot task, whereas the other methods require fine-tuning. Despite this, GIT achieves the second-best performance across all settings, outperforming 5 out of 6 few-shot learning methods. This highlights GIT's strong potential in scenarios with limited instances and labels.

Table 28: The few-shot performance on `Arxiv`, comparing to few-shot learning methods.

| | 5-way | | | 3-way | | |
| --- | --- | --- | --- | --- | --- | --- |
| | 5-shot | 3-shot | 1-shot | 5-shot | 3-shot | 1-shot |
| GPN | 50.53 ± 3.07 | 48.32 ± 3.80 | 38.58 ± 1.61 | 62.25 ± 4.94 | 58.52 ± 3.00 | 48.45 ± 5.60 |
| TENT | 60.83 ± 7.45 | 56.03 ± 8.90 | 45.62 ± 10.70 | 74.20 ± 9.93 | 70.48 ± 11.50 | 59.38 ± 13.55 |
| GLITTER | 56.00 ± 4.40 | 57.44 ± 4.90 | 47.12 ± 2.73 | 62.13 ± 10.85 | 60.93 ± 12.12 | 59.20 ± 5.48 |
| TLP-BGRL | 50.13 ± 8.78 | 46.21 ± 7.92 | 35.81 ± 8.58 | 62.93 ± 11.74 | 58.37 ± 11.34 | 46.30 ± 10.83 |
| TLP-SURGL | 77.89 ± 6.46 | 74.19 ± 7.55 | 61.75 ± 10.07 | 86.27 ± 7.54 | 83.75 ± 8.86 | 73.46 ± 12.68 |
| Prodigy | 61.09 ± 5.85 | 58.64 ± 5.84 | 48.23 ± 6.18 | 73.64 ± 6.93 | 71.43 ± 7.28 | 61.59 ± 8.53 |
| GIT - G | 70.50 ± 0.47 | **64.07 ± 0.50** | 49.18 ± 0.56 | 80.20 ± 0.67 | 74.65 ± 0.54 | 61.93 ± 0.18 |
| GIT - S | **70.70 ± 0.28** | 64.06 ± 0.58 | **50.94 ± 0.57** | **80.51 ± 0.68** | **76.05 ± 0.53** | **63.42 ± 0.46** |

