# OpenReview forum: "Towards Graph Foundation Models: Learning Generalities Across Graphs via Task-Trees"
_ICML.cc/2025/Conference — ICML 2025 poster_

### Official Review · Reviewer_xh4d · 2025-03-11

**Overall Recommendation:** 3

**Summary:**

This paper introduces an approach to enhance the generalization of GNNs across diverse tasks, which typically vary in their inductive biases, such as node classification, link prediction, and graph classification. The authors propose the concept of Task-Trees, a framework designed to align task spaces at different levels (node, edge, and graph) by introducing virtual task nodes that connect task-related components.  The paper presents Graph Generality Identifier on Task-Trees (GIT), a pre-trained model that leverages Task-Trees to learn generalized representations. Theoretical analysis confirms the framework's stability, transferability, and generalization potential. Extensive experiments demonstrate the effectiveness of GIT in fine-tuning, in-context learning, and zero-shot learning tasks, showcasing superior generalization and computational efficiency compared to subgraph-based methods.

## update after rebuttal
I have improved the score according to the rebuttal.

**Claims And Evidence:**

Most claims are supported by clear and convincing evidence.

**Essential References Not Discussed:**

Some research on graph foundation models are not discussed, e.g., All in One and One for All: A Simple yet Effective Method towards Cross-domain Graph Pretraining published in SIGKDD 2024.

**Experimental Designs Or Analyses:**

This paper provides abundant experiments on diverse downstream tasks, which is a lightspot of the paper. However, most datasets are homophilc, I advise the authors to conduct some experiments on heterophilc graphs.

**Methods And Evaluation Criteria:**

The proposed method in this paper   bridges the differences between tasks.

**Other Comments Or Suggestions:**

One limitation of this work is that it does not account for the reliability of information across different domains. We observe that most experiments are conducted on homophily graphs, with little consideration of graph noise. It would be beneficial to include additional experiments on heterophily graphs to enhance the analysis.

**Other Strengths And Weaknesses:**

By leveraging a tree-like structure, the approach transfers different types of tasks to a task node, thereby enhancing its generalization capability. However, the paper contains numerous unclear descriptions regarding both the methodology and experiments, revealing certain shortcomings in academic writing.

**Questions For Authors:**

1. Some diagrams appear to be inconsistent. For example, in Figure 1, the task-tree corresponding to the graph-level task should ideally connect all nodes. However, in the first layer, only three nodes are connected. It may be helpful to add some textual clarification to improve understanding.
2. Another question is based on your selected datasets. Why all experiments are conducted on TAGs? Does it mean that your method relies on text attributes?
3. How is the performance when the number of shots increase to 10,100, or more? Provided experiments only focus on few-shot scenarios.
4. See the experiment part.
5. See the theory part.

**Relation To Broader Scientific Literature:**

The key contributions of this paper build upon prior work in graph representation learning, graph foundation models, and multi-task learning, providing a perspective on generalization across graph-based tasks. Existing approaches often use graphon-based methods or contrastive learning, but they may struggle to align tasks effectively. Task-Trees bridge this gap by introducing a unified structure for task representation, enabling the pre-trained GIT model to transfer knowledge more effectively across datasets. This aligns with broader trends in task-agnostic pretraining, similar to foundation models in NLP and vision. Additionally, the work contributes to multi-task learning and representation alignment, where prior approaches in graphs have mainly relied on adversarial training or knowledge distillation. By introducing a tree-based structure to connect different task levels, this paper offers a fresh approach to improving the transferability of graph models, advancing the study of generalization in graph learning.

**Theoretical Claims:**

1. The definition of the Task-Tree Generality Assumption here is quite imprecise. What specific generalities shared across graphs should be included?
2. I am somewhat concerned about the generalization bound proven in Section 3.5. Its form is similar to the generalization bounds in PAC learning, but for models like GNNs, which deal with non-i.i.d. data, such bounds may not be directly applicable.
3. In Theorem 3.3, it seems that the domain shift issue in downstream tasks is overlooked. If the distribution gap between the pretraining and fine-tuning tasks is too large, the bound may completely fail.

---

> ### Author Rebuttal · Authors · 2025-04-01
>
> We sincerely thank the reviewer for their thorough and thoughtful feedback. We appreciate the recognition of our contributions and also value the constructive suggestions. We address each point in detail below.
>
> > Theory
> >
>
> **Q1: Imprecise Task-Tree Generality Assumption**
>
> **A1:** The generalities means the common substructures shared across graphs, as tree structure naturally serve as an approximation of substructure patterns. However, discovering exact common substructures is intractable due to the variety of graph data. We will revise the assumption for greater clarity.
>
> **Q2: Concerns about generalization bound**
>
> **A2:** It is true that graph data at the node level can violate the i.i.d. assumption due to inter-node dependencies. However, in our formulation, each **task-tree** is treated as an independent training instance. We assume that task-trees are **i.i.d. sampled** from a uniform distribution over the tree space, which makes the PAC-style generalization analysis applicable in our setting. We will explicitly add this assumption in the revised manuscript for clarity.
>
> **Q3: Domain shift in Theorem 3.3**
>
> **A3**: Theorem 3.3 are derived under the standard assumption that the pretraining (source) and downstream (target) task distributions are not completely disjoint. This reflects a core principle of transfer learning—there must be some **semantic alignment** between domains. Without such overlap, **negative transfer** can occur, where a pretrained model may underperform compared to one trained from scratch. We agree that in cases of **severe domain shift**, the bound may become loose or even vacuous. This is a known limitation of theoretical transfer bounds, which inherently depend on distributional similarity. We will revise the discussion to explicitly state this assumption and implications.
>
> > More Related Works
> >
>
> We have included the discussion: GCOPE addresses challenges in cross-domain graph pretraining by applying SVD for feature alignment and introducing virtual coordinator nodes to unify graph structures. It is designed for handling feature heterogeneity across graphs on node classification, whereas our method focuses on aligning different tasks (handling task heterogeneity) on graphs.
>
> We run experiments to compare GCOPE and GIT-G on academic datasets. Both models are pretrained on Arxiv, Arxiv-23, and DBLP, and finetuned on Cora, Citeseer, and Pubmed. GIT-G achieves better performance on two out of three datasets.
>
> |  | Cora | Citeseer | Pubmed |
> | --- | --- | --- | --- |
> | GCOPE | 76.39 | 78.75 | **75.39** |
> | GIT-G | **77.84** | **80.87** | 74.80 |
>
> > Why use TAGs in experiments?
> >
>
> Our method does **not** rely on TAGs. We use TAGs because they allow node features to be aligned through a textual encoder. Since our primary contribution is a framework for addressing **task heterogeneity**, TAGs help us isolate this aspect by minimizing the confounding effects of **feature heterogeneity**. This design choice allows us to more clearly demonstrate the effectiveness of task-trees in aligning tasks across different levels.
>
> Importantly, our method is fully applicable to **non-textual graphs** as well. To show this, we introduce a simple SVD-based module to align features across graphs. We pretrain the model on purely non-textual datasets—PubMed (node classification), Citeseer (link prediction), and IMDB-B (graph classification)—and finetune on Cora (link prediction), as shown below.
>
> | Setting | GraphMAE | GIT-G |
> | --- | --- | --- |
> | w/o SVD + w/o pretrain | **95.02** | 94.27 |
> | w. SVD + w/o pretrain | 94.89 | **95.50** |
> | **w. SVD + w. pretrain** | 95.10 | **95.70** |
>
> > Additional Experiments
> >
>
> **Q1: Heterophily Graphs**
>
> **A1:** We have already included heterophily graphs in our experiments (Table 15). Specifically, the **Children** and **Ratings** datasets in the e-commerce domain have homophily ratios of **0.42** and **0.38**, respectively—lower than other datasets (greater than 0.60). GIT-G outperforms baselines on these heterophily graphs.
>
> |  | GraphMAE | OFA | GIT-G |
> | --- | --- | --- | --- |
> | Children | 56.76 | 55.43 | **59.09** |
> | Ratings | 52.39 | 51.79 | **52.45** |
>
> **Q2: Few-shot Learning**
>
> **A2:** We evaluate GIT-G in few-shot learning settings within the academic domain, averaged over six graphs. We observe that increasing the number of shots significantly improves performance. GIT-G consistently outperforms GraphMAE across all settings, likely due to its ability to better transfer knowledge across domains via task-trees.
>
> |  | GraphMAE | GIT-G |
> | --- | --- | --- |
> | *5-way* |  |  |
> | 10-shot | 45.27 | **49.61** |
> | 50-shot | 53.49 | **55.60** |
> | 100-shot | 59.73 | **60.13** |
> | *10-way* |  |  |
> | 10-shot | 37.73 | **40.09** |
> | 50-shot | 48.04 | **50.26** |
> | 100-shot | 53.02 | **56.09** |
>
> > Visualization: Node Count
> >
>
> We have revised the figure to indicate that more nodes exist, using dotted edges to clarify that the figure is a partial illustration.

---

### Official Review · Reviewer_sWNJ · 2025-03-12

**Overall Recommendation:** 3

**Summary:**

The authors propose a various task alignment method based on task trees.

**Claims And Evidence:**

See weaknesses part.

**Essential References Not Discussed:**

Other graph foundation models [1,2,3,6] should be discussed or compared with.

The idea of using subgraph as the basic instances is proposed in previous works [1,2]，which can also be applied to solve various tasks. These methods should be discussed and compared with.

[1] Xia et al. Anygraph: Graph foundation model in the wild. arXiv 2024.\
[2] Xia et al. Opengraph: Towards open graph foundation models. arXiv 2024.\
[3] Yu et al. SAMGPT: Text-free Graph Foundation Model for Multi-domain Pre-training and Cross-domain Adaptation. WWW 2025.\
[4] Liu et al. Graphprompt: Unifying pre-training and downstream tasks for graph neural networks. WWW 2023.\
[5] Yu et al. Generalized graph prompt: Toward a unification of pre-training and downstream tasks on graphs. IEEE Transactions on Knowledge and Data Engineering, 2024.

**Experimental Designs Or Analyses:**

I check the experimental setting.

**Methods And Evaluation Criteria:**

The evaluation criteria makes sense for the problem.

**Other Comments Or Suggestions:**

See weaknesses part.

**Other Strengths And Weaknesses:**

Strength:
1. The authors conduct extensive experiments.
2. The paper is overall well written.

Weaknesses:\
1.This paper is far from graph foundation models. A key challenge in developing such models is bridging the feature and structure gaps across domains. However, this work primarily proposes a unified instance for various tasks, leaving these challenges unresolved.\
2. Although the authors discuss the differences between using task trees and subgraphs as unified instances, the contribution of introducing task trees is limited, making the work lack novelty.

**Questions For Authors:**

See weaknesses part.

**Relation To Broader Scientific Literature:**

See weaknesses part.

**Theoretical Claims:**

I check the correctness of theoretical claims.

---

> ### Author Rebuttal · Authors · 2025-04-01
>
> We thank the reviewer for acknowledging the soundness of our theoretical claims, evaluation criteria, and experimental setup. We address each point in detail below and are committed to improving clarity and rigor in future revisions.
>
> > More Related Works and Comparative Results
> >
>
> We thank the reviewer for this valuable comment. In response, we have expanded both the discussion and experiments to include several recent and relevant graph foundation models. Below is the updated discussion:
>
> **GraphPrompt** [6] is the first graph prompt learning method that aligns graph tasks to subgraph-level tasks by extracting subgraphs for each graph instance. **GraphPrompt+** [1] builds upon this by incorporating additional pretraining tasks and a more generalizable prompt design, enhancing the model’s ability to capture hierarchical information. **All in One** [2] further advances this line of work by introducing a learnable graph template and meta-learning techniques to better align pretraining with downstream tasks. **OpenGraph** [3] proposes a unified graph tokenizer and a scalable transformer to address token and structural differences across graphs, also leveraging LLM-based knowledge distillation to mitigate data scarcity. **AnyGraph** [4] extends OpenGraph by incorporating a Mixture-of-Experts (MoE) mechanism to handle feature and structure heterogeneity, using SVD for node feature alignment. Lastly, **SAMGPT** [5] introduces a prompt-based approach designed specifically for non-textual graphs, aligning both node semantics and structural knowledge. While most of these models aim to address multiple challenges in building graph foundation models, **GIT** focuses specifically on task alignment, similar in spirit to **GraphPrompt**, but with a novel formulation using **task-trees**.
>
> To demonstrate the benefits of GIT, we conducted experiments comparing it against **GraphPrompt+** [1], **All in One** [2], **OpenGraph** [3], and **AnyGraph** [4]. We exclude **GraphPrompt** [6] as it has been superseded by GraphPrompt+, and **SAMGPT** [5] due to its focus on non-textual graphs and the unavailability of official implementation. Nonetheless, we include both in the extended discussion.
>
> We evaluate **GIT-G** and baselines across three domains—academic (node classification), knowledge graphs (edge classification), and molecules (graph classification)—under the pretrain-then-finetune setting. As shown below, **GIT-G** consistently outperforms all recent baselines.
>
> | **Domain** | **Task** | **GraphPrompt+ [1]** | **All in one [2]** | **OpenGraph [3]** | **AnyGraph [4]** | **GIT - G (ours)** |
> | --- | --- | --- | --- | --- | --- | --- |
> | **Academic -  (avg. over 6 graphs)** | Node Classification | 74.80 | 75.25 | 74.64 | 75.01 | **75.82** |
> | **KG -  (avg. over 8 graphs)** | Edge Classification | 74.78 | 74.92 | 71.38 | 74.30 | **75.73** |
> | **Molecule (avg. over 8 graphs)** | Graph Classification | 72.99 | 71.87 | 72.84 | 72.49 | **74.57** |
>
> [1] Generalized Graph Prompt: Toward a Unification of Pre-Training and Downstream Tasks on Graphs, TKDE 24.
>
> [2] All in One: Multi-task Prompting for Graph Neural Networks, KDD 23
>
> [3] OpenGraph: Towards Open Graph Foundation Models, EMNLP 24
>
> [4] AnyGraph: AnyGraph: Graph Foundation Model in the Wild, Arxiv 24
>
> [5] SAMGPT: Text-free Graph Foundation Model for Multi-domain Pre-training and Cross-domain Adaptation, WWW 25
>
> [6] Graphprompt: Unifying pre-training and downstream tasks for graph neural networks, WWW 23
>
> > This paper is far from graph foundation models.
> >
>
> We appreciate this important point. Actually, we apply two light-weight approaches to solve these two challenges. We address feature heterogeneity by using text-attributed graphs, where textual descriptions are encoded via a text encoder to align node features. For structural heterogeneity, we design a regularizer (see Appendix B.2) to encourage structural consistency across graphs.
>
> That said, we would like to emphasize that the primary focus of our work is on addressing task heterogeneity, rather than simultaneously tackling all challenges of graph foundation models. By focusing on this aspect, we are able to clearly isolate and demonstrate the effectiveness of task-trees for aligning node-, edge-, and graph-level tasks. However, for better readability, we will revise some statements to make the expression more precise.
>
> > The contribution of introducing task trees is limited.
> >
>
> Thank you for raising this concern. We would like to clarify that our major contribution lies in the **theoretical foundation** we provide for task-trees. To the best of our knowledge, this is the first work that formally derives theoretical results on handling **task heterogeneity** across graphs. Our analysis includes stability guarantees, transferability, and generalization bounds, providing a principled foundation for this new formulation. We believe this significantly advances the understanding of task alignment in graph learning.

---

> > ### Comment · Reviewer_sWNJ · 2025-04-05
> >
> > Thank you for your response, which has solved my concerns. I hope that the future version will include the above discussion and comparison with the related work. Consequently, I have improved my score to 3.

---

> > > ### Author Response · Authors · 2025-04-06
> > >
> > > Thank you for your response and for raising the score. We will include the discussion and comparison in our future work. Your feedback is genuinely appreciated and helps us continue to improve our work.

---

### Official Review · Reviewer_g6zE · 2025-03-14

**Overall Recommendation:** 4

**Summary:**

The paper proposes a graph foundation model called GIT for multiple graph learning tasks across various domains. GIT introduces task-trees as basic learning instances to align task spaces (node, link, graph) on graphs, acquire transferable knowledge, and effective adaptation to downstream tasks.  A series of theoretical analyses is provided to demonstrate the stability, transferability, and generalization of GIT. Comprehensive experiments over 30 graphs in five domains are conducted to demonstrate the effectiveness of GIT for multiple graph learning tasks.

**Claims And Evidence:**

Most claims and evidence are well provided. The proposed idea is interesting, the theoretical analysis is sufficient. The experiments are solid and comprehensive.

Question/issue: The authors claim efficiency of task-tree based model compared to subgraph-based methods. How does this work? It is better to provide any complexity analysis and experiment to support this claim.

**Essential References Not Discussed:**

N/A

**Experimental Designs Or Analyses:**

The experiments and analyses are comprehensive. The authors conduct experiments over 30 datasets across five domains. The results are sufficient and significant.
Question/issue: The model GIT-S includes a supervised fine-tuning step and outperforms all models. For a fair comparison, it is necessary to compare the results of the baseline methods with the same step.

**Methods And Evaluation Criteria:**

Make sense.

**Other Comments Or Suggestions:**

N/A

**Other Strengths And Weaknesses:**

The paper writing and presentation are good.

**Questions For Authors:**

Questions are included in the above comments.

**Relation To Broader Scientific Literature:**

This paper develops a graph foundation model that can benefit various graph learning studies and applications.

**Theoretical Claims:**

The theoretical claims to demonstrate the stability, transferability, and generalization of GIT are sufficient. The authors have provided detailed proofs in the appendix.

---

> ### Author Rebuttal · Authors · 2025-04-01
>
> We sincerely thank the reviewer for the positive and encouraging feedback. We appreciate the recognition of our contributions in theory, methodology, and experimentation, as well as the thoughtful questions regarding model efficiency and fair comparisons—we have addressed these points in detail below.
>
> > The authors claim efficiency of task-tree based model compared to subgraph-based methods. How does this work? It is better to provide any complexity analysis and experiment to support this claim.
> >
>
> We appreciate the reviewer’s comment and the opportunity to clarify. Both subgraph-based and task-tree-based methods require GNNs for encoding. However, the computational bottleneck in subgraph-based approaches lies in the explicit extraction of subgraphs. Assuming a graph with $n$ nodes, extracting subgraphs using adjacency matrix-based BFS incurs a time complexity of $O(n^3)$. In contrast, our task-tree-based approach augments the original graph by appending virtual nodes, avoiding subgraph extraction. The time complexity of this augmentation is linear with respect to the number of nodes, edges, or graphs, depending on the task, making it efficient in practice.
>
> Empirically, we have provided a runtime and memory comparison in **Figure 4**. Specifically, we implemented **GIT-SubG**, a variant of our model that replaces task-trees with subgraphs. Below is the comparison of **time per epoch** and **GPU memory usage** during pretraining (on ~1.7 million instances) between **GIT-Task-Tree** and **GIT-SubG**. For reference, **GraphMAE**, using a batch size of 2048, requires 193 seconds per epoch and 35% memory allocation on a 48GB GPU.
>
> | **Batch Size** | **512** | **1024** | **2048** | **4096** | **8192** | **16384** | **32768** |
> | --- | --- | --- | --- | --- | --- | --- | --- |
> | *GIT - Task-Tree* |  |  |  |  |  |  |  |
> | **Time / Epoch (s)** | 208 | 180 | 176 | 172 | 172 | 163 | 162 |
> | **Memory Allocation (%)** | 6 | 8 | 18 | 41 | 75 | 93 | 98 |
> | *GIT - SubG* |  |  |  |  |  |  |  |
> | **Time / Epoch (s)** | 280 | 243 | 234 | 223 | OOM | OOM | OOM |
> | **Memory Allocation (%)** | 21 | 39 | 74 | 97 | OOM | OOM | OOM |
>
> > The model GIT-S includes a supervised fine-tuning step and outperforms all models. For a fair comparison, it is necessary to compare the results of the baseline methods with the same step.
> >
>
> Thank you for this insightful comment. We agree and have already included the fair comparison in **Table 3** of the paper. Below is a summary of the results on **academic networks**, showing performance across three settings: 0-shot, 3-shot, and finetuning. As shown, specialization through fine-tuning improves performance for all models. Notably, **GIT** outperforms both **GraphMAE** and **OFA** in both general and specialized forms across all evaluation settings.
>
> | **Method** | **0-shot** | **3-shot** | **Finetune** |
> | --- | --- | --- | --- |
> | *General Model* |  |  |  |
> | **GraphMAE-G** | 15.42 | 49.25 | 73.81 |
> | **OFA-G** | 13.98 | 45.93 | 72.18 |
> | **GIT-G** | **14.88** | **54.00** | **75.82** |
> | *Specialized Model* |  |  |  |
> | **GraphMAE-S** | 20.31 | 51.21 | 74.05 |
> | **OFA-S** | 20.05 | 46.87 | 73.04 |
> | **GIT-S** | **23.45** | **55.18** | **75.88** |

---

### Official Review · Reviewer_Bjq8 · 2025-03-14

**Overall Recommendation:** 3

**Summary:**

This paper introduces a novel approach for learning generalities across graphs via task-trees, which unify node-, edge-, and graph-level tasks by introducing virtual task nodes. The theoretical analysis demonstrates the stability, transferability, and generalization properties of task-trees. Empirically, the proposed pretrained model GIT achieves strong performance across 32 graphs from 5 domains via fine-tuning, in-context learning, and zero-shot learning. Specialization through instruction tuning further enhances domain-specific performance.

**Claims And Evidence:**

The key claims are supported by experiments and theory, but issue remains:

Task-Tree Generality: The assumption that task-trees capture cross-graph generalities is validated only on text-attributed graphs. Non-textual graphs  are not tested.

**Essential References Not Discussed:**

The references are generally comprehensive.

**Experimental Designs Or Analyses:**

Baselines: The baselines is outdated. OFA[1] is published in 2023. Recent graph foundation models are missing.
[1]Liu, Hao, et al. "One for all: Towards training one graph model for all classification tasks." arXiv preprint arXiv:2310.00149 (2023).

**Methods And Evaluation Criteria:**

The proposed methods and evaluation criteria generally make sense.

**Other Comments Or Suggestions:**

1. It is recommended that the authors include a comparison between their method and node-of-interest or graph prompt approaches, as well as add relevant baselines for evaluation.
2. It is recommended to conduct ablation studies on the choice of text encoder.

**Other Strengths And Weaknesses:**

Although the authors propose task-trees, they do not differentiate this method from node-of-interest[1] or graph prompt[2] approaches.
[1]Liu, Hao, et al. "One for all: Towards training one graph model for all classification tasks." arXiv preprint arXiv:2310.00149 (2023).
[2]Sun, Xiangguo, et al. "All in one: Multi-task prompting for graph neural networks." Proceedings of the 29th ACM SIGKDD Conference on Knowledge Discovery and Data Mining. 2023.

**Questions For Authors:**

1. Does your method rely on a text encoder?
2. Can your method be transferred to non-text-attributed graphs (non-TAGs)?

**Relation To Broader Scientific Literature:**

The key contributions of this paper are closely tied to the literature on Graph Foundation Models (GFMs) and cross-task alignment:

Graph Foundation Models: Prior works like OFA (Liu et al., 2024) align tasks using subgraphs as basic units. This paper proposes task-trees as a more efficient and learnable alternative, directly addressing the computational overhead and limited expressiveness of subgraph-based methods.

Task Heterogeneity: Earlier studies (e.g., Sun et al., 2023) align tasks through reformulation (e.g., converting node classification to subgraph classification). However, this work is the first to theoretically justify the superiority of task-trees in terms of stability and transferability.

Instruction Tuning: Inspired by instruction tuning in large language models (e.g., LLaMA), the authors extend this paradigm to graphs, complementing graph in-context learning frameworks like Prodigy (Huang et al., 2023).

The novelty of this work lies in:

Unified Task Representation: Unlike subgraphs or graphon-based methods, task-trees unify node-, edge-, and graph-level tasks into a tree structure via virtual nodes, achieving structural alignment across tasks for the first time.

Theoretical Grounding: Existing GFMs (e.g., GraphMAE, BGRL) lack theoretical explanations for task heterogeneity. This paper fills the gap with stability theorems (Theorem 3.1) and generalization bounds (Theorem 3.5).

**Theoretical Claims:**

The theoretical claims are generally correct.

---

> ### Author Rebuttal · Authors · 2025-04-01
>
> We sincerely thank the reviewer for the thoughtful and constructive feedback. We are encouraged by the positive recognition of our contributions, including the unified task-tree representation, theoretical grounding, and strong empirical performance across diverse settings. In the following, we address each comment point-by-point.
>
> > More comparing results
> >
>
> We thank the reviewer for this valuable comment. In response, we have updated our experiments to include several recent and relevant graph foundation models, including **GraphPrompt+** [1], **All in One** [2], **OpenGraph** [3], **AnyGraph** [4]. We compare our **GIT-G** to these method across three domains, including academic (node classification), knowledge graph (edge classification), and molecules (graph classification), in the pretrain-then-finetune setting, where our GIT-G consistently outperforms these recent baselines. The results are shown in the following table.
>
> | **Domain** | **Task** | **GraphPrompt+ [1]** | **All in one [2]** | **OpenGraph [3]** | **AnyGraph [4]** | **GIT - G (ours)** |
> | --- | --- | --- | --- | --- | --- | --- |
> | **Academic -  (avg. over 6 graphs)** | Node Classification | 74.80 | 75.25 | 74.64 | 75.01 | **75.82** |
> | **KG -  (avg. over 8 graphs)** | Edge Classification | 74.78 | 74.92 | 71.38 | 74.30 | **75.73** |
> | **Molecule (avg. over 8 graphs)** | Graph Classification | 72.99 | 71.87 | 72.84 | 72.49 | **74.57** |
>
> [1] Generalized Graph Prompt: Toward a Unification of Pre-Training and Downstream Tasks on Graphs, TKDE 24.
>
> [2] All in One: Multi-task Prompting for Graph Neural Networks, KDD 23
>
> [3] OpenGraph: Towards Open Graph Foundation Models, EMNLP 24
>
> [4] AnyGraph: AnyGraph: Graph Foundation Model in the Wild, Arxiv 24
>
> > Discussion on node-of-interest and graph prompt
> >
>
> We appreciate the reviewer’s comment and the opportunity to clarify the distinction. As discussed in Section 2.3 of our paper, our proposed method is fundamentally different from node-of-interest (NOI) and graph prompt-based approaches, such as **All in One** [2] and **OFA** [1], which rely on subgraph extraction paradigms.
>
> Specifically, **All in One** constructs task-specific subgraphs (e.g., ego-graphs centered on task-relevant nodes) and then applies GNNs to these subgraphs. Similarly, **OFA** formalizes the concept of NOI to further unify various tasks, but it ultimately follows the same principle—extracting and operating on subgraphs derived from nodes of interest.
>
> In contrast, our method introduces **task-trees**, which are more efficient and learnable structures that augment the original graph rather than extract subgraphs. Task-trees provide an efficient and learnable way of encoding task semantics. This difference is empirically validated in our paper (Table 4 and Figure 4), and further supported by the updated results shown above, where **GIT-G** consistently outperforms subgraph-based baselines.
>
> > Questions about the textual encoder.
> >
>
> **Q1: Does the method rely on text encoder?**
>
> **A1**: No, the method does not necessarily rely on a text encoder. We use text-attributed graphs in our experiments to focus solely on our main contribution—handling ***task heterogeneity***—and avoid introducing additional components for addressing *feature heterogeneity*. Textual attributes allow node features across graphs to be aligned using a textual encoder, helping us isolate and demonstrate the effectiveness of task-tree generalization.
>
> **Q2: Can the method be applied to non-text-attributed graphs?**
>
> **A2**: Yes, our method can be applied to non-text-attributed graphs. To demonstrate this, we introduced a simple module to handle feature heterogeneity by applying SVD to align features into a shared space. We pretrain on pure non-textual graphs—PubMed (node classification), Citeseer (link prediction), and IMDB-B (graph classification)—and finetune on Cora (link prediction). The results, reported below using AUC as metric, show that **GIT-G** remains effective without textual attributes.
>
> | **Setting** | **GraphMAE** | **GIT-G** |
> | --- | --- | --- |
> | w/o SVD + w/o pretrain | **95.02** | 94.27 |
> | w. SVD + w/o pretrain | 94.89 | **95.50** |
> | **w. SVD + w. pretrain** | 95.10 | **95.70** |
>
> **Q3: The task-tree generality assumption is only validated on text-attributed graphs.**
>
> **A3:** As shown in the discussion and results above, our method is generalizable to both text-attributed and non-text-attributed graphs. Therefore, the task-tree generality assumption does not depend on the presence of text information.
>
> > The ablation on textual encoder
> >
>
> We further evaluate the impact of different textual encoders on model performance, including **MiniLM**, **MPNet**, and **SentenceBERT** (our default choice), on the academic domain in the finetuning setting.
>
> | **Domain** | **MiniLM** | **MPNet** | **SentenceBERT (default)** |
> | --- | --- | --- | --- |
> | Academic | 75.42 | 75.75 | **75.82** |

---

### Decision · Program_Chairs · 2025-05-01

**Decision:**

Accept (poster)

**Comment:**

This paper investigates the problem of identifying generalities across graph-structured data. To address this challenge, the authors propose a novel approach for learning cross-task generalities in graphs. Specifically, they introduce task-trees as basic learning instances to align task spaces—including node, edge, and graph-level tasks—within a unified structural framework. To empirically validate their method, the authors develop a pretrained graph model based on task-trees. Experimental results on several datasets demonstrate the effectiveness of the proposed model.


Strengths:
1. The paper’s key contributions are well-aligned with recent advancements in Graph Foundation Models (GFMs) and the challenge of cross-task alignment.
2. The proposed task-tree structure effectively unifies node-, edge-, and graph-level tasks by introducing virtual nodes, enabling structural alignment across different learning objectives.
3. The theoretical analysis supporting the stability, transferability, and generalization capabilities of GIT is sound and well-motivated.



Weaknesses:

1. The experimental evaluation could be improved by including a broader range of baselines for discussion and comparison.
2. The current framework does not consider the varying reliability of information across domains, such as in heterophily graphs, which may limit its applicability in real-world heterophily settings.


Overall, this paper presents a novel and compelling approach to unifying learning paradigms across graph tasks. For the camera-ready version, I recommend the authors further refine some aspects, particularly by incorporating more competitive baselines. Additionally, exploring more challenging settings—-such as unifying a wider range of graph properties or addressing heterophily structures—-is also recommended to enhance the generality and applicability of the proposed method.